# Offline Multitask Representation Learning for Reinforcement Learning

**Haque Ishfaq**[*]
Mila, McGill University
haque.ishfaq@mail.mcgill.ca

**Thanh Nguyen-Tang**
Johns Hopkins University
nguyent@cs.jhu.edu

**Songtao Feng**
University of Florida
sfeng1@ufl.edu

**Raman Arora**
Johns Hopkins University
arora@cs.jhu.edu

**Mengdi Wang**
Princeton University
mengdiw@princeton.edu

**Ming Yin**[*]
Princeton University
my0049@princeton.edu

**Doina Precup**[*]
Mila, McGill University
dprecup@cs.mcgill.ca

## Abstract

We study offline multitask representation learning in reinforcement learning (RL), where a learner is provided with an offline dataset from different tasks that share a common representation and is tasked to learn the shared representation. We theoretically investigate offline multitask low-rank RL, and propose a new algorithm called MORL for offline multitask representation learning. Furthermore, we examine downstream RL in reward-free, offline and online scenarios, where a new task is introduced to the agent that shares the same representation as the upstream offline tasks. Our theoretical results demonstrate the benefits of using the learned representation from the upstream offline task instead of directly learning the representation of the low-rank model.

## 1 Introduction

Recent advances in offline reinforcement learning (RL) (Levine et al., 2020) have opened up possibilities for training policies for real-world problems using pre-collected datasets, such as robotics (Kalashnikov et al., 2018; Rafailov et al., 2021; Kalashnikov et al., 2021), natural language processing (Jaques et al., 2019), education (De Lima and Krohling, 2021), electricity supply (Zhan et al., 2022) and healthcare (Guez et al., 2008; Shortreed et al., 2011; Wang et al., 2018; Killian et al., 2020). While most offline RL studies focused on single-task problems, there are many practical scenarios where multiple tasks are correlated and it is beneficial to learn multiple tasks jointly by utilizing all of the data available (Kalashnikov et al., 2018; Yu et al., 2021, 2022; Xie and Finn, 2022). One popular approach in such cases is multitask representation learning, where the agent aims to tackle the problem by extracting a shared low-dimensional representation function among related tasks and then using a simple function (e.g., linear) on top of this common representation to solve each task (Caruana, 1997; Baxter, 2000). Despite the empirical success of multitask representation learning, particularly in reinforcement learning for its efficacy in reducing the sample complexity (Teh et al., 2017; Sodhani et al., 2021; Arulkumaran et al., 2022), the theoretical understanding of it is still in its early stages (Brunskill and Li, 2013; Calandriello et al., 2014; Arora et al., 2020; D'Eramo et al., 2020; Hu et al., 2021; Lu et al., 2021; Müller and Pacchiano, 2022). Although some works

---

[*]The corresponding authors.

38th Conference on Neural Information Processing Systems (NeurIPS 2024).

theoretically studied the online multitask representation learning for RL where the agent is allowed to interact with multiple source tasks to learn the shared representation (Cheng et al., 2022; Agarwal et al., 2023; Sam et al., 2024), there is currently no theoretical understanding on the effectiveness of multitask RL in the *offline* setting. This is crucial as in many practical scenarios (Kumar et al., 2022; Yoo et al., 2022; Lin et al., 2022), it is not feasible to interact with the different task environments in an online manner.

Moreover, when the tasks share the same representation, offline multitask representation learning can serve as a launchpad for effectively solving many other downstream tasks (Kumar et al., 2023). Consider the problem of learning to control robotic arms where we may already have offline datasets from different pick-and-place tasks in a kitchen such as the Bridge Dataset (Ebert et al., 2022). From this one can consider many possible downstream RL tasks where representation learned from these offline datasets can be beneficial. For example, one may consider solving a new pick-and-place task with different previously unseen objects in either an online or offline manner. Alternatively, one may consider a downstream reward-free RL (Jin et al., 2020a) task where the agent would first gather additional novel and diverse data without a pre-specified reward function and afterward, when provided with any reward function (e.g. slightly different target placing spot for the picked object), would be asked to provide a good policy without additional interaction.

In this work, we study the provable benefits of offline multi-task representation learning for RL in which the learner is only given access to pre-collected data from different source tasks which are modeled by low-rank MDPs (Agarwal et al., 2020b) with a shared (yet unknown) representation.

**Our contributions.** We develop a new offline multitask reinforcement learning algorithm that enables sample efficient representation learning in low-rank MDPs (Agarwal et al., 2020b) and further provide improved sample complexity to the downstream learning. In summary, our main contributions are:

- We propose a new offline multitask representation learning algorithm called Multitask Offline Representation Learning (MORL) under low-rank MDPs. MORL represents a standard training procedure in modern machine learning, by pooling the data from all source tasks to learn a shared representation of the dynamics via maximum likelihood estimation oracle.

- We prove that, MORL can learn a near-accurate model, and, when combined with the pessimism principle, find a near-optimal policy for each of the source tasks $T$ in the average sense, more sample-efficiently, than learning each task in isolation. To our knowledge, this is the first theoretical result demonstrating the benefit of representation learning in offline multitask RL.

- We then show theoretical benefits of using the learned representation from MORL in downstream reward-free RL Jin et al. (2020a); Wang et al. (2020). In particular, we show that, to guarantee an $\epsilon$-suboptimal policy for uniformly over any reward function, our algorithm requires at most $\widetilde{O}\left(\frac{H^4 d^3}{\epsilon^2}\right)$ episodes during the exploration phase where $d$ is the dimension of the feature and $H$ is the planning horizon. This improves the best known sample complexity for the reward-free RL in low-rank MDP (Cheng et al., 2023) by a factor of $\widetilde{O}(HdK)$, where $K$ is the cardinality of the action space. In addition, as a complementary result, we show that using the learned representation from MORL improves the suboptimality gap bound in both offline and online downstream task.

## 2 Preliminary

**Episodic MDP.** We consider an episodic discrete-time Markov Decision Process (MDP), denoted by $\mathcal{M} = (\mathcal{S}, \mathcal{A}, H, P, r)$, where $\mathcal{S}$ is the state space, $\mathcal{A}$ is the action space with cardinality $K$, $H$ is the finite episode length, $P = \{P_h\}_{h=1}^H$ are the state transition probability distributions with $P_h : \mathcal{S} \times \mathcal{A} \to \Delta(\mathcal{S})$, and $r = \{r_h\}_{h=1}^H$ are the deterministic reward functions with $r_h : \mathcal{S} \times \mathcal{A} \to [0, 1]$. Following prior work (Jiang et al., 2017; Sun et al., 2019), we assume that the initial state $s_1$ is fixed for each episode. A policy $\pi$ is a collection of $H$ functions $\{\pi_h : \mathcal{S} \to \mathcal{A}\}_{h \in [H]}$ where $\pi_h(s)$ is the action that the agent takes at state $s$ and at the $h$-th step in the episode. Given a starting state $s_h$, $s_{h'} \sim (P, \pi)$ denotes a state sampled by executing policy $\pi$ under the transition model $P$ for $h' - h$ steps and $\mathbb{E}_{(s_h, a_h) \sim (P, \pi)}[\cdot]$ denotes the expectation over states $s_h \sim (P, \pi)$ and actions $a_h \sim \pi$. Moreover, for each $h \in [H]$, we define the value function under policy $\pi$ when starting from an

arbitrary state $s_h = s$ at the $h$-th time step as

$$V_{h,P,r}^\pi(s) = \mathbb{E}_{(s_{h'},a_{h'}) \sim (P,\pi)}\left[\sum_{h'=h}^{H} r_{h'}(s_{h'}, a_{h'})|s_h = s\right].$$

We define the action-value function for a given state-action pair $(s,a)$ under policy $\pi$ at step $h$ as

$$Q_{h,P,r}^\pi(s,a) = \mathbb{E}_{(s_{h'},a_{h'}) \sim (P,\pi)}\left[\sum_{h'=h}^{H} r_{h'}(s_{h'}, a_{h'})|s_h = s, a_h = a\right].$$

Defining $(P_h f)(s,a) = \mathbb{E}_{s' \sim P(\cdot|s,a)}[f(s')]$ for any function $f : \mathcal{S} \to \mathbb{R}$, we write the Bellman equation associated with a policy $\pi$ as

$$Q_{h,P,r}^\pi(s,a) = (r_h + P_h V_{h+1,P,r}^\pi)(s,a), \ \ V_{h,P,r}^\pi(s) = Q_{h,P,r}^\pi(s, \pi_h(s)), \ \ V_{H+1,P,r}^\pi(s) = 0. \ \ (2.1)$$

Since the MDP begins with the same initial state $s_1$, for simplicity, we use $V_{P,r}^\pi$ to denote $V_{1,P,r}^\pi(s_1)$. Another useful concept is the notion of occupancy measure of a policy $\pi$ at time step $h$ under transition kernel $P$. Specifically, we use $d_{P_h}^\pi(s,a)$ to denote the marginal probability of encountering the state-action pair $(s,a)$ at time step $h$ when executing policy $\pi$ under MDP with transition kernel $P$. Finally, we denote $\mathcal{U}(\mathcal{S})$ and $\mathcal{U}(\mathcal{S}, \mathcal{A})$ as the uniform distribution over $\mathcal{S}$ and $\mathcal{S} \times \mathcal{A}$ respectively.

We study low-rank MDPs (Jiang et al., 2017; Agarwal et al., 2020b) defined as follows.

**Definition 2.1** (Low-rank MDPs). A transition kernel $P_h^* : \mathcal{S} \times \mathcal{A} \to \Delta(\mathcal{S})$ admits a low-rank decomposition with dimension $d \in \mathbb{N}$ if there exists two unknown embedding functions $\phi_h^* : \mathcal{S} \times \mathcal{A} \to \mathbb{R}^d$ and $\mu_h^* : \mathcal{S} \to \mathbb{R}^d$ such that for all $s, s' \in \mathcal{S}$ and $a \in \mathcal{A}$, $P_h^*(s'\,|\,s,a) = \langle \phi_h^*(s,a), \mu_h^*(s') \rangle$. Without loss of generality, we assume $\|\phi_h^*(s,a)\|_2 \le 1$ for all $(s,a) \in \mathcal{S} \times \mathcal{A}$ and for any function $g : \mathcal{S} \to [0,1]$, $\|\int \mu_h^*(s)g(s)ds\|_2 \le \sqrt{d}$.

We remark that the upper bounds on the norm of $\phi^*$ and $\mu^*$ are just for normalization. As the function class $\Phi$ for $\phi^*$ can be a non-linear, flexible function class, the low-rank MDP generalizes prior works with linear representations (Jin et al., 2020b; Hu et al., 2021) where it is assumed that the true representation $\phi^*$ is known to the agent a priori.

## 2.1 Offline Multitask RL with Downstream Learning

In **offline multitask RL** upstream learning, the agent is provided with an offline dataset collected from $T$ source tasks, where the reward functions $\{r^t\}_{t \in [T]}$ are assumed to be known. Each task $t \in [T]$ is associated with a low-rank MDP $\mathcal{M}^t = (\mathcal{S}, \mathcal{A}, H, P^t, r^t)$. Here, all $T$ tasks are identical except for (1) their true transition model $P^{(*,t)}$, which admits a low-rank decomposition with dimension $d$: $P_h^{(*,t)}(s_h'\,|\,s_h, a_h) = \langle \phi_h^*(s_h, a_h), \mu_h^{(*,t)}(s_h') \rangle$ for all $h \in [H], t \in [T]$, and (2) their reward $r_h^t$. While the tasks may differ in $\mu_h^{(*,t)}$ and $r_h^t$, we emphasize that all tasks share the same feature function $\phi_h^*$. We have access to offline dataset $\mathcal{D} = \bigcup_{t \in [T], h \in [H]} \mathcal{D}_h^{(t)}$, where $\mathcal{D}_h^{(t)} = \{(s_h^{(i,t)}, a_h^{(i,t)}, r_h^{(i,t)}, s_{h+1}^{(i,t)})\}_{i \in [n]}$ with $s_{h+1}^{(i,t)} \sim P_h^{(*,t)}(\cdot\,|\,s_h^{(i,t)}, a_h^{(i,t)})$ and $\mathcal{D}_h^{(t)}$ was collected using a *fixed behavior policy* $\pi_t^b$. In the upstream learning stage, the goal is to find a near-optimal policy and a near-accurate model for any task $t \in [T]$ and any reward function $\{r_t\}_{t \in [T]}$ through the use of offline dataset $\mathcal{D}$ and provide a well-learned representation for the downstream task. In order to achieve bounded sample complexity in offline RL, we need additional coverage assumption on the behavior policy $\pi_t^b$. One common coverage assumption is the global coverage assumption (Antos et al., 2008; Munos and Szepesvári, 2008), which assumes the occupancy measure under the behavior policy $\pi_t^b$ globally covers the the occupancy measure under any possible policies, i.e., the concentrability ratio satisfies, $\max_{\pi,s,a} d_{P_h^{(*,t)}}^\pi(s,a)/d_{P_h^{(*,t)}}^{\pi_t^b}(s,a) < \infty$. Instead, we make a partial coverage assumption and our suboptimality bound scales with the relative condition number (Agarwal et al., 2020a, 2021) instead of the global concentrability ratio, where the former can be substantially smaller than the latter. Under this assumption, we want to compete against any comparator policy covered by the offline data. In Section 3.2, we define the partial coverage condition using relative condition number, which was previously used in the context of single-task offline RL (Uehara and Sun, 2021; Uehara et al., 2022) and generalize it to offline multitask setting.

---

**Algorithm 1** Multitask Offline Representation Learning (MORL)

1: **Input:**
   Dataset $\mathcal{D} = \{(s_h^{(i,t)}, a_h^{(i,t)}, r_h^{(i,t)}, s_{h+1}^{(i,t)})\}_{i \in [n], h \in [H], t \in [T]}$, Regularizer $\lambda$, Parameter $\alpha$, Models
   $\{(\phi, \mu) : \phi \in \Psi, \mu \in \Psi\}$
2: **for** $h = 1, \ldots, H$ **do**
3:     Learn $\left(\widehat{\phi}_h, \widehat{\mu}_h^{(1)}, \ldots, \widehat{\mu}_h^{(T)}\right)$ via MLE $\left(\bigcup_{t \in [T]} \mathcal{D}_h^{(t)}\right)$ as (3.1)
4: **end for**
5: **for** $t = 1, \ldots, T$ **do**
6:     Update estimated transitioned kernels $\widehat{P}_h^{(t)}(\cdot \mid \cdot, \cdot)$ as (3.2), empirical covariance matrix $\widehat{\Sigma}_h^{(t)}$
       as (3.3) and penalty term $\widehat{b}_h^{(t)}$ as (3.4)
7:     Get policy $\widehat{\pi}_t = \operatorname{argmax}_\pi V_{\widehat{P}^{(t)}, r^t - \widehat{b}^{(t)}}$
8: **end for**
9: **Output:** $\widehat{\phi}, \widehat{P}^{(1)}, \ldots, \widehat{P}^{(T)}, \widehat{\pi}_1, \ldots, \widehat{\pi}_T$

---

In **downstream learning** stage, a new target task $T + 1$ with a low-rank transition kernel $P^{(*, T+1)}$ and the same $\mathcal{S}$, $\mathcal{A}$ and $H$ is assigned to the agent. The transition kernel $P^{(*, T+1)}$ shares the same representation $\phi^*$ with the $T$ upstream tasks, but has a task-distinct $\mu^{(*, T+1)}$. We consider three settings for downstream tasks – reward-free, offline and online RL, where the agent needs to use the representation function $\widehat{\phi}$ learned during the upstream stage to interact with the new task environment.

In the reward-free setting, firstly proposed in Jin et al. (2020a), the agent first interacts with the new task environment without accessing the reward function in the exploration phase for up to $K_{\mathrm{RFE}}$ episodes. Afterwards, it is provided with a reward function $r = \{r_h\}_{h=1}^H$ and asked to output an $\epsilon$-optimal policy $\pi$ in the planning phase. We define the sample complexity to be the number of episodes $K_{\mathrm{RFE}}$ required in the exploration phase to output an $\epsilon$- optimal policy $\pi$ in the planning phase for any given reward function $r$.

In the offline and online setting the downstream task $T + 1$ is already assigned with an unknown reward function $r^{T+1}$ and the goal is to find a near-optimal policy for the new task. The agent is expected to expedite its downstream learning through using the representation learned from the offline upstream task. In the online setting, it is allowed to interact with the new task environment and in the offline setting, it is instead provided with an offline dataset $\mathcal{D}_{\mathrm{off}} = \bigcup_{h \in [H]} \mathcal{D}_h$, where $\mathcal{D}_h = \{(s_h^\tau, a_h^\tau, r_h^\tau, s_{h+1}^\tau)\}_{\tau \in [N_{\mathrm{off}}]}$ and $\mathcal{D}_{\mathrm{off}}$ were collected using some behavior policy $\rho$.

## 3 Upstream Offline Multitask Representation Learning

In this section, we introduce our algorithm Multitask Offline Reinforcement Learning (MORL) designed for upstream offline multitask RL in low-rank MDPs and describe its theoretical properties.

### 3.1 Algorithm Design

The details of the algorithm MORL is depicted in Algorithm 1. The agent passes all input offline data to estimate low-rank components $\widehat{\phi}_h, \widehat{\mu}_h^{(1)}, \ldots, \widehat{\mu}_h^{(T)}$ simultaneously via the Maximum Likelihood Estimation (MLE) oracle $MLE \left(\bigcup_{t \in [T]} \mathcal{D}_h^{(t)}\right)$ on the joint distribution defined as follows:

$$\left(\widehat{\phi}_h, \widehat{\mu}_h^{(1)}, \ldots, \widehat{\mu}_h^{(T)}\right) = \operatorname*{argmax}_{\substack{\phi_h \in \Phi, \\ \mu_h^{(1)}, \ldots, \mu_h^{(T)} \in \Psi}} \sum_{i=1}^n \sum_{t=1}^T \log \left(\left\langle \phi_h(s_h^{(i,t)}, a_h^{(i,t)}), \mu_h^t(s_{h+1}^{(i,t)}) \right\rangle\right). \quad (3.1)$$

The MLE oracle in (3.1) is the offline multitask counterpart to the celebrated MLE oracle in the online multitask RL (Agarwal et al., 2020b; Cheng et al., 2022; Agarwal et al., 2023). The MLE oracle can be reasonably approximated in practice whenever optimizing over $\Phi$ and $\Psi$ is feasible through proper parameterization such as by neural network. For each task $t$, we obtain the estimated

transition kernel $\widehat{P}^{(t)}$ at each step $h$ using the learned embeddings $\widehat{\phi}_h, \widehat{\mu}_h^{(t)}$:

$$\widehat{P}_h^{(t)}(s' \mid s, a) = \langle \widehat{\phi}_h(s, a), \widehat{\mu}_h^{(t)}(s') \rangle. \tag{3.2}$$

Using the representation estimator $\widehat{\phi}_h$, we set the empirical covariance matrix $\widehat{\Sigma}_{h,\widehat{\phi}}^{(t)}$ for task $t$ as

$$\widehat{\Sigma}_{h,\widehat{\phi}}^{(t)} = \sum_{i=1}^{n} \widehat{\phi}_h(s_h^{(i,t)}, a_h^{(i,t)}) \widehat{\phi}_h(s_h^{(i,t)}, a_h^{(i,t)})^\top + \lambda I. \tag{3.3}$$

Using both $\widehat{\phi}_h$ and $\widehat{\Sigma}_{h,\widehat{\phi}}^{(t)}$, we construct a lower confidence bound penalty term as follows:

$$\widehat{b}_h^{(t)}(s_h, a_h) = \min \left\{ \alpha \|\widehat{\phi}_h(s_h, a_h)\|_{(\widehat{\Sigma}_{h,\widehat{\phi}}^{(t)})^{-1}}, 1 \right\}, \tag{3.4}$$

where $\alpha$ is a pre-determined parameter.

Finally, for each task $t$, with the learned model $\widehat{P}^{(t)}$ and the reward $r^t - \widehat{b}^{(t)}$, we do planning to get policy $\widehat{\pi}_t$.

## 3.2 Theoretical Result on Upstream Task

To facilitate the model selection task using the joint MLE oracle in (3.1), we posit a realizability assumption which is standard in low-rank MDP literature (Agarwal et al., 2020b; Cheng et al., 2022).

**Assumption 3.1** (Realizability). A learning agent has access to a model class $\{\Phi, \Psi\}$ that contains the true model, i.e., for any $h \in [H], t \in [T]$, the embeddings $\phi_h^* \in \Phi, \mu^{(*,t)} \in \Psi$. For normalization, we assume that for any $\phi \in \Phi, \|\phi(s, a)\|_2 \leq 1$ and for any $\mu \in \Psi$ and any function $g : \mathcal{S} :\to [0, 1]$, $\|\int \mu_h(s)g(s)ds\|_2 \leq \sqrt{d}$.

For simplicity, we assume that the cardinality of the function classes $\Phi$ and $\Psi$ are finite.

Next, we define the multitask relative condition number $C^*$, which is a natural extension of the standard relative condition number (Agarwal et al., 2020a, 2021; Uehara et al., 2022).

**Definition 3.2** (Multi-task relative condition number). For task $t$ and time step $h$, we define $C_{t,h}^*(\pi_t, \pi_t^b)$ as the relative condition number under $\phi_h^*$:

$$C_{t,h}^*(\pi_t, \pi_t^b) := \sup_{x \in \mathbb{R}^d} \frac{x^\top \mathbb{E}_{(s_h, a_h) \sim (P^{(*,t)}, \pi_t)}[\phi_h^*(s_h, a_h)\phi_h^*(s_h, a_h)^\top]x}{x^\top \mathbb{E}_{(s_h, a_h) \sim (P^{(*,t)}, \pi_t^b)}[\phi_h^*(s_h, a_h)\phi_h^*(s_h, a_h)^\top]x}. \tag{3.5}$$

We define $C_t^* := \max_{h \in [H]} C_{t,h}^*(\pi_t, \pi_t^b)$ and $C^* := \max_{t \in [T]} C_t^*$.

Intuitively, $C_{t,h}^*(\pi_t, \pi_t^b)$ defined in (3.5) measures the deviation between a comparator policy $\pi_t$ and the behavior policy $\pi_t^b$ at time step $h$. When tabular MDP is considered (i.e., $\phi_h^*$ is a one-hot encoding vector), this relative condition number reduces to the density ratio based single-policy concentrability coefficient, $C_{t,h,\infty}^*(\pi_t, \pi_t^b) = \max_{s,a} d_{P_h^{(*,t)}}^{\pi_t}(s, a)/d_{P_h^{(*,t)}}^{\pi_t^b}(s, a)$ (Chen and Jiang, 2022). The relative condition number $C_{t,h}^*(\pi_t, \pi_t^b)$ is always bounded by the concentrability coefficient (Uehara and Sun, 2021). In addition, the relative condition number is computed under the averaging over the state, actions, which could be much smaller than $\max_{s,a} d_{P_h^{(*,t)}}^{\pi_t}(s, a)/d_{P_h^{(*,t)}}^{\pi_t^b}(s, a)$ since the latter will be very large as long as any $s, a$ pair gives large $d_{P_h^{(*,t)}}^{\pi_t}(s, a)/d_{P_h^{(*,t)}}^{\pi_t^b}(s, a)$ ratio. Therefore, our $C_t^*$ could be much smaller compared to the concentrability coefficient, especially in large-scale MDPs (e.g. continuous state space). Moreover, in our definition of relative condition number, we use the unknown true representation $\phi^*$. Finally, we generalize single task relative condition number to multitask setting by defining $C^*$, by simply taking the maximum over the single-task relative condition numbers.

Now we describe our main theorem.

**Theorem 3.3.** *(a) Under Assumption 3.1, with probability at least $1 - \delta$, for any step $h \in [H]$, we have*

$$\frac{1}{T}\sum_{t=1}^{T}\mathbb{E}_{\substack{(s_h, a_h) \\ \sim (P^{(*,t)}, \pi_t^b)}}\left[\left\|\widehat{P}_h^{(t)}(\cdot \mid s_h, a_h) - P_h^{(*,t)}(\cdot \mid s_h, a_h)\right\|_{TV}\right] \leq \sqrt{\frac{2\log(2|\Phi||\Psi|^T nH/\delta)}{nT}},$$
(3.6)

*where $\widehat{\phi}, \widehat{P}^{(1)}, \ldots, \widehat{P}^{(T)}$ be the output of Algorithm 1.*

*(b) In addition, in Algorithm 1, if we set $\alpha = \sqrt{2n\omega\zeta_n + \lambda d}$, $\lambda = cd\log(|\Phi||\Psi|^T nH/\delta)$ with $\zeta_n := \frac{2\log(2|\Phi||\Psi|^T nH/\delta)}{n}$ and $c$ being a constant, where we assume that $\omega := \max_t \max_{s,a}(1/\pi_t^b(a \mid s)) < \infty$, then under Assumption 3.1, with probability at least $1 - \delta$, we have*

$$\frac{1}{T}\sum_{t=1}^{T}\left[V_{P^{(*,t)}, r^t}^{\pi_t} - V_{P^{(*,t)}, r^t}^{\widehat{\pi}_t}\right] \leq \omega\alpha dH\sqrt{\frac{C^*}{n}} + 2dH^2\sqrt{\frac{\lambda C^*}{n}} + \omega H^2\sqrt{\frac{dC^*\zeta_n}{T}} + \alpha\sqrt{\frac{d}{n}} + 2H\sqrt{\frac{\omega\zeta_n}{T}},$$
(3.7)

*where $\{\widehat{\pi}_t\}_{t\in[T]}$ is the output of Algorithm 1.*

Theorem 3.3 (a) shows a potential benefit of a joint learning of the source-task transition kernels, as compared to independent learning, as measured by the task-average TV distance (defined in the LHS of Equation (3.6)). In particular, to obtain an $\epsilon$-suboptimal transition kernels, it suffices for the joint learning to use $\widetilde{O}\left(\frac{\log|\Phi|}{T\epsilon^2} + \frac{\log|\Psi|}{\epsilon^2}\right)$ samples per task, yet for the independent learning to use $\widetilde{O}\left(\frac{\log|\Phi|}{\epsilon^2} + \frac{\log|\Psi|}{\epsilon^2}\right)$ samples per task. This benefit naturally comes from the inductive bias that all the source tasks share a representation in $\Phi$ which can be learned more accurately with the aggregated data pooled from all the source tasks. Equation (3.7) in Theorem 3.3 further shows that, for all tasks on average, we can uniformly compete with any set of comparator policies $\{\pi_t\}_{t\in[T]}$ satisfying the partial coverage through $C^* < \infty$. In particular, if the optimal policy $\pi_t^*$ is covered by the offline data for all $t \in [T]$, then the output $\{\widehat{\pi}_t\}_{t\in[T]}$ of Algorithm 1 is able to compete against it on average as well.

*Remark* 3.4. We note that in order for the bound to hold in Theorem 3.3, Algorithm 1 requires the knowledge of $\omega$ as it is required to set the value of $\alpha$ which is used in the lower confidence bound penalty term defined in (3.4). However, in practice, we expect that $\alpha$ can be treated as a hyperparameter that might be tuned using grid search.

**Proof outline.** Here, we highlight the key steps for the proof of Theorem 3.3. The detailed proof is deferred to Appendix D. Using offline multitask MLE lemma (Lemma F.3) and one-step back lemma (Lemma G.1), we first develop a new upper bound on model estimation error for each task which encapsulates the benefit of joint offline MLE model estimation over single-task offline learning. We then use the following lemma to show near pessimism in the average sense.

**Lemma 3.5.** *For any policy $\pi_t$ and reward $r^t$, we have, with probability $1 - \delta$*

$$\frac{1}{T}\sum_{t=1}^{T}\left[V_{\widehat{P}^{(t)}, r^t - \widehat{b}^{(t)}}^{\pi_t} - V_{P^{(*,t)}, r^t}^{\pi_t}\right] \leq H\sqrt{\omega\zeta_n/T},$$

*where $\zeta_n := \frac{2\log(2|\Phi||\Psi|^T nH/\delta)}{n}$.*

The proof of Lemma 3.5 relies on simulation lemma and a concentration argument for the penalty term defined in (3.4). Finally, to obtain a result only depending on the relative condition number using the true representation $\phi^*$ but not the learned feature $\widehat{\phi}$, we translate the penalty defined with $\widehat{\phi}$ to the potential function $\|\phi_h^*(s_h, a_h)\|_{(\Sigma_{h, \pi_t^b, \phi^*})^{-1}}$ where $\Sigma_{h, \pi_t^b, \phi^*} = n\mathbb{E}_{(s_h, a_h)\sim(P^{(*,t)}, \pi_t^b)}[\phi^*\phi^{*\top}] + \lambda I$ using a one-step back inequality (Lemma G.1) and a distribution shift lemma (Lemma O.2).

# 4 Downstream RL: Reward-free Exploration, Offline RL and Online RL

## 4.1 Relationship between upstream and downstream MDPs

In order for us to theoretically study the downstream RL tasks where we would use the learned feature $\widehat{\phi}$ from the upstream tasks, first we need to make certain connection between the upstream

and downstream MDPs. Naturally, we would have to resort to some assumptions on the transition kernels to make such connections. Next, we describe these assumptions, which we largely adopt from Cheng et al. (2022).

**Assumption 4.1.** We make the following assumptions

1. For each upstream source task $t$ with transition kernel $P^{(*,t)}$, the behavior policy $\pi_t^b$ is such that $\min_{s \in \mathcal{S}} \mathbb{P}_h^{(\pi_t^b,t)}(s) \geq \kappa$, where $\mathbb{P}_h^{(\pi_t^b,t)}(\cdot) : \mathcal{S} \to \mathbb{R}$ is the marginalized state occupancy measure over $\mathcal{S}$ using policy $\pi_t^b$ at time step $h$.

2. The state space $\mathcal{S}$ is compact and has finite measure $1/\nu$, and the induced uniform probability density function is $f(s) = \nu, s \in \mathcal{S}$.

3. For any two models $P^1(s'|s,a) = \langle \phi^1(s,a), \mu^1(s') \rangle$ and $P^2(s'|s,a) = \langle \phi^2(s,a), \mu^2(s') \rangle$ in the model class $\Phi \times \Psi$, we have,
   $$\|P^1(\cdot|s,a) - P^2(\cdot|s,a)\|_{\text{TV}} \leq C_R \mathbb{E}_{(s,a) \sim \mathcal{U}(\mathcal{S},\mathcal{A})} \|P^1(\cdot|s,a) - P^2(\cdot|s,a)\|_{\text{TV}},$$
   for all $(s,a) \in \mathcal{S} \times \mathcal{A}$ and $h \in [H]$ where $C_R$ is an absolute constant.

4. The transition kernel of task $T + 1$, $P^{(*,T+1)}$ can be $\xi$-approximated by a linear combination of the $T$ source upstream tasks, i.e. there exist $T$ unknown coefficients $c_1, \ldots, c_T \geq 0$ such that $\sum_{t=1}^{T} c_t \leq C_L$ and $\xi \geq 0$ such that for all $(s,a) \in \mathcal{S} \times \mathcal{A}$ and $h \in [H]$, we have
   $$\left\| P^{(*,T+1)}(\cdot|s,a) - \sum_{t=1}^{T} c_t P^{(*,t)}(\cdot|s,a) \right\|_{\text{TV}} \leq \xi.$$
   Here, $\xi$ is called the linear combination misspecification.

The first point in Assumption 4.1 ensures that the behavior policy $\pi_t^b$ for each task $t$ can reach any state in $\mathcal{S}$ at any time step with a positive probability, an assumption that is previously used in (Yin et al., 2021). Compared to Cheng et al. (2022), which assumes the existence of a policy with reachability property in each of the upstream online tasks, ours assumes reachability property for the behavior policies used to collect the upstream offline dataset.

The third point in Assumption 4.1 ensures that for each source task $t$, the point-wise TV error between the learned estimated transition kernel $\widehat{P}^{(t)}$ and the true transition kernel $P^{(*,t)}$ is bounded by the population-level TV error. This assumption is necessary to transfer the MLE error from the upstream source tasks to the downstream target task.

Finally, the fourth point in Assumption 4.1 connects the upstream source tasks with the downstream target task by assuming that the transition kernel of the target task $P^{(*,T+1)}$ can be approximated by a linear combination of transition kernels of $T$ upstream source tasks.

The precision of the feature estimation in the upstream has a significant impact on the downstream task's performance because the downstream task utilizes the estimated feature from the upstream. We use the following notion of $\epsilon$-approximate linear MDP to provide a guarantee for the estimated feature.

**Definition 4.2** ($\epsilon$-approximate linear MDP (Jin et al., 2020b; Cheng et al., 2022)). For any $\epsilon > 0$, we say that MDP $\mathcal{M} = (\mathcal{S}, \mathcal{A}, \mathcal{H}, P, r)$ is an $\epsilon$-approximate linear MDP with a feature map $\phi_h : \mathcal{S} \times \mathcal{A} :\to \mathbb{R}^d$, if for any $h \in [H]$, there exist $d$ unknown (signed) measures $\mu_h = (\mu_h^{(1)}, \ldots, \mu_h^{(d)})$ over $\mathcal{S}$ such that for any $(s,a) \in \mathcal{S} \times \mathcal{A}$, we have
$$\|P_h(\cdot|s,a) - \langle \phi_h(s,a), \mu_h(\cdot) \rangle\|_{\text{TV}} \leq \epsilon. \tag{4.1}$$
Any $\phi$ satisfying (4.1) is called an $\epsilon$-approximate feature map of $\mathcal{M}$.

The next lemma shows that the learned feature $\widehat{\phi}$ from the upstream offline tasks can approximate the true feature in the new downstream task.

**Lemma 4.3.** *Under Assumption 4.1, the output $\widehat{\phi}$ of Algorithm 1 is a $\xi_{down}$-approximate feature for MDP $\mathcal{M}^{T+1}$ where $\xi_{down} = \xi + \frac{C_L C_R \nu}{\kappa} \sqrt{\frac{2T \log(2|\Phi||\Psi|^T nH/\delta)}{n}}$, i.e. there exist a time-dependent unknown (signed) measure $\widehat{\mu}^*$ over $\mathcal{S}$ such that for any $(s,a) \in \mathcal{S} \times \mathcal{A}$, we have*
$$\|P_h^{(*,T+1)}(\cdot|s,a) - \langle \widehat{\phi}_h(s,a), \widehat{\mu}_h^*(\cdot) \rangle\|_{\text{TV}} \leq \xi_{down}.$$
*Furthermore, for any $g : \mathcal{S} \to [0,1]$, we have $\|\int \widehat{\mu}_h^*(s) g(s) ds\|_2 \leq C_L \sqrt{d}$.*

## 4.2 Downstream Reward-Free RL

Our goal in this part is to investigate the statistical efficiency of reward-free RL in low-rank MDP while having access to offline datasets from the upstream tasks.

Our algorithm for the reward-free setting is presented in Algorithm 2 (exploration phase) and Algorithm 3 (planning phase) which is built on the procedure of optimistic learning as Wang et al. (2020); Zhang et al. (2021). While having similar design principle as in Wang et al. (2020), our algorithm differs from them due to the misspecification of representation from the upstream task. Thus, the upstream learning error affects the learning accuracy and downstream suboptimality gap and we need to account for that in our analysis. Another difference with Wang et al. (2020) is that in the exploration phase, like Chen et al. (2022b), we construct more aggressive reward function to avoid overly-conservative exploration, which removes the extra dependency of sample complexity on episode length $H$. Below, we provide our main theorem for downstream reward-free RL task and defer the proof to Appendix I.

**Theorem 4.4.** *Under Assumption 4.1, after collecting $K_{RFE}$ trajectories during the exploration phase in Algorithm 2, with probability at least $1 - \delta$, the output of Algorithm 3, policy $\pi$ satisfies*

$$\mathbb{E}_{s_1 \sim \mu}[V_1^*(s_1, r) - V_1^\pi(s_1, r)] \leq c'\sqrt{d^3 H^4 \log(dK_{RFE}H/\delta)/K_{RFE}} + 6H^2 \xi_{down}. \quad (4.2)$$

*If the linear combination misspecification error $\xi$ in Assumption 4.1 satisfies $\widetilde{O}(\sqrt{d^3/K_{RFE}})$ and the number of trajectories in the offline dataset for each upstream task is at least $\widetilde{O}(TK_{RFE}/d^3)$, then, provided $K_{RFE}$ is at least $O(H^4 d^3 \log(dH\delta^{-1}\epsilon^{-1})/\epsilon^2)$, with probability $1 - \delta$, the policy $\pi$ will be an $\epsilon$-optimal policy for any given reward during the planning phase.*

We compare the above result with other algorithms developed for the reward-free RL under low-rank MDPs and summarize the comparison in Table 1. FLAMBE (Agarwal et al., 2020b) achieves a sample complexity of $\widetilde{O}(\frac{H^{22}d^7K^9}{\epsilon^{10}})$ whereas MOFFLE (Modi et al., 2024) achieves a sample complexity of $\widetilde{O}(\frac{H^7 d^{11} K^{14}}{\min\{\epsilon^2 \eta, \eta^5\}})$, where $\eta$ is a reachability probability to all states. More recently, Cheng et al. (2023) proposed RAFFLE which has the best-known sample complexity of $\widetilde{O}(\frac{H^5 d^4 K}{\epsilon^2})$.[2] Cheng et al. (2023) further shows that the dependence of sample complexity on action space cardinality $K$ is unavoidable when performing reward-free exploration in low-rank MDPs from which they conclude that it is strictly harder to find a near-optimal policy under low-rank MDPs than under linear MDPs. However, as we see from Theorem 4.4, by using estimated representation from the upstream offline datasets, we can avoid this dependence of sample complexity on $K$ and overall improve the sample complexity by $\widetilde{O}(HdK)$ compared to that of RAFFLE. Moreover, compared to standard linear MDP, where the true representation $\{\phi_h\}_{h=1}^H$ is known (Jin et al., 2020b), the suboptimality gap in (4.2) contains an additional term $H^2 \xi_{down}$ which is due to the upstream misspecification error $\xi_{down}$. When $\xi_{down}$ is small enough, our resulting sample complexity of $\widetilde{O}(\frac{H^4 d^3}{\epsilon^2})$ matches the reward-free exploration sample complexity for linear mixture MDPs (Chen et al., 2022b), which is worse off by only $\widetilde{O}(d)$ compared to the best known sample complexity of $\widetilde{O}(\frac{H^4 d^2}{\epsilon^2})$ (Hu et al., 2022) for linear MDP.

| Algorithm | Sample Complexity | Task |
|---|---|---|
| FLAMBE (Agarwal et al., 2020b) | $\widetilde{O}(\frac{H^{22}d^7K^9}{\epsilon^{10}})$ | Single task |
| MOFFLE (Modi et al., 2024) | $\widetilde{O}(\frac{H^7 d^{11} K^{14}}{\min\{\epsilon^2 \eta, \eta^5\}})$ | Single task |
| RAFFLE (Cheng et al., 2023) | $\widetilde{O}(\frac{H^5 d^4 K}{\epsilon^2})$ | Single task |
| This work (Algorithm 2 and Algorithm 3) | $\widetilde{O}(\frac{H^4 d^3}{\epsilon^2})$ | Multi-task |

Table 1: Sample complexities of different approaches to learning an $\epsilon$-optimal policy for the reward-free RL setting with low-rank MDPs.

---

[2] We rescale the result in (Cheng et al., 2023) by a factor of $H^2$ as we do not assume the sum of rewards to be within $[0, 1]$.

### 4.3 Downstream Offline and Online RL

For completeness, we also consider downstream offline and online RL which was previously studied in Cheng et al. (2022) to show the effectiveness of our offline representation. In both cases, we assume that the reward function $r^{T+1}$ in the downstream task $T + 1$ is linear with respect to the unknown feature $\phi^* : \mathcal{S} \times \mathcal{A} \to \mathbb{R}^d$. We emphasize that, unlike Cheng et al. (2022), we assume the reward function $r^{T+1}$ is unknown.

**Offline RL.** For downstream offline RL task, similar to Cheng et al. (2022), we use standard pessimistic value iteration algorithm (PEVI) (Jin et al., 2021) with approximate feature learned from upstream task. We make the following data-coverage type of assumption which is standard in the study of offline RL (Xie et al., 2021b; Wang et al., 2021; Yin et al., 2021). Moreover, this assumption has been shown to be necessary for sample efficient offline RL for tabular and linear MDPs (Wang et al., 2021; Yin and Wang, 2021).

**Assumption 4.5** (Feature coverage)**.** There exists an absolute constant $\kappa_\rho$ such that for all $h \in [H]$ and $\phi_h \in \Phi_h$, $\lambda_{\min}(\mathbb{E}_\rho[\phi_h(s_h, a_h)\phi_h(s_h, a_h)^\top | s_1 = s]) \geq \kappa_\rho$.

Next, we provide our result for the downstream offline RL task and defer the proof to Appendix K.

**Theorem 4.6.** *Under Assumption 4.1, setting $\lambda_d = 1$, $\beta = O(Hd\sqrt{\iota} + H\sqrt{dN_{off}}\xi_{down})$, where $\iota = \log(HdN_{off}\max(\xi_{down}, 1)/\delta)$, with probability at least $1 - \delta$, the suboptimality gap of Algorithm 4 is at most*

$$V_{P^{(*,T+1)},r}^{\pi^*}(s) - V_{P^{(*,T+1)},r}^{\widehat{\pi}}(s) \leq 2H^2\xi_{down} + 2\beta \sum_{h=1}^{H} \mathbb{E}_{\pi^*}\left[\left\|\widehat{\phi}_h(s_h, a_h)\right\|_{\Lambda_h^{-1}} \Big| s_1 = s\right]. \quad (4.3)$$

*Additionally if Assumption 4.5 holds, and the sample size satisfies $N_{off} \geq 40/\kappa_\rho \cdot \log(4dH/\delta)$, then with probability $1 - \delta$, we have,*

$$V_{P^{(*,T+1)},r}^{\pi^*}(s) - V_{P^{(*,T+1)},r}^{\widehat{\pi}}(s) \leq O\left(\kappa_\rho^{-1/2}H^2d\sqrt{\frac{\log(HdN_{off}\max(\xi_{down}, 1)/\delta)}{N_{off}}} + \kappa_\rho^{-1/2}H^2d^{1/2}\xi_{down}\right). \quad (4.4)$$

**Online RL.** For downstream online RL task, where the agent is allowed to interact with the new task MDP $\mathcal{M}^{T+1}$ for policy optimization, similar to Cheng et al. (2022), we use standard LSVI-UCB algorithm (Jin et al., 2020b) with approximate feature. We next provide our result for downstream online RL task and defer the proof to Appendix M.

**Theorem 4.7.** *Let $\widetilde{\pi}$ be the uniform mixture of $\pi^1, \ldots, \pi^{N_{on}}$ in Algorithm 5. Under Assumption 4.1, setting $\lambda = 1$, $\beta_n = O(Hd\sqrt{\iota_n(\delta)} + H\sqrt{dn}\xi_{down} + C_L\sqrt{Hd})$, where $\iota_n = \log(Hdn\max(\xi_{down}, 1)/\delta)$, with probability $1 - \delta$, the suboptimality gap of Algorithm 5 satisfies*

$$V_{P^{(*,T+1)},r}^* - V_{P^{(*,T+1)},r}^{\widetilde{\pi}} \leq \widetilde{O}(H^2d^{3/2}N_{on}^{-1/2} + H^2d\xi_{down}).$$

## 5 Related Work

**Offline Reinforcement Learning.** Offline RL (Ernst et al., 2005; Riedmiller, 2005; Lange et al., 2012; Levine et al., 2020) studies the problem of learning a policy from a static dataset without interacting with the environment. The key challenge in offline RL is the insufficient coverage of the dataset, due to the lack of exploration (Levine et al., 2020; Liu et al., 2020). One prevalent approach to address this challenge is the pessimism principle to penalize the estimated value of the under-covered state-action pairs. There have been extensive studies on incorporating pessimism into the development of different approaches in single-task offline RL, including model-based approach (Rashidinejad et al., 2021; Uehara and Sun, 2022; Jin et al., 2021; Yu et al., 2020; Xie et al., 2021b; Uehara et al., 2022; Yin et al., 2022), model-free approaches (Kumar et al., 2020; Wu et al., 2021; Bai et al., 2022; Ghasemipour et al., 2022; Yan et al., 2023; Nguyen-Tang et al., 2022, 2023; Nguyen-Tang and Arora, 2023), and policy-based approach (Rezaeifar et al., 2022; Xie et al., 2021a; Zanette et al., 2021b; Nguyen-Tang and Arora, 2024a,b). Our algorithm for upstream offline multitask RL is inspired by the uncertainty-based pessimism methods in single-task offline RL.

**Low-rank MDPs.** Agarwal et al. (2020b) initiates the study of low-rank MDPs. Uehara et al. (2022) proposed model-based algorithms for both online and offline RL, while Modi et al. (2024) put forward

a model-free algorithm for low-rank MDPs. Moreover, Du et al. (2019); Misra et al. (2020); Zhang et al. (2022) studied block MDPs, which is a special case of low-rank MDPs.

**Offline Data Sharing in RL.** There has been several empirical works that investigated the benefits of using offline datasets from multiple tasks to accelerate downstream learning (Eysenbach et al., 2020; Kalashnikov et al., 2021; Mitchell et al., 2021; Yu et al., 2021; Yoo et al., 2022). Yu et al. (2021) show that selectively sharing data between tasks can be helpful for offline multitask learning. For instance, earlier studies have investigated the development of data sharing strategies through human domain knowledge (Kalashnikov et al., 2021), inverse RL (Reddy et al., 2019; Eysenbach et al., 2020; Li et al., 2020), and estimated Q-values (Yu et al., 2021). More recently, Xu et al. (2023) uses offline dataset from diverse tasks to perform offline multitask pretraining of a world model which is then finetuned on a downstream target task. Hu et al. (2023) proposes a provably efficient self-supervised offline data-sharing algorithm for linear MDP. However, they assume access to reward-free data.

**Comparison to Cheng et al. (2022).** Closest to our work is Cheng et al. (2022) who studied online multitask RL. Their proposed REFUEL algorithm combines design principles from FLAMBE (Agarwal et al., 2020b) and REP-UCB (Uehara et al., 2022) and performs joint MLE based model learning while collecting data in an online manner. On the contrary, MORL first performs joint MLE based model learning using the offline dataset collected for each source task and then upon constructing penalty terms for each task, performs planning using pessimistic reward functions. While both works rely on an MLE oracle, first proposed in Agarwal et al. (2020b) for single-task RL, our proposed offline multitask MLE lemma (Lemma F.3) conveys fundamentally very different ideas compared to its online counterpart, Lemma 3 in Cheng et al. (2022). Lemma 3 in Cheng et al. (2022) says that when exploration policies for each upstream online source task is uniformly chosen, the summation of the estimation error of transition probability can be bounded with high probability. On the contrary, our Lemma F.3 states that when the offline datasets for each of the upstream offline source tasks are collected using respective behavior policies, the summation of the estimation error is bounded with high probability. For the downstream task, Cheng et al. (2022) studies only offline and online RL task, whereas our primary contribution in this part is in the study of downstream reward-free RL which has not been previously studied in the context of multitask representation learning. For completeness, we provide results in downstream offline and online RL setting as a complementary result. Moreover, unlike us, Cheng et al. (2022) assumes that the reward-function is known in the downstream task, which is a fairly strong assumption. Somewhat in a contrived manner, in Cheng et al. (2022), the reward-function is further assumed to be general and not necessarily linear in the feature which complicates their downstream analysis. On the contrary, we assume that the reward function is linear with respect to the feature. Finally, Cheng et al. (2022) assumes that for each episode in any task MDP, the sum of reward is normalized to be within $[0, 1]$. We do not make this assumption for fair comparison to literature on reward-free RL for low-rank MDPs.

**Comparison to Concurrent Work.** In a concurrent work, Bose et al. (2024) studies representational transfer in offline low-rank RL. In the upstream task, similar to ours, Bose et al. (2024) also uses an offline MLE oracle. Compared to our Theorem 3.3, where we provide bound for average accuracy of the estimated transition kernels of the upstream source tasks, Bose et al. (2024) provides bound for the sum of the point-wise errors in the transition dynamics averaged over the points in the source datasets. To bound the representational transfer error in the downstream target task, they introduce a notion of neighborhood occupancy density. Moreover, to connect the upstream tasks and the downstream target task, they make a pointwise linear span assumption from Agarwal et al. (2023). Finally, for the downstream target task, they only consider offline setting, whereas our primary focus and contribution is in the study of reward-free setting in the downstream task.

# 6 Conclusion

In this paper, we theoretically study multitask RL in the offline setting. We show that offline multitask representation learning is provably more sample efficient than learning each task individually. We further show the benefit of employing the learned representation from the upstream to learn a near-optimal policy of a new downstream task, in reward-free, offline and online setting, that shares the same representation. We believe our work will open up many promising directions for future work, for example, studying the general function class representation learning in offline multitask setting.

**Acknowledgments**

R. Arora's and T. Nguyen-Tang's research was supported, in part, by the NSF CAREER award IIS-1943251. M. Yin and M. Wang acknowledge the support by NSF IIS-2107304, NSF CPS-2312093, and ONR 1006977. The authors would like to thank Yingbin Liang and Yu-Xiang Wang for their helpful discussions.

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

# Contents

# A  Additional Related Work

**Reward-Free Exploration (RFE).** In reward-free RL setting, the agent does not have access to a reward function during exploration phase. However, the agent must propose a near-optimal policy for an arbitrary reward function revealed only after the initial exploration phase. This setting is particularly relevant when there are multiple reward functions of interest (Achiam et al., 2017) or in the batch RL setting (Ernst et al., 2005). In recent years, reward-free RL has been extensively studied in both tabular (Jin et al., 2020a; Kaufmann et al., 2021; Ménard et al., 2021) and linear function approximation (Wang et al., 2020; Zanette et al., 2020; Zhang et al., 2021; Hu et al., 2022; Wagenmaker et al., 2022) settings. Agarwal et al. (2020b); Modi et al. (2024); Chen et al. (2022a); Cheng et al. (2023) study reward-free RL in low-rank MDPs which is particularly interesting as here representation learning is interwined with reward-free exploration.

# B  Omitted Algorithms

## B.1  Algorithms for Downstream Reward-Free RL

---

**Algorithm 2** Downstream Reward-Free Exploration: Exploration Phase

---

1: **Input:** Feature $\widehat{\phi}$, Failure probability $\delta > 0$ and target accuracy $\varepsilon > 0$
2: $K_{\text{RFE}} \leftarrow c_K \cdot d^3 H^4 \log(dH\delta^{-1}\varepsilon^{-1})/\varepsilon^2$ for some $c_K > 0$
3: $\beta \leftarrow C_L H \sqrt{d} + dH\sqrt{\log(dK_{\text{RFE}}H \max(\xi_{\text{down}}, 1)/\delta)} + H\xi_{\text{down}}\sqrt{dK_{\text{RFE}}}$
4: **for** $k = 1, \ldots, K_{\text{RFE}}$ **do**
5: $\quad \widehat{Q}_{H+1}^k(\cdot, \cdot) \leftarrow 0, \widehat{V}_{H+1}^k(\cdot) \leftarrow 0$
6: $\quad$ **for** $h = H, H-1, \ldots, 1$ **do**
7: $\quad\quad \Lambda_h^k = \sum_{\tau=1}^{k-1} \widehat{\phi}_h(s_h^\tau, a_h^\tau)\widehat{\phi}_h(s_h^\tau, a_h^\tau)^\top + I_d$
8: $\quad\quad u_h^k(\cdot, \cdot) \leftarrow \beta\sqrt{\widehat{\phi}(\cdot, \cdot)^\top (\Lambda_h^k)^{-1}\widehat{\phi}(\cdot, \cdot)}$
9: $\quad\quad$ Define the exploration-driven reward function $r_h^k(\cdot, \cdot) \leftarrow u_h^k(\cdot, \cdot)$
10: $\quad\quad \widehat{w}_h^k = (\Lambda_h^k)^{-1}\sum_{\tau=1}^{k-1} \widehat{\phi}_h(s_h^\tau, a_h^\tau)\widehat{V}_{h+1}^k(s_{h+1}^\tau)$
11: $\quad\quad \widehat{Q}_h^k(\cdot, \cdot) = \min\{\widehat{\phi}_h(\cdot, \cdot)^\top \widehat{w}_h^k + r_h^k(\cdot, \cdot) + u_h^k(\cdot, \cdot), H\}$
12: $\quad\quad \widehat{V}_h^k(\cdot) = \max_{a\in\mathcal{A}} \widehat{Q}_h^k(\cdot, a)$
13: $\quad\quad \pi_h^k(\cdot) = \text{argmax}_{a\in\mathcal{A}} \widehat{Q}_h^k(\cdot, a)$
14: $\quad$ **end for**
15: $\quad$ Receive initial state $s_1^k \sim \mu$
16: $\quad$ **for** $h = 1, \ldots, H$ **do**
17: $\quad\quad$ Take action $a_h^k = \pi_h^k(s_h^k)$ and observe $s_{h+1}^k \sim P_h^{(*,T+1)}(s_h^k, a_h^k)$
18: $\quad$ **end for**
19: **end for**
20: **Output:** $\mathcal{D}_{\text{RFE}} \leftarrow \{(s_h^k, a_h^k)\}_{(k,h)\in[K]\times[H]}$

---

---

**Algorithm 3** Downstream Reward-Free Exploration: Planning Phase

---

1: **Input:** Feature $\widehat{\phi}$, dataset $\mathcal{D}_{\text{RFE}} = \{(s_h^k, a_h^k)\}_{k \in [K_{\text{RFE}}], h \in [H]}$, reward functions $r = \{r_h\}_{h \in [H]}$
2: **Initialization:** $\widehat{Q}_{H+1}(\cdot, \cdot) \leftarrow 0$ and $\widehat{V}_{H+1}(\cdot) \leftarrow 0$
3: **for** $h = H, H-1, \ldots, 1$ **do**
4: $\quad \Lambda_h = \sum_{\tau=1}^{K_{\text{RFE}}} \widehat{\phi}_h(s_h^\tau, a_h^\tau) \widehat{\phi}_h(s_h^\tau, a_h^\tau)^\top + I_d$
5: $\quad$ Let $u_h(\cdot, \cdot) \leftarrow \min \left\{ \beta \sqrt{\widehat{\phi}(\cdot, \cdot)^\top (\Lambda_h)^{-1} \widehat{\phi}(\cdot, \cdot)}, H \right\}$
6: $\quad \widehat{w}_h \leftarrow \Lambda_h^{-1} \sum_{\tau=1}^{K_{\text{RFE}}} \widehat{\phi}_h(s_h^\tau, a_h^\tau) \widehat{V}_{h+1}(s_{h+1}^\tau)$
7: $\quad \widehat{Q}_h(\cdot, \cdot) \leftarrow \min\{\widehat{\phi}_h(\cdot, \cdot)^\top \widehat{w}_h + r_h(\cdot, \cdot) + u_h(\cdot, \cdot), H\}$ and $\widehat{V}_h(\cdot) \leftarrow \max_{a \in \mathcal{A}} \widehat{Q}_h(\cdot, a)$
8: $\quad \widehat{\pi}_h(\cdot) \leftarrow \operatorname{argmax}_{a \in \mathcal{A}} \widehat{Q}_h(\cdot, a)$
9: **end for**
10: **Output:** $\widehat{\pi} = \{\widehat{\pi}_h\}_{h \in [H]}$

---

## B.2 Algorithm for Downstream Offline RL

---

**Algorithm 4** Pessimistic Value Iteration (PEVI) with Approximate Feature (Jin et al., 2021)

---

1: **Input:** Feature $\widehat{\phi}$, dataset $\mathcal{D}_{\text{down}} = \{(s_h^\tau, a_h^\tau, r_h^\tau, s_{h+1}^\tau)\}_{\tau \in [N_{\text{off}}], h \in [H]}$, parameters $\lambda_d, \beta, \xi_{\text{down}}$
2: **Initialization:** $\widehat{V}_{H+1} = 0$
3: **for** $h = H, H-1, \ldots, 1$ **do**
4: $\quad \Lambda_h = \sum_{\tau=1}^{N_{\text{off}}} \widehat{\phi}_h(s_h^\tau, a_h^\tau) \widehat{\phi}_h(s_h^\tau, a_h^\tau)^\top + \lambda_d I_d$
5: $\quad \widehat{w}_h = \Lambda_h^{-1} (\sum_{\tau=1}^{N_{\text{off}}} \widehat{\phi}_h(s_h^\tau, a_h^\tau) \cdot (r_h^\tau + \widehat{V}_{h+1}(s_{h+1}^\tau)))$
6: $\quad \Gamma_h(\cdot, \cdot) = H \xi_{\text{down}} + \beta [\widehat{\phi}_h^\top \Lambda_h^{-1} \widehat{\phi}_h(\cdot, \cdot)]^{1/2}$
7: $\quad \widehat{Q}_h(\cdot, \cdot) = \min\{\widehat{\phi}_h(\cdot, \cdot)^\top \widehat{w}_h - \Gamma_h(\cdot, \cdot), H - h + 1\}^+$
8: $\quad \widehat{\pi}_h(\cdot | \cdot) = \operatorname{argmax}_{\pi_h} \langle \widehat{Q}_h(\cdot, \cdot), \pi_h(\cdot | \cdot) \rangle_{\mathcal{A}}$
9: $\quad \widehat{V}_h(\cdot) = \langle \widehat{Q}_h(\cdot, \cdot), \widehat{\pi}_h(\cdot | \cdot) \rangle_{\mathcal{A}}$
10: **end for**
11: **Output:** $\{\widehat{\pi}_h\}_{h=1}^H$

---

## B.3 Algorithm for Downstream Online RL

---

**Algorithm 5** LSVI-UCB with Approximate Feature (Jin et al., 2020b)

---

1: **Input:** Feature $\widehat{\phi}$, parameters $\lambda_d, \beta_n, \xi_{\text{down}}$
2: **for** $n = 1, \ldots, N_{\text{on}}$ **do**
3: $\quad$ Receive the initial state $s_1^n = s_1$
4: $\quad$ **for** $h = H, H-1, \ldots, 1$ **do**
5: $\quad\quad \Lambda_h^n = \sum_{\tau=1}^{n-1} \widehat{\phi}_h(s_h^\tau, a_h^\tau) \widehat{\phi}_h(s_h^\tau, a_h^\tau)^\top + \lambda_d I_d$
6: $\quad\quad \widehat{w}_h^n = (\Lambda_h^n)^{-1} \sum_{\tau=1}^{n-1} \widehat{\phi}_h(s_h^\tau, a_h^\tau)[r_h(s_h^\tau, a_h^\tau) + \widehat{V}_{h+1}^n(s_{h+1}^\tau)]$
7: $\quad\quad \widehat{Q}_h^n(\cdot, \cdot) = \min\{\widehat{\phi}_h(\cdot, \cdot)^\top \widehat{w}_h^n + \beta_n \|\widehat{\phi}(\cdot, \cdot)\|_{(\Lambda_h^n)^{-1}}, H - h + 1\}^+$
8: $\quad\quad \widehat{V}_h^n(\cdot) = \max_{a \in \mathcal{A}} \widehat{Q}_h^n(\cdot, a)$
9: $\quad$ **end for**
10: $\quad \pi_h^n(\cdot) = \operatorname{argmax}_{a \in \mathcal{A}} \widehat{Q}_h^n(\cdot, a)$
11: $\quad$ **for** $h = 1, \ldots, H$ **do**
12: $\quad\quad$ Take action $a_h^n = \pi^n(s_h^n)$ and observe $s_{h+1}^n$
13: $\quad$ **end for**
14: **end for**
15: **Output:** $\pi^1, \ldots, \pi^{N_{\text{on}}}$ and $\widetilde{\pi}$ where $\widetilde{\pi}$ is the uniform mixture of $\pi^1, \ldots, \pi^{N_{on}}$

---

# C  Notations

We summarize frequently used notations in the following list.

$$f_h^{(t)}(s,a) \qquad \|\widehat{P}_h^{(t)}(\cdot\,|\,s,a) - P_h^{(\star,t)}(\cdot\,|\,s,a)\|_{TV}$$

$$\zeta_n \qquad \frac{2\log(2|\Phi||\Psi|^T nH/\delta)}{n}$$

$$\zeta_h^{(t)} \qquad \mathbb{E}_{(s_h,a_h)\sim(P^{(*,t)},\pi_t^b)}\left[f_h^{(t)}(s,a)^2\right]$$

$$\zeta_1^{(t)} \qquad \mathbb{E}_{(s_1,a_1)\sim(P^{(*,t)},\pi_t^b)}\left[f_1^{(t)}(s_1,a_1)^2\right]$$

$$\alpha_h^{(t)} \qquad \sqrt{n\omega\zeta_h^{(t)} + \lambda d + n\zeta_{h-1}^{(t)}}$$

$$\alpha \qquad \sqrt{2n\omega\zeta_n + \lambda d}$$

$$\Sigma_{h,\pi_t^b,\phi} \qquad n\mathbb{E}_{(s_h,a_h)\sim(P^{(*,t)},\pi_t^b)}[\phi\phi^\top] + \lambda I$$

$$\widehat{\Sigma}_{h,\phi}^{(t)} \qquad \sum_{i=1}^n \phi_h(s_h^{(i,t)}, a_h^{(i,t)})\phi_h(s_h^{(i,t)}, a_h^{(i,t)})^\top + \lambda I$$

$$\widehat{b}_h^{(t)}(s_h,a_h) \qquad \min\left\{\alpha\|\widehat{\phi}_h(s_h,a_h)\|_{\widehat{\Sigma}_{h,\widehat{\phi}}^{(t)}}, 1\right\}$$

$$\omega_t \qquad \max_{s,a}(1/\pi_t^b(a\,|\,s))$$

$$\omega \qquad \max_t \omega_t$$

$$\xi_{\text{down}} \qquad \xi + \frac{C_L C_R \nu}{\kappa}\sqrt{\frac{2T\log(2|\Phi||\Psi|^T nH/\delta)}{n}}$$

$$\widehat{\mu}^*(\cdot) \qquad \sum_{t=1}^T c_t\widehat{\mu}^{(t)}(\cdot)$$

$$\widehat{w}_h^* \qquad \int_{s'}\widehat{\mu}^*(s')\widehat{V}_{h+1}(s')ds'$$

For any $h \in [H]$, we define

$$P_h^{(*,T+1)}(\cdot|s,a) = \langle\phi_h^*(s,a), \mu_h^{(*,T+1)}(\cdot)\rangle,$$
$$\overline{P}_h(\cdot|s,a) = \langle\widehat{\phi}_h(s,a), \widehat{\mu}_h^*(\cdot)\rangle.$$

Given a reward function $r$, for any function $f : \mathcal{S} \to \mathbb{R}$ and $h \in [H]$, we define the transition operators and their corresponding Bellman operators as follows

$$(P_h^{(*,T+1)}f)(s,a) = \int_{s'}\langle\phi_h^*(s,a), \mu_h^{(*,T+1)}(s')\rangle f(s')ds',$$

$$(\mathbb{B}_h f)(s,a) = r_h(s,a) + (P_h^{(*,T+1)}f)(s,a)$$

$$(\overline{P}_h f)(s,a) = \int_{s'}\langle\widehat{\phi}_h(s,a), \widehat{\mu}_h^*(s')\rangle f(s')ds',$$

$$(\overline{B}_h f)(s,a) = r_h(s,a) + (\overline{P}_h f)(s,a)$$

# D Proof of Multitask Offline Representation Learning

We first state the supporting lemmas that are used in the proof of Theorem 3.3. The proof of these lemmas are provided in Appendix E.

## D.1 Supporting Lemmas

In the following lemma, we provide an upper bound for the model estimation error for each task that captures the advantage of joint MLE model estimation over single-task learning.

**Lemma D.1.** *For any task $t$, policy $\pi_t$, and reward $r_t$, we have for all $h \geq 2$,*

$$\mathbb{E}_{(s_h,a_h)\sim(\widehat{P}^{(t)},\pi_t)}\left[f_h^{(t)}(s_h,a_h)\right] \leq \mathbb{E}_{(s_{h-1},a_{h-1})\sim(\widehat{P}^{(t)},\pi_t)}\left[\alpha_h^{(t)}\left\|\widehat{\phi}_{h-1}(s_{h-1},a_{h-1})\right\|_{(\Sigma_{h-1,\pi_t^b,\widehat{\phi}})^{-1}}\right],$$

*and for $h = 1$, we have*

$$\mathbb{E}_{a_1\sim\pi_t}\left[f_1^{(t)}(s_1,a_1)\right] \leq \sqrt{\omega_t\zeta_1^{(t)}},$$

*where $\omega_t = \max_{s,a}(1/\pi_t^b(a\,|\,s))$.*

In the next lemma, we prove that $V_{\widehat{P}^{(t)},r^t-\widehat{b}^{(t)}}^{\pi_t}$ is an almost pessimistic estimator of $V_{P^{(*,t)},r^t}^{\pi_t}$ in the average sense.

**Lemma D.2** (Restatement of Lemma 3.5). *For any policy $\pi_t$ and reward $r^t$, we have, with probability $1 - \delta$*

$$\frac{1}{T}\sum_{t=1}^{T}\left[V_{\widehat{P}^{(t)},r^t-\widehat{b}^{(t)}}^{\pi_t} - V_{P^{(*,t)},r^t}^{\pi_t}\right] \leq H\sqrt{\omega\zeta_n/T}, \quad \text{where } \zeta_n := \frac{2\log(2|\Phi||\Psi|^T nH/\delta)}{n}$$

## D.2 Proof of Theorem 3.3

We first restate Theorem 3.3.

**Theorem D.3** (Restatement of Theorem 3.3). *Under Assumption 3.1, with probability at least $1 - \delta$, for any step $h \in [H]$, we have*

$$\frac{1}{T}\sum_{t=1}^{T}\mathbb{E}_{\substack{(s_h,a_h)\\\sim(P^{(*,t)},\pi_t^b)}}\left[\left\|\widehat{P}_h^{(t)}(\cdot\,|\,s_h,a_h) - P_h^{(*,t)}(\cdot\,|\,s_h,a_h)\right\|_{TV}\right] \leq \sqrt{\frac{2\log(2|\Phi||\Psi|^T nH/\delta)}{nT}}, \tag{D.1}$$

*where $\widehat{\phi},\widehat{P}^{(1)},\ldots,\widehat{P}^{(T)}$ be the output of Algorithm 1.*

*In addition, in Algorithm 1, if we set $\alpha = \sqrt{2n\omega\zeta_n + \lambda d}$, $\lambda = cd\log(|\Phi||\Psi|^T nH/\delta)$ with $\zeta_n := \frac{2\log(2|\Phi||\Psi|^T nH/\delta)}{n}$ and $c$ being a constant term, where we assume that $\omega := \max_t\max_{s,a}(1/\pi_t^b(a\,|\,s)) < \infty$, then under Assumption 3.1, with probability at least $1 - \delta$, we have*

$$\frac{1}{T}\sum_{t=1}^{T}\left[V_{P^{(*,t)},r^t}^{\pi_t} - V_{P^{(*,t)},r^t}^{\widehat{\pi}_t}\right] \leq \omega\alpha dH\sqrt{\frac{C^*}{n}} + 2dH^2\sqrt{\frac{\lambda C^*}{n}} + \omega H^2\sqrt{\frac{dC^*\zeta_n}{T}} + \alpha\sqrt{\frac{d}{n}} + 2H\sqrt{\frac{\omega\zeta_n}{T}}, \tag{D.2}$$

*where $\{\widehat{\pi}_t\}_{t\in[T]}$ is the output of Algorithm 1.*

*Proof of Theorem D.3.* As in Lemma D.2, we condition on the events:

$$\sum_{t=1}^{T}\mathbb{E}_{(s_h,a_h)\sim(P^{(*,t)},\pi_t^b)}\left[f_h^{(t)}(s,a)^2\right] \leq \zeta_n, \quad \text{where } \zeta_n := \frac{2\log(2|\Phi||\Psi|^T nH/\delta)}{n}, \tag{D.3}$$

and

$$\forall\phi\in\Phi : \|\phi_h(s,a)\|_{(\widehat{\Sigma}_{h,\phi}^{(t)})^{-1}} = \Theta\left(\|\phi_h(s,a)\|_{(\Sigma_{h,\pi_t^b,\phi})^{-1}}\right). \tag{D.4}$$

From Lemma F.3 and Lemma G.2, this event happens with probability $1 - \delta$. Conditioning on this event, we have

$$
\frac{1}{T} \sum_{t=1}^{T} \mathbb{E}_{(s_h, a_h) \sim (P^{(*,t)}, \pi_t^b)} \left[ \left\| \widehat{P}_h^{(t)}(\cdot \mid s_h, a_h) - P_h^{(*,t)}(\cdot \mid s_h, a_h) \right\|_{TV} \right]
$$

$$
= \frac{1}{T} \sum_{t=1}^{T} \mathbb{E}_{(s_h, a_h) \sim (P^{(*,t)}, \pi_t^b)} \left[ f_h^{(t)}(s, a) \right]
$$

$$
\overset{(i)}{\leq} \frac{1}{T} \sqrt{ T \sum_{t=1}^{T} \left( \mathbb{E}_{(s_h, a_h) \sim (P^{(*,t)}, \pi_t^b)} \left[ f_h^{(t)}(s, a) \right] \right)^2 }
$$

$$
\overset{(ii)}{\leq} \frac{1}{T} \sqrt{ T \sum_{t=1}^{T} \mathbb{E}_{(s_h, a_h) \sim (P^{(*,t)}, \pi_t^b)} \left[ f_h^{(t)}(s, a)^2 \right] }
$$

$$
\overset{(iii)}{\leq} \sqrt{ \frac{\zeta_n}{T} }
$$

$$
= \sqrt{ \frac{2 \log(2 |\Phi| |\Psi|^T n H / \delta)}{nT} },
$$

where $(i)$ follows from Cauchy-Schwarz inequality, $(ii)$ follows from Jensen's inequality and $(iii)$ follows from (D.3). This completes the first part of the proof.

Conditioning on the event in (D.3) and (D.4), for any set of policies $\pi_1, \ldots, \pi_T$, we have

$$
\sum_{t=1}^{T} \left[ V_{P^{(*,t)}, r^t}^{\pi_t} - V_{P^{(*,t)}, r^t}^{\widehat{\pi}_t} \right]
$$

$$
= \sum_{t=1}^{T} \left[ V_{P^{(*,t)}, r^t}^{\pi_t} - V_{\widehat{P}^{(t)}, r^t - \widehat{b}^{(t)}}^{\widehat{\pi}_t} + V_{\widehat{P}^{(t)}, r^t - \widehat{b}^{(t)}}^{\widehat{\pi}_t} - V_{P^{(*,t)}, r^t}^{\widehat{\pi}_t} \right]
$$

$$
\overset{(i)}{\leq} \sum_{t=1}^{T} \left[ V_{P^{(*,t)}, r^t}^{\pi_t} - V_{\widehat{P}^{(t)}, r^t - \widehat{b}^{(t)}}^{\pi_t} + V_{\widehat{P}^{(t)}, r^t - \widehat{b}^{(t)}}^{\widehat{\pi}_t} - V_{P^{(*,t)}, r^t}^{\widehat{\pi}_t} \right]
$$

$$
\overset{(ii)}{\leq} \sum_{t=1}^{T} \sum_{h=1}^{H} \mathbb{E}_{(s_h, a_h) \sim (P^{(*,t)}, \pi_t)} \left[ \widehat{b}_h^{(t)}(s_h, a_h) + (P_h^{(*,t)} - \widehat{P}_h^{(t)}) V_{h+1, \widehat{P}^{(t)}, r^t - \widehat{b}^{(t)}}^{\pi_t}(s_h, a_h) \right]
$$

$$
+ \sum_{t=1}^{T} \left[ V_{\widehat{P}^{(t)}, r^t - \widehat{b}^{(t)}}^{\widehat{\pi}_t} - V_{P^{(*,t)}, r^t}^{\widehat{\pi}_t} \right]
$$

$$
\overset{(iii)}{\leq} \sum_{t=1}^{T} \sum_{h=1}^{H} \mathbb{E}_{(s_h, a_h) \sim (P^{(*,t)}, \pi_t)} \left[ \widehat{b}_h^{(t)}(s_h, a_h) \right] + H \sum_{t=1}^{T} \sum_{h=1}^{H} \mathbb{E}_{(s_h, a_h) \sim (P^{(*,t)}, \pi_t)} \left[ f_h^{(t)}(s_h, a_h) \right]
$$

$$
+ H \sqrt{\omega T \zeta_n}, \tag{D.5}
$$

where $(i)$ follows from the observation that $\widehat{\pi}_t$ is the argmax over all Markovian policies as well as all history-dependent policies for $\widehat{P}^{(t)}$, $(ii)$ follows from the simulation lemma, Lemma O.1, $(iii)$ follows from the observation that $V_{h, \widehat{P}^{(t)}, r^t - \widehat{b}^{(t)}}^{\pi_t} \leq H$ and Lemma D.2.

Now, using Lemma G.1 (with setting $P = P^{(*,t)}$ and $\phi = \phi^*$) and noting that $|\widehat{b}_h^{(t)}|_\infty \leq 1$, for $h \geq 2$, we have

$$
\mathbb{E}_{(s_h, a_h) \sim (P^{(*,t)}, \pi_t)} \left[ \widehat{b}_h^{(t)}(s_h, a_h) \right]
$$

$$
\leq \mathbb{E}_{(s_{h-1}, a_{h-1}) \sim (P^{(*,t)}, \pi_t)} \left[ \left\| \phi_{h-1}^*(s_{h-1}, a_{h-1}) \right\|_{(\Sigma_{h-1, \pi_t^b, \phi^*})^{-1}} \times \right.
$$

$$
\left. \sqrt{ n \omega \mathbb{E}_{(s_h, a_h) \sim (P^{(*,t)}, \pi_t^b)} \left[ \widehat{b}_h^{(t)}(s, a) \right]^2 + \lambda d } \right] \tag{D.6}
$$

From (D.4), we have

$$
\begin{aligned}
&n\mathbb{E}_{(s_h,a_h)\sim(P^{(\star,t)},\pi_t^b)}\left[\widehat{b}_h^{(t)}(s,a)\right]^2\\
&\leq n\mathbb{E}_{(s_h,a_h)\sim(P^{(\star,t)},\pi_t^b)}\left[\min\left\{\alpha^2\|\widehat{\phi}_h(s_h,a_h)\|_{(\Sigma_{h,\pi_t^b,\widehat{\phi}})^{-1}}^2,1\right\}\right]\\
&\leq n\mathbb{E}_{(s_h,a_h)\sim(P^{(\star,t)},\pi_t^b)}\left[\alpha^2\|\widehat{\phi}_h(s_h,a_h)\|_{(\Sigma_{h,\pi_t^b,\widehat{\phi}})^{-1}}^2\right]\\
&\leq \alpha^2\operatorname{Tr}\left[n\mathbb{E}_{(s_h,a_h)\sim(P^{(\star,t)},\pi_t^b)}[\widehat{\phi}_h\widehat{\phi}_h^\top]\{n\mathbb{E}_{(s_h,a_h)\sim(P^{(\star,t)},\pi_t^b)}[\widehat{\phi}_h\widehat{\phi}_h^\top]+\lambda I\}^{-1}\right]\\
&\leq \alpha^2\operatorname{Tr}\left[\{n\mathbb{E}_{(s_h,a_h)\sim(P^{(\star,t)},\pi_t^b)}[\widehat{\phi}_h\widehat{\phi}_h^\top]+\lambda I\}\{n\mathbb{E}_{(s_h,a_h)\sim(P^{(\star,t)},\pi_t^b)}[\widehat{\phi}_h\widehat{\phi}_h^\top]+\lambda I\}^{-1}\right]\\
&= \alpha^2\operatorname{Tr}[I_d]\\
&= \alpha^2 d
\end{aligned}
\tag{D.7}
$$

Next, we upper bound $\mathbb{E}_{(s_{h-1},a_{h-1})\sim(P^{(*,t)},\pi_t)}\left[\|\phi_{h-1}^*(s_{h-1},a_{h-1})\|_{(\Sigma_{h-1,\pi_t^b,\phi^*})^{-1}}^2\right]$ as the following

$$
\begin{aligned}
&\mathbb{E}_{(s_{h-1},a_{h-1})\sim(P^{(*,t)},\pi_t)}\left[\|\phi_{h-1}^*(s_{h-1},a_{h-1})\|_{(\Sigma_{h-1,\pi_t^b,\phi^*})^{-1}}^2\right]\\
&\overset{(i)}{\leq} C_{t,h}^*(\pi_t,\pi_t^b)\mathbb{E}_{(s_{h-1},a_{h-1})\sim(P^{(*,t)},\pi_t^b)}\left[\|\phi_{h-1}^*(s_{h-1},a_{h-1})\|_{(\Sigma_{h-1,\pi_t^b,\phi^*})^{-1}}^2\right]\\
&\overset{(ii)}{\leq} \frac{dC_{t,h}^*(\pi_t,\pi_t^b)}{n}\\
&\leq \frac{dC^*}{n},
\end{aligned}
\tag{D.8}
$$

where $(i)$ follows from Lemma O.2 and $(ii)$ follows from similar steps as in (D.7).

Combining (D.6), (D.7) and (D.8), we get

$$\sum_{t=1}^{T} \mathbb{E}_{(s_h,a_h)\sim(P^{(*,t)},\pi_t)} \left[\widehat{b}_h^{(t)}(s_h,a_h)\right]$$

$$\leq \sum_{t=1}^{T} \mathbb{E}_{(s_{h-1},a_{h-1})\sim(P^{(*,t)},\pi_t)} \left[\|\phi_{h-1}^*(s_{h-1},a_{h-1})\|_{(\Sigma_{h-1,\pi_t^b,\phi^*})^{-1}} \times \right.$$
$$\left. \sqrt{n\omega\mathbb{E}_{(s_h,a_h)\sim(P^{(\star,t)},\pi_t^b)}\left[\widehat{b}_h^{(t)}(s,a)\right]^2 + \lambda d}\right]$$

$$\leq \sum_{t=1}^{T} \sqrt{\mathbb{E}_{(s_{h-1},a_{h-1})\sim(P^{(*,t)},\pi_t)} \left[\|\phi_{h-1}^*(s_{h-1},a_{h-1})\|_{(\Sigma_{h-1,\pi_t^b,\phi^*})^{-1}}^2\right]} \times$$
$$\sqrt{\mathbb{E}_{(s_{h-1},a_{h-1})\sim(P^{(*,t)},\pi_t)} \left[n\omega\mathbb{E}_{(s_h,a_h)\sim(P^{(\star,t)},\pi_t^b)}\left[\widehat{b}_h^{(t)}(s,a)\right]^2 + \lambda d\right]}$$

$$\leq \sum_{t=1}^{T} \sqrt{\frac{dC^*}{n}} \sqrt{\mathbb{E}_{(s_{h-1},a_{h-1})\sim(P^{(*,t)},\pi_t)} \left[n\omega\mathbb{E}_{(s_h,a_h)\sim(P^{(\star,t)},\pi_t^b)}\left[\widehat{b}_h^{(t)}(s,a)\right]^2\right] + \lambda d}$$

$$\leq \sqrt{\frac{dC^*}{n}} \left(\sum_{t=1}^{T} \sqrt{n\omega\mathbb{E}_{(s_{h-1},a_{h-1})\sim(P^{(*,t)},\pi_t)}\left[\mathbb{E}_{(s_h,a_h)\sim(P^{(\star,t)},\pi_t^b)}\left[\widehat{b}_h^{(t)}(s,a)\right]^2\right]} + T\sqrt{\lambda d}\right)$$

$$\leq \sqrt{\frac{dC^*}{n}} \left(\sum_{t=1}^{T} \sqrt{n\omega^2\mathbb{E}_{(s_h,a_h)\sim(P^{(\star,t)},\pi_t^b)}\left[\widehat{b}_h^{(t)}(s,a)\right]^2} + T\sqrt{\lambda d}\right)$$

$$= \omega\sqrt{dC^*}\sum_{t=1}^{T} \sqrt{\mathbb{E}_{(s_h,a_h)\sim(P^{(\star,t)},\pi_t^b)}\left[\widehat{b}_h^{(t)}(s,a)\right]^2} + Td\sqrt{\frac{\lambda C^*}{n}}$$

$$\leq \omega\sqrt{dC^*}\sqrt{T\sum_{t=1}^{T} \mathbb{E}_{(s_h,a_h)\sim(P^{(\star,t)},\pi_t^b)}\left[\widehat{b}_h^{(t)}(s,a)\right]^2} + Td\sqrt{\frac{\lambda C^*}{n}}$$

$$\leq \omega\sqrt{dC^*T}\sqrt{\frac{T\alpha^2 d}{n}} + Td\sqrt{\frac{\lambda C^*}{n}}$$

$$= \omega\alpha Td\sqrt{\frac{C^*}{n}} + Td\sqrt{\frac{\lambda C^*}{n}}. \tag{D.9}$$

Following similar steps as in (D.7), we can further show that

$$\mathbb{E}_{(s_1,a_1)\sim(P^{(*,t)},\pi_t)} \left[\widehat{b}_1^{(t)}(s_1,a_1)\right] \leq \sqrt{\mathbb{E}_{(s_1,a_1)\sim(P^{(*,t)},\pi_t)} \left[\widehat{b}_1^{(t)}(s_1,a_1)\right]^2} \leq \alpha\sqrt{\frac{d}{n}}.$$

Now noting $|f_h^{(t)}|_\infty \le 1$, for $h \ge 2$,we get

$$\sum_{t=1}^{T} \mathbb{E}_{(s_h,a_h)\sim(P^{(*,t)},\pi_t)} \left[ f_h^{(t)}(s_h,a_h) \right]$$

$$\le \sum_{t=1}^{T} \mathbb{E}_{(s_{h-1},a_{h-1})\sim(P^{(*,t)},\pi_t)} \left[ \|\phi_{h-1}^*(s_{h-1},a_{h-1})\|_{(\Sigma_{h-1,\pi_t^b,\phi^*})^{-1}} \times \right.$$
$$\left. \sqrt{n\omega \mathbb{E}_{(s_h,a_h)\sim(P^{(\star,t)},\pi_t^b)} \left[ f_h^{(t)}(s_h,a_h)^2 \right] + \lambda d} \right]$$

$$\le \sum_{t=1}^{T} \sqrt{\mathbb{E}_{(s_{h-1},a_{h-1})\sim(P^{(*,t)},\pi_t)} \left[ \|\phi_{h-1}^*(s_{h-1},a_{h-1})\|_{(\Sigma_{h-1,\pi_t^b,\phi^*})^{-1}}^2 \right]} \times$$
$$\sqrt{\mathbb{E}_{(s_{h-1},a_{h-1})\sim(P^{(*,t)},\pi_t)} \left[ n\omega \mathbb{E}_{(s_h,a_h)\sim(P^{(\star,t)},\pi_t^b)} \left[ f_h^{(t)}(s_h,a_h)^2 \right] + \lambda d \right]}$$

$$\le \sqrt{\frac{dC^*}{n}} \sum_{t=1}^{T} \sqrt{n\omega \mathbb{E}_{(s_{h-1},a_{h-1})\sim(P^{(*,t)},\pi_t)} \left[ \mathbb{E}_{(s_h,a_h)\sim(P^{(\star,t)},\pi_t^b)} \left[ f_h^{(t)}(s_h,a_h)^2 \right] \right] + \lambda d}$$

$$\le \sqrt{\frac{dC^*}{n}} \sum_{t=1}^{T} \left( \sqrt{n\omega \mathbb{E}_{(s_{h-1},a_{h-1})\sim(P^{(*,t)},\pi_t)} \left[ \mathbb{E}_{(s_h,a_h)\sim(P^{(\star,t)},\pi_t^b)} \left[ f_h^{(t)}(s_h,a_h)^2 \right] \right]} + \sqrt{\lambda d} \right)$$

$$\le \sqrt{\frac{dC^*}{n}} \sum_{t=1}^{T} \sqrt{n\omega^2 \mathbb{E}_{(s_h,a_h)\sim(P^{(\star,t)},\pi_t^b)} \left[ f_h^{(t)}(s_h,a_h)^2 \right]} + Td\sqrt{\frac{\lambda C^*}{n}}$$

$$= \omega\sqrt{dC^*} \sum_{t=1}^{T} \sqrt{\mathbb{E}_{(s_h,a_h)\sim(P^{(\star,t)},\pi_t^b)} \left[ f_h^{(t)}(s_h,a_h)^2 \right]} + Td\sqrt{\frac{\lambda C^*}{n}}$$

$$\le \omega\sqrt{dC^*} \sqrt{T \sum_{t=1}^{T} \mathbb{E}_{(s_h,a_h)\sim(P^{(\star,t)},\pi_t^b)} \left[ f_h^{(t)}(s_h,a_h)^2 \right]} + Td\sqrt{\frac{\lambda C^*}{n}}$$

$$\le \omega\sqrt{dC^*T\zeta_n} + Td\sqrt{\frac{\lambda C^*}{n}} \tag{D.10}$$

Further, note that,

$$\sum_{t=1}^{T} \mathbb{E}_{(s_1,a_1)\sim(P^{(*,t)},\pi_t)} \left[ f_1^{(t)}(s_1,a_1) \right] \le \sum_{t=1}^{T} \sqrt{\mathbb{E}_{(s_1,a_1)\sim(P^{(*,t)},\pi_t)} \left[ f_1^{(t)}(s_1,a_1)^2 \right]}$$

$$\le \sum_{t=1}^{T} \sqrt{\omega \mathbb{E}_{(s_1,a_1)\sim(P^{(*,t)},\pi_t^b)} \left[ f_1^{(t)}(s_1,a_1)^2 \right]}$$

$$= \sqrt{\omega} \sum_{t=1}^{T} \sqrt{\zeta_1^{(t)}}$$

$$\le \sqrt{\omega T \zeta_n},$$

where the last inequality follows from the Cauchy-Schwarz inequality and (D.3).

Finally, from (D.5) we get

$$
\sum_{t=1}^{T} \left[ V_{P^{(*,t)},r^t}^{\pi_t} - V_{P^{(*,t)},r^t}^{\widehat{\pi}_t} \right]
$$

$$
\leq \sum_{t=1}^{T} \sum_{h=1}^{H} \mathbb{E}_{(s_h,a_h)\sim(P^{(*,t)},\pi_t)} \left[ \widehat{b}_h^{(t)}(s_h,a_h) \right] + H \sum_{t=1}^{T} \sum_{h=1}^{H} \mathbb{E}_{(s_h,a_h)\sim(P^{(*,t)},\pi_t)} \left[ f_h^{(t)}(s_h,a_h) \right]
$$

$$
\qquad + H\sqrt{\omega T \zeta_n}
$$

$$
\leq \sum_{h=2}^{H} \sum_{t=1}^{T} \mathbb{E}_{(s_h,a_h)\sim(P^{(*,t)},\pi_t)} \left[ \widehat{b}_h^{(t)}(s_h,a_h) \right] + H \sum_{h=2}^{H} \sum_{t=1}^{T} \mathbb{E}_{(s_h,a_h)\sim(P^{(*,t)},\pi_t)} \left[ f_h^{(t)}(s_h,a_h) \right]
$$

$$
\quad + \sum_{t=1}^{T} \mathbb{E}_{(s_1,a_1)\sim(P^{(*,t)},\pi_t)} \left[ \widehat{b}_1^{(t)}(s_1,a_1) \right] + H \sum_{t=1}^{T} \mathbb{E}_{(s_1,a_1)\sim(P^{(*,t)},\pi_t)} \left[ f_1^{(t)}(s_1,a_1) \right] + H\sqrt{\omega T \zeta_n}
$$

$$
\leq H\omega\alpha T d \sqrt{\frac{C^*}{n}} + HTd\sqrt{\frac{\lambda C^*}{n}} + H^2\omega\sqrt{dC^*T\zeta_n} + H^2 T d\sqrt{\frac{\lambda C^*}{n}} + \alpha T\sqrt{\frac{d}{n}} + 2H\sqrt{\omega T \zeta_n}
$$

$$
\leq H\omega\alpha T d \sqrt{\frac{C^*}{n}} + 2H^2 T d\sqrt{\frac{\lambda C^*}{n}} + H^2\omega\sqrt{dC^*T\zeta_n} + \alpha T\sqrt{\frac{d}{n}} + 2H\sqrt{\omega T \zeta_n}. \qquad \text{(D.11)}
$$

So, we have

$$
\frac{1}{T}\sum_{t=1}^{T} \left[ V_{P^{(*,t)},r^t}^{\pi_t} - V_{P^{(*,t)},r^t}^{\widehat{\pi}_t} \right] \leq H\omega\alpha d\sqrt{\frac{C^*}{n}} + 2H^2 d\sqrt{\frac{\lambda C^*}{n}} + H^2\omega\sqrt{\frac{dC^*\zeta_n}{T}} + \alpha\sqrt{\frac{d}{n}} + 2H\sqrt{\frac{\omega\zeta_n}{T}}.
$$

$$
\text{(D.12)}
$$

$\square$

# E  Proof of Supporting Lemmas in Appendix D

In this section, we provide the proofs of the lemmas that we used in the proof of Theorem D.3.

## E.1  Proof of Lemma D.1

*Proof of Lemma D.1.* For $h = 1$,

$$
\mathbb{E}_{(s_1,a_1)\sim(\widehat{P}^{(t)},\pi_t)} \left[ f_1^{(t)}(s_1,a_1) \right] \leq \sqrt{\mathbb{E}_{(s_1,a_1)\sim(\widehat{P}^{(t)},\pi_t)} \left[ f_1^{(t)}(s_1,a_1)^2 \right]}
$$

$$
\leq \sqrt{\omega_t \mathbb{E}_{(s_1,a_1)\sim(P^{(*,t)},\pi_t^b)} \left[ f_1^{(t)}(s_1,a_1)^2 \right]}
$$

$$
= \sqrt{\omega_t \zeta_1^{(t)}}
$$

where the first inequality follows from Jensen's inequality and the second inequality follows from importance sampling. Denoting $\zeta_h^{(t)} = \mathbb{E}_{(s_h,a_h)\sim(P^{(*,t)},\pi_t^b)}\left[f_h^{(t)}(s,a)^2\right]$, for $h \geq 2$, we have

$$
\begin{aligned}
&\mathbb{E}_{\substack{s_h\sim(\widehat{P}^{(t)},\pi_t)\\a_h\sim\pi_t}}\left[f_h^{(t)}(s_h,a_h)\right]\\
&\overset{(i)}{\leq} \mathbb{E}_{(s_{h-1},a_{h-1})\sim(\widehat{P}^{(t)},\pi_t)}\left[\left\|\widehat{\phi}_{h-1}(s_{h-1},a_{h-1})\right\|_{(\Sigma_{h-1,\pi_t^b,\widehat{\phi}})^{-1}} \times\right.\\
&\qquad\qquad \left.\sqrt{n\omega_t\mathbb{E}_{(s_h,a_h)\sim(P^{(\star,t)},\pi_t^b)}[f_h^{(t)}(s_h,a_h)^2]+\lambda d+n\mathbb{E}_{(s_{h-1},a_{h-1})\sim(P^{(\star,t)},\pi_t^b)}\left[f_{h-1}^{(t)}(s_{h-1},a_{h-1})^2\right]}\right]\\
&\overset{(ii)}{=} \mathbb{E}_{(s_{h-1},a_{h-1})\sim(\widehat{P}^{(t)},\pi_t)}\left[\sqrt{n\omega_t\zeta_h^{(t)}+\lambda d+n\zeta_{h-1}^{(t)}}\left\|\widehat{\phi}_{h-1}(s_{h-1},a_{h-1})\right\|_{(\Sigma_{h-1,\pi_t^b,\widehat{\phi}})^{-1}}\right]\\
&= \mathbb{E}_{(s_{h-1},a_{h-1})\sim(\widehat{P}^{(t)},\pi_t)}\left[\alpha_h^{(t)}\left\|\widehat{\phi}_{h-1}(s_{h-1},a_{h-1})\right\|_{(\Sigma_{h-1,\pi_t^b,\widehat{\phi}})^{-1}}\right]
\end{aligned}
$$

where $(i)$ follows from Lemma G.1 and $|f_h^{(t)}(s_h,a_h)| \leq 1$, $(ii)$ uses notations defined in Appendix C. $\qquad\square$

## E.2 Proof of Lemma D.2

*Proof of Lemma D.2.* We condition on the events:

$$
\sum_{t=1}^{T}\mathbb{E}_{(s_h,a_h)\sim(P^{(*,t)},\pi_t^b)}\left[f_h^{(t)}(s,a)^2\right] \leq \zeta_n, \quad \text{where } \zeta_n := \frac{2\log(2|\Phi||\Psi|^T nH/\delta)}{n}, \qquad (\text{E.1})
$$

and

$$
\forall\phi\in\Phi:\|\phi_h(s,a)\|_{(\widehat{\Sigma}_{h,\phi}^{(t)})^{-1}} = \Theta\left(\|\phi_h(s,a)\|_{(\Sigma_{h,\pi_t^b,\phi})^{-1}}\right). \qquad (\text{E.2})
$$

From Lemma F.3 and Lemma G.2, this event happens with probability $1 - \delta$. Conditioning on this event, we have

$$
\alpha_h^{(t)} = \sqrt{n\omega_t\zeta_h^{(t)}+\lambda d+n\zeta_{h-1}^{(t)}} \leq \sqrt{2n\omega\zeta_n+\lambda d} = \alpha \qquad (\text{E.3})
$$

We have

$$\sum_{t=1}^{T} \left[ V_{\widehat{P}^{(t)}, r^t - \widehat{b}^{(t)}}^{\pi_t} - V_{P^{(*,t)}, r^t}^{\pi_t} \right]$$

$$\overset{(i)}{=} \sum_{t=1}^{T} \sum_{h=1}^{H} \mathbb{E}_{(s_h, a_h) \sim (\widehat{P}^{(t)}, \pi_t)} \left[ -\widehat{b}_h^{(t)}(s_h, a_h) + (P_h^{(*,t)} - \widehat{P}_h^{(t)}) V_{h+1, P^{(*,t)}, r^t}^{\pi_t}(s_h, a_h) \right]$$

$$\overset{(ii)}{\leq} H \sum_{t=1}^{T} \sum_{h=1}^{H} \mathbb{E}_{(s_h, a_h) \sim (\widehat{P}^{(t)}, \pi_t)} \left[ -\widehat{b}_h^{(t)}(s_h, a_h) + f_h^{(t)}(s_h, a_h) \right]$$

$$\overset{(iii)}{\leq} H \sum_{t=1}^{T} \sum_{h=2}^{H} \mathbb{E}_{(s_{h-1}, a_{h-1}) \sim (\widehat{P}^{(t)}, \pi_t)} \left[ \min \left\{ \alpha_h^{(t)} \left\| \widehat{\phi}_{h-1}(s_{h-1}, a_{h-1}) \right\|_{(\Sigma_{h-1, \pi_t^b, \widehat{\phi}})^{-1}}, 1 \right\} \right]$$

$$+ H \sum_{t=1}^{T} \sqrt{\omega \zeta_1^{(t)}} + H \sum_{t=1}^{T} \sum_{h=1}^{H} \mathbb{E}_{(s_h, a_h) \sim (\widehat{P}^{(t)}, \pi_t)} \left[ -\widehat{b}_h^{(t)}(s_h, a_h) \right]$$

$$\overset{(iv)}{\leq} H \sum_{t=1}^{T} \sum_{h=2}^{H} \mathbb{E}_{(s_{h-1}, a_{h-1}) \sim (\widehat{P}^{(t)}, \pi_t)} \left[ \min \left\{ \alpha_h^{(t)} \left\| \widehat{\phi}_{h-1}(s_{h-1}, a_{h-1}) \right\|_{(\Sigma_{h-1, \pi_t^b, \widehat{\phi}})^{-1}}, 1 \right\} \right]$$

$$+ H \sqrt{\omega T \zeta_n} + H \sum_{t=1}^{T} \sum_{h=1}^{H} \mathbb{E}_{(s_h, a_h) \sim (\widehat{P}^{(t)}, \pi_t)} \left[ -\widehat{b}_h^{(t)}(s_h, a_h) \right]$$

$$\overset{(v)}{\lesssim} H \sum_{t=1}^{T} \sum_{h=2}^{H} \mathbb{E}_{(s_{h-1}, a_{h-1}) \sim (\widehat{P}^{(t)}, \pi_t)} \left[ \min \left\{ \alpha \left\| \widehat{\phi}_{h-1}(s_{h-1}, a_{h-1}) \right\|_{(\Sigma_{h-1, \pi_t^b, \widehat{\phi}})^{-1}}, 1 \right\} \right]$$

$$+ H \sqrt{\omega T \zeta_n} + H \sum_{t=1}^{T} \sum_{h=1}^{H} \mathbb{E}_{(s_h, a_h) \sim (\widehat{P}^{(t)}, \pi_t)} \left[ -\min \left\{ \alpha \left\| \widehat{\phi}_h(s_h, a_h) \right\|_{(\Sigma_{h, \pi_t^b, \widehat{\phi}})^{-1}}, 1 \right\} \right]$$

$$\overset{(vi)}{\leq} H \sqrt{\omega T \zeta_n} + H \sum_{t=1}^{T} \mathbb{E}_{(s_1, a_1) \sim (\widehat{P}^{(t)}, \pi_t)} \left[ -\min \left\{ \alpha \left\| \widehat{\phi}_1(s_1, a_1) \right\|_{(\Sigma_{1, \pi_t^b, \widehat{\phi}})^{-1}}, 1 \right\} \right]$$

$$\leq H \sqrt{\omega T \zeta_n}.$$

where $(i)$ follows from Lemma O.1, $(ii)$ follows from the observation $V_{P^{(*,t)}, r^t}^{\pi_t} \leq H$, $(iii)$ follows from Lemma D.1, $(iv)$ follows from Cauchy-Schwarz inequality and the fact that $\sum_{t=1}^{T} \zeta_h^{(t)} \leq \zeta_n$, $(v)$ follows from (E.2). $\square$

# F  Multitask Offline MLE

Consider a sequential conditional probability estimation setting with an instance space $\mathcal{X}$ and target space $\mathcal{Y}$ and with a conditional probability density $p(y \mid x) = f^*(x, y)$. We consider a function class $\mathcal{F} : (\mathcal{X} \times \mathcal{Y}) \to \mathbb{R}$ for modeling the condition distribution $f^*$, and we further assume that the realizability condition holds i.e. $f^* \in \mathcal{F}$. We are given a dataset $D := \{(x_i, y_i)\}_{i=1}^{n}$, where $x_i \sim \mathcal{D}_i = \mathcal{D}_i(x_{1:i-1}, y_{1:i-1})$ and $y_i \sim p(\cdot \mid x_i)$. Let $D'$ denote a tangent sequence $\{(x_i', y_i')\}_{i=1}^{n}$ where $x_i' \sim \mathcal{D}_i(x_{1:i-1}, y_{1:i-1})$ and $y_i' \sim p(\cdot \mid x_i')$. Note that here $x_i'$ depends on the original sequence, and so the tangent sequence is independent conditional on $D$.

**Lemma F.1** (Lemma 24 of Agarwal et al. (2020b)). *Let $D$ be a dataset of $n$ examples, and let $D'$ be a tangent sequence. Let $L(f, D) = \sum_{i=1}^{n} \ell(f, (x_i, y_i))$ be any function that decomposes additively across examples where $\ell$ is any function, and let $\widehat{f}(D)$ be any estimator taking as input random variable $D$ and with range $\mathcal{F}$. Then*

$$\mathbb{E}_D \left[ \exp \left( L(\widehat{f}(D), D) - \log \mathbb{E}_{D'} \left[ \exp(L(\widehat{f}(D), D')) \right] - \log |\mathcal{F}| \right) \right] \leq 1.$$

**Lemma F.2** (Lemma 25 of Agarwal et al. (2020b)). *For any two conditional probability densities $f_1, f_2$ and any distribution $\mathcal{D} \in \Delta(\mathcal{X})$ we have*

$$\mathbb{E}_{x \sim \mathcal{D}} \| f_1(x, \cdot) - f_2(x, \cdot) \|_{TV}^2 \leq -2 \log \mathbb{E}_{x \sim \mathcal{D}, y \sim f_2(\cdot \mid x)} \exp \left( -\frac{1}{2} \log(f_2(x, y) / f_1(x, y)) \right).$$

**Lemma F.3** (Multitask offline MLE guarantee). *Given $\delta \in (0,1)$, consider the transition kernels learned in Algorithm 1. For any $n, h$ with probability at least $1 - \delta/2$, we have*

$$\sum_{t=1}^{T} \mathbb{E}_{(s_h, a_h) \sim (P^{(*,t)}, \pi_t^b)} \left[ f_h^{(t)}(s, a)^2 \right] \leq \zeta_n, \quad \text{where } \zeta_n := \frac{2 \log(2|\Phi| |\Psi|^T n H / \delta)}{n}. \tag{F.1}$$

*Proof of Lemma F.3.* Let $\widehat{f}(D)$ denote empirical maximum likelihood estimator:

$$\widehat{f}(D) := \operatorname*{argmax}_{f \in \mathcal{F}} \sum_{(x_i, y_i) \in D} \log f(x_i, y_i)$$

We combine Lemma F.1 with the Chernoff method to obtain the following exponential tail bound: with probability $1 - \delta$, we have

$$-\log \mathbb{E}_{D'} \left[ \exp(L(\widehat{f}(D), D')) \right] \leq -L(\widehat{f}(D), D) + \log |\mathcal{F}| + \log(1/\delta). \tag{F.2}$$

Now, we set $L(f, D) = \sum_{i=1}^{n} -\frac{1}{2} \log(f^*(x_i, y_i)/f(x_i, y_i))$ where $D = \{x_i, y_i\}_{i=1}^{n}$ is a dataset and $D' = \{x_i', y_i'\}_{i=1}^{n}$ is a tangent sequence. In the multitask offline RL setting, let $x = \{(s_h^t, a_h^t)\}_{t=1}^{T}$, $y = \{s_{h+1}^t\}_{t=1}^{T}$ and $f(x, y) = \prod_{t=1}^{T} P_h^t(s_{h+1}^t \mid s_h^t, a_h^t)$. Then, the dataset $D_h$ can be decomposed into $D_h = \cup_{t=1}^{T} D_h^{(t)}$ where $D_h^{(t)} = \{(s_h^{(i,t)}, a_h^{(i,t)}, s_{h+1}^{(i,t)})\}_{i \in [n]}$. Similarly, $D_h' = \cup_{t=1}^{T} (D_h')^{(t)}$, and $(\mathcal{D}_h^t)_i := (\mathcal{D}_h^t)_i((s_h^t, a_h^t)_{1:i-1}, (s_{h+1}^t)_{1:i-1})$. Thus, in the multitask offline RL setting, we have the cardinality $|\mathcal{F}| = |\Phi| |\Psi|^T$. With this choice, the right hand side of (F.2) is

$$\sum_{i=1}^{n} \frac{1}{2} \log(f^*(x_i, y_i)/\widehat{f}(x_i, y_i)) + \log |\mathcal{F}| + \log(1/\delta) \leq \log |\mathcal{F}| + \log(1/\delta) = \log(|\Phi| |\Psi|^T / \delta), \tag{F.3}$$

where the first inequality follows because $\widehat{f}$ is the empirical maximum likelihood estimator and the realizability assumption. The equality follows because $|\mathcal{F}| = |\Phi| |\Psi|^T$. On the other hand, the left hand side of (F.2) is

$$-\log \mathbb{E}_{D_h'} \left[ \exp \left( \sum_{i=1}^{n} -\frac{1}{2} \log \left( \frac{f^*(x_i', y_i')}{\widehat{f}(x_i', y_i')} \right) \right) \,\middle|\, D_h \right]$$

$$\overset{(i)}{=} -\log \mathbb{E}_{D_h'} \left[ \exp \left( \sum_{i=1}^{n} -\frac{1}{2} \log \left( \prod_{t=1}^{T} \frac{P_h^{(*,t)}(s_{h+1}^{(i,t)} \mid s_h^{(i,t)}, a_h^{(i,t)})}{\widehat{P}_h^{(t)}(s_{h+1}^{(i,t)} \mid s_h^{(i,t)}, a_h^{(i,t)})} \right) \right) \,\middle|\, D_h \right]$$

$$\overset{(ii)}{=} -\sum_{t=1}^{T} \log \mathbb{E}_{(D_h')^{(t)}} \left[ \exp \left( \sum_{i=1}^{n} -\frac{1}{2} \log \left( \frac{P_h^{(*,t)}(s_{h+1}^{(i,t)} \mid s_h^{(i,t)}, a_h^{(i,t)})}{\widehat{P}_h^{(t)}(s_{h+1}^{(i,t)} \mid s_h^{(i,t)}, a_h^{(i,t)})} \right) \right) \,\middle|\, D_h \right]$$

$$\overset{(iii)}{=} -\sum_{t=1}^{T} \sum_{i=1}^{n} \log \mathbb{E}_{(D_h^{(t)})_i} \left[ \exp \left( -\frac{1}{2} \log \left( \frac{P_h^{(*,t)}(s_{h+1}^{(i,t)} \mid s_h^{(i,t)}, a_h^{(i,t)})}{\widehat{P}_h^{(t)}(s_{h+1}^{(i,t)} \mid s_h^{(i,t)}, a_h^{(i,t)})} \right) \right) \right]$$

$$\overset{(iv)}{\geq} \sum_{t=1}^{T} \frac{1}{2} \sum_{i=1}^{n} \mathbb{E}_{(s_h, a_h) \sim (\mathcal{D}_h^t)_i} \left\| \widehat{P}^{(t)}(\cdot \mid s_h, a_h) - P^{(*,t)}(\cdot \mid s_h, a_h) \right\|_{TV}^2$$

$$\overset{(v)}{=} \frac{n}{2} \sum_{t=1}^{T} \mathbb{E}_{(s_h, a_h) \sim (P^{(*,t)}, \pi_t^b)} \left[ f_h^{(t)}(s, a)^2 \right], \tag{F.4}$$

where $(i)$ follows from the above definition of $f(x, y)$, $(ii)$ follows because the data of $T$ tasks are independent conditional on $D_h$, $(iii)$ follows because $\widehat{P}^{(t)}$ is independent of the dataset $(D_h')^{(t)}$ and from the definition of $D_h'$, $(iv)$ follows from Lemma F.2, and $(v)$ follows because in task $T$, the data is collected using behavior policy $\pi_t^b$.

Combining (F.2), (F.3), (F.4), we get

$$\frac{n}{2} \sum_{t=1}^{T} \mathbb{E}_{(s_h, a_h) \sim (P^{(*,t)}, \pi_t^b)} \left[ f_h^{(t)}(s, a)^2 \right] \leq \log(|\Phi| |\Psi|^T / \delta) \tag{F.5}$$

Using union bound, we obtain that for any $h \in [H]$ and $n$ with probability at least $1 - \delta/2$, it holds that

$$\sum_{t=1}^{T} \mathbb{E}_{(s_h, a_h) \sim (P^{(*,t)}, \pi_t^b)} \left[ f_h^{(t)}(s, a)^2 \right] \leq \frac{2 \log(2|\Phi||\Psi|^T n H/\delta)}{n}.$$

This completes the proof. $\qquad\qquad\qquad\qquad\qquad\qquad\qquad\qquad\qquad\qquad\qquad\qquad\qquad$ □

# G   One-Step Back Lemma and Concentration of Penalty Term

## G.1   One-step back lemma

The following one-step back lemma is a key technical lemma for our proof. One-step back lemma for offline setting was first introduced in Uehara et al. (2022) for infinite-horizon stationary MDP. Our lemma extends their result to finite-horizon non-stationary MDP for offline setting. For any function $g \in \mathcal{S} \times \mathcal{A} \to \mathbb{R}$, policy $\pi$ and transition kernel $P$, the lemma shows that we can relate the expected value $\mathbb{E}_{(s_h, a_h) \sim (P, \pi)}[g(s_h, a_h)]$ to the potential function $\mathbb{E}_{(s_{h-1}, a_{h-1}) \sim (P, \pi)} \|\phi_{h-1}(s_{h-1}, a_{h-1})\|_{(\Sigma_{h-1, \pi_t^b, \phi})^{-1}}$.

**Lemma G.1** (One-step back inequality for non-stationary finite-horizon MDP in offline setting). *For each task $t \in [T]$, let $P \in \{\widehat{P}^{(t)}, P^{(*,t)}\}$ with embedding $\phi \in \{\widehat{\phi}, \phi^*\}$ and $\mu$ be an MDP model, and $\Sigma_{h, \pi_t^b, \phi} = n \mathbb{E}_{(s_h, a_h) \sim (P^{(*,t)}, \pi_t^b)}[\phi \phi^\top] + \lambda I$ be the covariance matrix following the behavior policy $\pi_t^b$ under the true environment $P^{(*,t)}$. Denote the total variation distance between $P^{(*,t)}$ and $P$ at time step $h$ by $f^t(s_h, a_h)$. Take any $g \in \mathcal{S} \times \mathcal{A} \to \mathbb{R}$ such that $\|g\|_\infty \leq B$. Then, letting $\omega = \max_{s,a}(1/\pi_t^b(a \mid s))$ for all $h \geq 2$, and for any policy $\pi$, we have*

$$\mathbb{E}_{(s_h, a_h) \sim (P, \pi)}[g(s_h, a_h)] \leq \mathbb{E}_{(s_{h-1}, a_{h-1}) \sim (P, \pi)} \left[ \|\phi_{h-1}(s_{h-1}, a_{h-1})\|_{(\Sigma_{h-1, \pi_t^b, \phi})^{-1}} \times \right.$$
$$\left. \sqrt{n \omega \mathbb{E}_{(s_h, a_h) \sim (P^{\star,t}, \pi_t^b)}[g^2(s_h, a_h)] + \lambda d B^2 + n B^2 \mathbb{E}_{(s_{h-1}, a_{h-1}) \sim (P^{(*,t)}, \pi_t^b)}[f^t(s_{h-1}, a_{h-1})^2]} \right].$$

*Proof of Lemma G.1.* First, we have

$$\mathbb{E}_{(s_h, a_h) \sim (P, \pi)}[g(s_h, a_h)]$$
$$= \mathbb{E}_{(s_{h-1}, a_{h-1}) \sim (P, \pi)} \left[ \int_{s_h} \sum_{a_h} g(s_h, a_h) \pi(a_h \mid s_h) \langle \phi_{h-1}(s_{h-1}, a_{h-1}), \mu_{h-1}(s_h) \rangle ds_h \right]$$
$$\leq \mathbb{E}_{(s_{h-1}, a_{h-1}) \sim (P, \pi)} \left[ \|\phi_{h-1}(s_{h-1}, a_{h-1})\|_{(\Sigma_{h-1, \pi_t^b, \phi})^{-1}} \left\| \int \sum_{a_h} g(s_h, a_h) \pi(a_h \mid s_h) \mu_{h-1}(s_h) ds_h \right\|_{\Sigma_{h-1, \pi_t^b, \phi}} \right],$$

where the inequality follows from Cauchy-Schwarz inequality.

Then,

$$\left\| \int \sum_{a_h} g(s_h, a_h) \pi(a_h \mid s_h) \mu_{h-1}(s_h) ds_h \right\|^2_{\Sigma_{h-1, \pi_t^b, \phi}}$$

$$= \left\{ \int \sum_{a_h} g(s_h, a_h) \pi(a_h \mid s_h) \mu_{h-1}(s_h) ds_h \right\}^\top \left\{ n \mathbb{E}_{\substack{s_{h-1} \sim P^{(*,t)} \\ a_{h-1} \sim \pi_b}} [\phi \phi^\top] + \lambda I \right\} \left\{ \int \sum_{a_h} g(s_h, a_h) \pi(a_h \mid s_h) \mu_{h-1}(s_h) ds_h \right\}$$

$$\overset{(i)}{\leq} n \mathbb{E}_{(s_{h-1}, a_{h-1}) \sim (P^{(*,t)}, \pi_b)} \left[ \left( \int \sum_{a_h} g(s_h, a_h) \pi(a_h \mid s_h) \mu_{h-1}(s_h)^\top \phi(s_{h-1}, a_{h-1}) ds_h \right)^2 \right] + B^2 \lambda d$$

$$= n \mathbb{E}_{(s_{h-1}, a_{h-1}) \sim (P^{(*,t)}, \pi_b)} \left[ \mathbb{E}_{\substack{s_h \sim P(\cdot \mid s_{h-1}, a_{h-1}) \\ a_h \sim \pi}} [g(s_h, a_h)^2] \right] + B^2 \lambda d$$

$$\overset{(ii)}{\leq} n \mathbb{E}_{(s_{h-1}, a_{h-1}) \sim (P^{(*,t)}, \pi_b)} \left[ \mathbb{E}_{\substack{s_h \sim P^{(*,t)} \\ a_h \sim \pi}} [g(s_h, a_h)^2] \right] + B^2 \lambda d + n B^2 \mathbb{E}_{(s_{h-1}, a_{h-1}) \sim (P^{(*,t)}, \pi_b)} [f^t(s_{h-1}, a_{h-1})^2]$$

$$\overset{(iii)}{\leq} n \omega \mathbb{E}_{(s_h, a_h) \sim (P^{(*,t)}, \pi_b)} [g(s_h, a_h)^2] + B^2 \lambda d + n B^2 \mathbb{E}_{(s_{h-1}, a_{h-1}) \sim (P^{(*,t)}, \pi_b)} [f^t(s_{h-1}, a_{h-1})^2]$$

where $(i)$ follows from the assumption $\|g\|_\infty \leq B$ and for any function $h : \mathcal{S} \to [0, 1]$, $\|\int \mu_h(s) h(s) ds\|_2 \leq \sqrt{d}$, $(ii)$ follows from the definition of $f^t(s_h, a_h)$ which is the total variation distance between $P^*$ and $P$ at time step $h$, and finally $(iii)$ follows from importance sampling. This completes the proof. $\qquad \square$

## G.2 Concentration of penalty term

Recall that $\Sigma_{h, \pi_t^b, \phi} = n \mathbb{E}_{(s_h, a_h) \sim (P^{(*,t)}, \pi_t^b)} [\phi \phi^\top] + \lambda I$. Thus, $\widehat{\Sigma}_h^{(t)}$ is equal to $\Sigma_{h, \pi_t^b, \widehat{\phi}}$ in expectation. We now provide an important lemma to ensure the concentration of the penalty term. The version for fixed $\phi$ is proved in Zanette et al. (2021a). Here, we take a union bound over the whole feature $\phi \in \Phi$, number of total tasks $T$, horizon $H$ and cardinality $n$ of each offline dataset from individual tasks.

**Lemma G.2** (Concentration of the penalty term). *Fix $\delta \in (0, 1)$ and set $\lambda = O(d \log(2nTH|\Phi|/\delta))$ for any $n$. With probability at least $1 - \delta/2$, we have that for any $n \in \mathbb{N}$, $h \in [H]$, $t \in [T]$ and $\phi \in \Phi$,*

$$\beta_1 \|\phi_h(s, a)\|_{(\Sigma_{h, \pi_t^b, \phi})^{-1}} \leq \|\phi_h(s, a)\|_{(\widehat{\Sigma}_h^{(t)})^{-1}} \leq \beta_2 \|\phi_h(s, a)\|_{(\Sigma_{h, \pi_t^b, \phi})^{-1}},$$

*where $\beta_1$ and $\beta_2$ are some absolute constants.*

## H Proof of Lemma 4.3: Approximate Feature for New Task

We first restate Lemma 4.3.

**Lemma H.1.** *Under Assumption 4.1, the output $\widehat{\phi}$ of Algorithm 1 is a $\xi_{down}$-approximate feature for MDP $\mathcal{M}^{T+1}$ where $\xi_{down} = \xi + \frac{C_L C_R \nu}{\kappa} \sqrt{\frac{2T \log(2|\Phi||\Psi|^T nH/\delta)}{n}}$, i.e. there exist a time-dependent unknown (signed) measure $\widehat{\mu}^*$ over $\mathcal{S}$ such that for any $(s, a) \in \mathcal{S} \times \mathcal{A}$, we have*

$$\|P_h^{(*, T+1)}(\cdot|s, a) - \langle \widehat{\phi}_h(s, a), \widehat{\mu}_h^*(\cdot) \rangle\|_{TV} \leq \xi_{down}.$$

*Furthermore, for any $g : \mathcal{S} \to [0, 1]$, we have $\|\int \widehat{\mu}_h^*(s) g(s) ds\|_2 \leq C_L \sqrt{d}$.*

The following proof is motivated from the proof of Lemma 1 in Cheng et al. (2022).

*Proof of Lemma H.1.* For all $(s,a) \in \mathcal{S} \times \mathcal{A}$, $h \in [H]$ and for any $t \in [T]$ we have

$$\sum_{t=1}^{T} \|\widehat{P}_h^{(t)}(\cdot|s,a) - P_h^{(*,t)}(\cdot|s,a)\|_{\mathrm{TV}}$$

$$\leq \sum_{t=1}^{T} \max_{s \in \mathcal{S}, a \in \mathcal{A}} \|\widehat{P}_h^{(t)}(\cdot|s,a) - P_h^{(*,t)}(\cdot|s,a)\|_{\mathrm{TV}}$$

$$\overset{(i)}{\leq} \sum_{t=1}^{T} C_R \mathbb{E}_{(s_h,a_h) \sim \mathcal{U}(\mathcal{S},\mathcal{A})} \|\widehat{P}_h^{(t)}(\cdot|s_h,a_h) - P_h^{(*,t)}(\cdot|s_h,a_h)\|_{\mathrm{TV}}$$

$$\overset{(ii)}{\leq} \frac{C_R \nu}{\kappa} \sum_{t=1}^{T} \mathbb{E}_{(s_h,a_h) \sim (P^{(*,t)}, \pi_t^b)} \|\widehat{P}_h^{(t)}(\cdot|s_h,a_h) - P_h^{(*,t)}(\cdot|s_h,a_h)\|_{\mathrm{TV}}$$

$$\overset{(iii)}{\leq} \frac{C_R \nu}{\kappa} \sqrt{\frac{2T \log(2|\Phi||\Psi|^T n H/\delta)}{n}} \tag{H.1}$$

where $(i)$, $(ii)$ follows from Assumption 4.1 and $(iii)$ follows from Theorem 3.3.

Defining $\widehat{\mu}^*(\cdot) = \sum_{t=1}^{T} c_t \widehat{\mu}^{(t)}(\cdot)$, we have

$$\|P_h^{(*,T+1)}(\cdot|s,a) - \langle \widehat{\phi}_h(s,a), \widehat{\mu}_h^*(\cdot) \rangle\|_{\mathrm{TV}}$$

$$= \|P_h^{(*,T+1)}(\cdot|s,a) - \langle \widehat{\phi}_h(s,a), \sum_{t=1}^{T} c_t \widehat{\mu}^{(t)}(\cdot) \rangle\|_{\mathrm{TV}}$$

$$= \|P_h^{(*,T+1)}(\cdot|s,a) - \sum_{t-1}^{T} c_t \widehat{P}_h^{(t)}(\cdot|s,a)\|_{\mathrm{TV}}$$

$$\leq \|P_h^{(*,T+1)}(\cdot|s,a) - \sum_{t-1}^{T} c_t P_h^{(*,t)}(\cdot|s,a)\|_{\mathrm{TV}} + \sum_{t=1}^{T} c_t \|P_h^{(*,t)}(\cdot|s,a) - \widehat{P}_h^{(t)}(\cdot|s,a)\|_{\mathrm{TV}}$$

$$\overset{(i)}{\leq} \xi + \frac{C_L C_R \nu}{\kappa} \sqrt{\frac{2T \log(2|\Phi||\Psi|^T n H/\delta)}{n}},$$

where $(i)$ follows from Assumption 4.1, (H.1) and the fact that $c_t \in [0, C_L]$ for all $t \in [T]$. Moreover, by normalization, for any $g : \mathcal{S} \to [0,1]$, we get

$$\left\| \int \widehat{\mu}_h^*(s) g(s) ds \right\|_2 \leq \sum_{t=1}^{T} \left\| \int \widehat{\mu}_h^{(t)}(s) g(s) ds \right\|_2$$

$$\leq C_L \sqrt{d},$$

where the last inequality follows from Assumption 3.1. $\qquad\qquad\square$

# I  Proof for Downstream Reward-Free RL

For any $h \in [H]$, we define

$$P_h^{(*,T+1)}(\cdot|s,a) = \langle \phi_h^*(s,a), \mu_h^{(*,T+1)}(\cdot) \rangle,$$

$$\overline{P}_h(\cdot|s,a) = \langle \widehat{\phi}_h(s,a), \widehat{\mu}_h^*(\cdot) \rangle.$$

Given a reward function $r$ (as is provided in the planning phase of reward-free RL setting), for any function $f : \mathcal{S} \to \mathbb{R}$ and $h \in [H]$, we define the transition operators as follows

$$(P_h^{(*,T+1)} f)(s,a,r) = \int_{s'} \langle \phi_h^*(s,a), \mu_h^{(*,T+1)}(s') \rangle f(s') ds',$$

$$(\overline{P}_h f)(s,a,r) = \int_{s'} \langle \widehat{\phi}_h(s,a), \widehat{\mu}_h^*(s') \rangle f(s') ds'.$$

When no reward function is provided as is the case in the exploration phase of reward-free RL setting, we simply omit $r$ from the above operator notation.

In this section for notational simplicity we denote $V_{h,P^{(*,T+1)},r}^{\pi}(s)$ and $Q_{h,P^{(*,T+1)},r}^{\pi}(s,a)$ by $V_h^{\pi}(s,r)$ and $Q_h^{\pi}(s,a,r)$ respectively where $r$ is reward function provided in the planning phase of downstream reward-free RL task. We similarly denote the optimal value function and action-value function under reward function $r$ as $V_h^*(s,r)$ and $Q_h^*(s,a,r)$ respectively.

We also introduce the truncated optimal value function $\widetilde{V}_h^*(s,r)$ in the planning phase, which is recursively defined from step $H+1$ to step 1. Compared to the definition of standard optimal value function $V_h^*(s,r)$, the main difference is that we take minimization over the value function and $H$ in each step in this definition. We provide the formal definition as follows.

**Definition I.1** (Truncated Optimal Value Function). We introduce the truncated optimal value function $\widetilde{V}_h^*(s,r)$ which is recursively defined from step $H+1$ to step 1:

$$
\widetilde{V}_{H+1}^*(s,r) = 0, \ \forall s \in \mathcal{S}
$$
$$
\widetilde{Q}_h^*(s,a,r) = r_h(s,a) + P_h^{(*,T+1)}\widetilde{V}_{h+1}^*(s,a,r), \ \forall(s,a) \in \mathcal{S} \times \mathcal{A}
$$
$$
\widetilde{V}_h^*(s,r) = \min\left\{ \max_{a \in \mathcal{A}} \left\{ r_h(s,a) + P_h^{(*,T+1)}\widetilde{V}_{h+1}^*(s,a,r) \right\}, H \right\}, \ \forall s \in \mathcal{S}, h \in [H].
$$

We can similarly define $\widetilde{V}_h^{\pi}(s,r)$ and $\widetilde{Q}_h^{\pi}(s,a,r)$.

## I.1 Supporting Lemmas

Now we state the supporting lemmas that are used in the proof of Theorem 4.4. The proof of these lemmas are provided in Appendix J.

The following lemma shows that the linear weight $\widehat{w}_h^k$ in Algorithm 2 is bounded.

**Lemma I.2** (Bounds on Weights in Algorithm 2). *For any $h \in [H]$, the weight $\widehat{w}_h^k$ in Algorithm 2 satisfies*

$$
\left\| \widehat{w}_h^k \right\|_2 \leq H\sqrt{dK}.
$$

**Lemma I.3.** *Let $\mathcal{E}$ be the event that for all $(k,h) \in [K_{RFE}] \times [H]$,*

$$
\left\| \sum_{\tau=1}^{k-1} \widehat{\phi}_h^{\tau} \left( \widehat{V}_{h+1}^k(s_{h+1}^{\tau}) - P_h^{(*,T+1)}\widehat{V}_{h+1}^k(s_h^{\tau}, a_h^{\tau}) \right) \right\|_{(\Lambda_h^k)^{-1}} \lesssim dH\sqrt{\log\left(\frac{dK_{RFE}H\max(\xi_{down},1)}{\delta}\right)}.
$$

*Then $Pr[\mathcal{E}] \geq 1 - \delta/8$.*

**Lemma I.4.** *With probability $1 - \delta/8$, we have for all $(s,a) \in \mathcal{S} \times \mathcal{A}$,*

$$
\left| \widehat{\phi}_h(s,a)^{\top}\widehat{w}_h^k - P_h^{(*,T+1)}\widehat{V}_{h+1}^k(s,a) \right| \lesssim \beta\|\widehat{\phi}_h(s,a)\|_{(\Lambda_h^k)^{-1}} + H\xi_{down}
$$

**Lemma I.5.** *With probability $1 - \delta/4$, for all $(h,k) \in [H] \times [K_{RFE}]$, and any $s \in \mathcal{S}$, we have*

$$
\widetilde{V}_h^*(s,r^k) \lesssim \widehat{V}_h^k(s) + H(H-h+1)\xi_{down}
$$

*and*

$$
\sum_{k=1}^{K_{RFE}} \widehat{V}_1^k(s_1^k) \leq c\sqrt{d^3H^4K_{RFE}\log(dK_{RFE}H/\delta)} + H^2K_{RFE}\xi_{down},
$$

*where $c > 0$ is a constant.*

**Lemma I.6.** *With probability $1 - \delta/2$, for the function $u_h(\cdot,\cdot)$ defined in Line 5 of Algorithm 3, we have*

$$
\mathbb{E}_{s \sim \mu}\left[ \widetilde{V}_1^*(s,u) \right] \leq c'\sqrt{d^3H^4\log(dK_{RFE}H/\delta)/K_{RFE}} + 2H^2\xi_{down}.
$$

**Lemma I.7.** *With probability $1 - \delta/2$, for any reward function which is linear with respect to the unknown feature $\phi^* : \mathcal{S} \times \mathcal{A} \to \mathbb{R}^d$, for all $(s,a) \in \mathcal{S} \times \mathcal{A}$ and $h \in [H]$, we have*

$$
Q_h^*(s,a,r) - H(H-H+1)\xi_{down} \leq \widehat{Q}_h(s,a) \leq r_h(s,a) + P_h^{(*,T+1)}\widehat{V}_{h+1}(s,a) + 2u_h(s,a) + H\xi_{down}
$$

## I.2 Proof of Theorem 4.4

We first restate Theorem 4.4

**Theorem I.8.** *Under Assumption 4.1, after collecting $K_{RFE}$ trajectories during the exploration phase in Algorithm 2, with probability at least $1 - \delta$, the output of Algorithm 3, policy $\pi$ satisfies*

$$\mathbb{E}_{s_1 \sim \mu}[V_1^*(s_1, r) - V_1^\pi(s_1, r)] \leq c' \sqrt{d^3 H^4 \log(dK_{RFE}H/\delta)/K_{RFE}} + 6H^2 \xi_{down}. \qquad (\text{I.1})$$

*If the linear combination misspecification error $\xi$ in Assumption 4.1 satisfies $\widetilde{O}(\sqrt{d^3/K_{RFE}})$ and the number of trajectories in the offline dataset for each upstream task is at least $\widetilde{O}(TK_{RFE}/d^3)$, then, provided $K_{RFE}$ is at least $O(d^3 H^4 \log(dH\delta^{-1}\epsilon^{-1})/\epsilon^2)$, with probability $1 - \delta$, the policy $\pi$ will be an $\epsilon$-optimal policy for any given reward during the planning phase.*

*Proof of Theorem I.8.* We condition on the events defined in Lemma I.6 and Lemma I.7 which, by union bound, hold with probability at least $1 - \delta$. By Lemma I.7, for any $s \in \mathcal{S}$, we have

$$\widehat{V}_1(s) = \max_{a \in \mathcal{A}} \widehat{Q}_1(s, a) \geq \max_{a \in \mathcal{A}} Q_1^*(s, a, r) - H^2 \xi_{down}$$
$$= V_1^*(s, r) - H^2 \xi_{down}.$$

This implies

$$\mathbb{E}_{s_1 \sim \mu}[V_1^*(s_1, r) - V_1^\pi(s_1, r)] \leq \mathbb{E}_{s_1 \sim \mu}[\widehat{V}_1(s_1) - V_1^\pi(s_1, r)] + H^2 \xi_{down},$$

where $\pi$ is the policy returned by Algorithm 3.

Observe that, using Lemma I.7, we have

$$\mathbb{E}_{s_1 \sim \mu}[\widehat{V}_1(s_1) - V_1^\pi(s_1, r)]$$
$$= \mathbb{E}_{s_1 \sim \mu}[\widehat{Q}_1(s_1, \pi_1(s_1)) - Q_1^\pi(s_1, \pi_1(s_1), r)]$$
$$\leq \mathbb{E}_{s_1 \sim \mu}[r_1(s_1, \pi_1(s_1)) + P_1^{(*,T+1)}\widehat{V}_2(s_1, \pi_1(s_1), r) + 2u_1(s_1, \pi_1(s_1)) + H\xi_{down} - r_1(s_1, \pi_1(s_1))$$
$$\qquad - P_1^{(*,T+1)}V_2^\pi(s_1, \pi_1(s_1), r)]$$
$$= \mathbb{E}_{s_1 \sim \mu, s_2 \sim P_1^{(*,T+1)}(\cdot|s_1, \pi_1(s_1))}[\widehat{V}_2(s_2) - V_2^\pi(s_2, r) + 2u_1(s_1, \pi_1(s_1))] + H\xi_{down}$$
$$\leq \cdots$$
$$\leq 2\mathbb{E}_{s \sim \mu}[V_1^\pi(s, u)] + H^2 \xi_{down}.$$

Moreover, note that $0 \leq \widehat{V}_h(s) \leq H$ and $0 \leq V_h^\pi(s, r) \leq H$ as $0 \leq r(s, a) \leq 1$. Thus, we would always have $\widehat{V}_h(s) - V_h^\pi(s, r) \leq H$. Along with the previous derivation, this implies,

$$\mathbb{E}_{s_1 \sim \mu}[\widehat{V}_1(s_1) - V_1^\pi(s_1, r)] \leq 2\mathbb{E}_{s \sim \mu}[\widetilde{V}_1^\pi(s, u)] + H^2 \xi_{down}.$$

By definition of $\widetilde{V}_1^*(s, u)$, we further have $\mathbb{E}_{s \sim \mu}[\widetilde{V}_1^\pi(s, u)] \leq \mathbb{E}_{s \sim \mu}[\widetilde{V}_1^*(s, u)]$. By Lemma I.6, we have

$$\mathbb{E}_{s \sim \mu}\left[\widetilde{V}_1^*(s, u)\right] \leq c' \sqrt{d^3 H^4 \log(dK_{RFE}H/\delta)/K_{RFE}} + 2H^2 \xi_{down}.$$

So, we have,

$$\mathbb{E}_{s_1 \sim \mu}[V_1^*(s_1, r) - V_1^\pi(s_1, r)] \leq \mathbb{E}_{s_1 \sim \mu}[\widehat{V}_1(s_1) - V_1^\pi(s_1, r)] + H^2 \xi_{down}$$
$$\leq 2\mathbb{E}_{s \sim \mu}[\widetilde{V}_1^\pi(s, u)] + 2H^2 \xi_{down}$$
$$\leq 2\mathbb{E}_{s \sim \mu}[\widetilde{V}_1^*(s, u)] + 2H^2 \xi_{down}$$
$$\leq 2c' \sqrt{d^3 H^4 \log(dK_{RFE}H/\delta)/K_{RFE}} + 6H^2 \xi_{down}. \qquad (\text{I.2})$$

Recall the definition of $\xi_{\text{down}}$ from Lemma 4.3, that is, $\xi_{\text{down}} = \xi + \frac{C_L C_R \nu}{\kappa} \sqrt{\frac{2T \log(2|\Phi||\Psi|^T n H / \delta)}{n}}$. If the linear combination misspecification error $\xi$ in Assumption 4.1 satisfies $\widetilde{O}(\sqrt{d^3 / K_{\text{RFE}}})$ and the number of trajectories in the offline dataset for each task in the upstream stage is at least $\widetilde{O}(TK_{\text{RFE}}/d^3)$, then the first term in Equation (I.2) dominates the second term $6H^2 \xi_{\text{down}}$. Then, by taking $K_{\text{RFE}} = c_K d^3 H^4 \log(dH\delta^{-1}\epsilon^{-1})/\epsilon^2$ for a sufficiently large constant $c_K > 0$, we have

$$\mathbb{E}_{s_1 \sim \mu}[V_1^*(s_1, r) - V_1^\pi(s_1, r)] \leq 2c' \sqrt{d^3 H^4 \log(dK_{\text{RFE}} H / \delta) / K_{\text{RFE}}} + 6H^2 \xi_{\text{down}} \leq \epsilon.$$

This completes the proof. $\qquad\square$

# J  Proof of Supporting Lemmas in Appendix I

## J.1  Proof of Lemma I.2

*Proof of Lemma I.2.* We have

$$\begin{aligned}
\|\widehat{w}_h^k\| &= \left\| (\Lambda_h^k)^{-1} \sum_{\tau=1}^{k-1} \widehat{\phi}_h(s_h^\tau, a_h^\tau) \widehat{V}_{h+1}^k(s_{h+1}^\tau) \right\| \\
&\leq \sqrt{k} \left( \sum_{\tau=1}^{k-1} \left\| \widehat{V}_{h+1}^k(s_{h+1}^\tau) \widehat{\phi}_h(s_h^\tau, a_h^\tau) \right\|_{(\Lambda_h^k)^{-1}}^2 \right)^{1/2} \\
&\leq \sqrt{K_{\text{RFE}}} \cdot H \cdot \left( \sum_{\tau=1}^{k-1} \left\| \widehat{\phi}_h(s_h^\tau, a_h^\tau) \right\|_{(\Lambda_h^k)^{-1}}^2 \right)^{1/2} \\
&\leq H \sqrt{d K_{\text{RFE}}}
\end{aligned}$$

where the first inequality follows from Lemma O.5 and the fact that the largest eigenvalue of $(\Lambda_h^k)^{-1}$ is at most 1, second inequality follows from the fact that $|\widehat{V}_{h+1}^k(s)| \leq H$ for all $s \in \mathcal{S}$ and the last inequality follows from Lemma O.4. $\qquad\square$

## J.2  Proof of Lemma I.3

*Proof of Lemma I.3.* The proof is similar to that of Lemma B.3 in (Jin et al., 2020b) with the major difference being the usage of approximate feature map $\widehat{\phi}(\cdot, \cdot)$ and different reward function at different episodes. We provide the full outline of the proof for completeness.

For all $(k, h) \in [K_{\text{RFE}}] \times [H]$, by Lemma I.2, we have $\|\widehat{w}_h^k\|_2 \leq H\sqrt{dK}$. Moreover, we have $r_h^k(\cdot, \cdot) = u_h^k(\cdot, \cdot)$ and hence we have

$$r_h^k(\cdot, \cdot) + u_h^k(\cdot, \cdot) = 2\beta \sqrt{\widehat{\phi}(\cdot, \cdot)^\top (\Lambda_h^k)^{-1} \widehat{\phi}(\cdot, \cdot)}.$$

Thus, our value function $\widehat{V}_{h+1}^k$ is of the form

$$V(\cdot) := \min \left\{ \max_{a \in \mathcal{A}} \widehat{\phi}(\cdot, a)^\top w + \beta \sqrt{\widehat{\phi}(\cdot, a)^\top (\Lambda)^{-1} \widehat{\phi}(\cdot, a)}, H \right\},$$

for some $\Lambda \in \mathbb{R}^{d \times d}$, and $w \in \mathbb{R}^d$ which matches the value function class defined in Lemma O.9. Moreover, by construction, the minimum eigenvalue of $\Lambda_h^k$ is lower bounded by 1. Combining Lemma O.7 and Lemma O.9, we have for any fixed $\varepsilon > 0$ that

$$\left\| \sum_{\tau=1}^{k-1} \widehat{\phi}_h^\tau \left( \widehat{V}_{h+1}^k(s_{h+1}^\tau) - P_h^{(*, T+1)} \widehat{V}_{h+1}^k(s_h^\tau, a_h^\tau) \right) \right\|_{(\Lambda_h^k)^{-1}}^2$$

$$\leq 4H^2 \left[ \frac{d}{2} \log(k+1) + d \log\left(1 + \frac{8H\sqrt{dk}}{\varepsilon}\right) + d^2 \log\left(1 + \frac{32\sqrt{d}\beta^2}{\varepsilon^2}\right) + \log\left(\frac{8}{\delta}\right) \right] + 8k^2 \varepsilon^2$$

We set the hyperparameter $\beta = C_L H \sqrt{d} + dH \sqrt{\log(dK_{\text{RFE}} H \max(\xi_{\text{down}}, 1)/\delta)} + H\xi_{\text{down}} \sqrt{dK_{\text{RFE}}}$. Finally, picking $\varepsilon = dH/k$, we have

$$\left\| \sum_{\tau=1}^{k-1} \widehat{\phi}_h^\tau \left( \widehat{V}_{h+1}^k(s_{h+1}^\tau) - P_h^{(*,T+1)} \widehat{V}_{h+1}^k(s_h^\tau, a_h^\tau) \right) \right\|_{(\Lambda_h^k)^{-1}} \lesssim dH \sqrt{\log\left( \frac{dK_{\text{RFE}} H \max(\xi_{\text{down}}, 1)}{\delta} \right)},$$

which concludes the proof. $\qquad\square$

### J.3  Proof of Lemma I.4

*Proof of Lemma I.4.* We condition on the event $\mathcal{E}$ defined in Lemma I.3, which holds with probability at least $1 - \delta/8$. We define $\widetilde{w}_h^k = \int_{s'} \widehat{V}_{h+1}^k(s') \widehat{\mu}^*(s') ds'$ where $\widehat{\mu}^*$ is defined in the proof of Lemma 4.3. Note that by Lemma 4.3, we have $\|\widetilde{w}_h^k\| \leq C_L H \sqrt{d}$.

Now,

$$\begin{aligned}
&\left| \widehat{\phi}_h(s,a)^\top \widehat{w}_h^k - P_h^{(*,T+1)} \widehat{V}_{h+1}^k(s,a) \right| \\
&= \left| \widehat{\phi}_h(s,a)^\top \widehat{w}_h^k - \widehat{\phi}_h(s,a)^\top \widetilde{w}_h^k + \widehat{\phi}_h(s,a)^\top \widetilde{w}_h^k - P_h^{(*,T+1)} \widehat{V}_{h+1}^k(s,a) \right| \\
&\leq \left| \widehat{\phi}_h(s,a)^\top \widehat{w}_h^k - \widehat{\phi}_h(s,a)^\top \widetilde{w}_h^k \right| + \left| \overline{P}_h \widehat{V}_{h+1}^k(s,a) - P_h^{(*,T+1)} \widehat{V}_{h+1}^k(s,a) \right| \\
&\leq \left| \widehat{\phi}_h(s,a)^\top \widehat{w}_h^k - \widehat{\phi}_h(s,a)^\top \widetilde{w}_h^k \right| + H\xi_{\text{down}}, \qquad\qquad\qquad\qquad\qquad (J.1)
\end{aligned}$$

where the last inequality follows from Lemma 4.3 and $|\widehat{V}_{h+1}^k| \leq H$.

The first term in (J.1) can be written as,

$$\begin{aligned}
&\widehat{\phi}_h(s,a)^\top \widehat{w}_h^k - \widehat{\phi}_h(s,a)^\top \widetilde{w}_h^k \\
&= \widehat{\phi}_h(s,a)^\top (\Lambda_h^k)^{-1} \sum_{\tau=1}^{k-1} \widehat{\phi}_h(s_h^\tau, a_h^\tau) \widehat{V}_{h+1}^k(s_{h+1}^\tau) - \widehat{\phi}_h(s,a)^\top (\Lambda_h^k)^{-1}(\Lambda_h^k) \widetilde{w}_h^k \\
&= \widehat{\phi}_h(s,a)^\top (\Lambda_h^k)^{-1} \left\{ \sum_{\tau=1}^{k-1} \widehat{\phi}_h(s_h^\tau, a_h^\tau) \widehat{V}_{h+1}^k(s_{h+1}^\tau) - \widetilde{w}_h^k - \sum_{\tau=1}^{k-1} \widehat{\phi}_h(s_h^\tau, a_h^\tau) \overline{P}_h \widehat{V}_{h+1}^k \right\} \\
&= \underbrace{-\widehat{\phi}_h(s,a)^\top (\Lambda_h^k)^{-1} \widetilde{w}_h^k}_{(a)} + \underbrace{\widehat{\phi}_h(s,a)^\top (\Lambda_h^k)^{-1} \left\{ \sum_{\tau=1}^{k-1} \widehat{\phi}_h(s_h^\tau, a_h^\tau) \left[ \widehat{V}_{h+1}^k(s_{h+1}^\tau) - P_h^{(*,T+1)} \widehat{V}_{h+1}^k(s_{h+1}^\tau) \right] \right\}}_{(b)} \\
&\quad + \underbrace{\widehat{\phi}_h(s,a)^\top (\Lambda_h^k)^{-1} \left\{ \sum_{\tau=1}^{k-1} \widehat{\phi}_h(s_h^\tau, a_h^\tau) \left( P_h^{(*,T+1)} - \overline{P}_h \right) \widehat{V}_{h+1}^k(s_{h+1}^\tau) \right\}}_{(c)} \qquad (J.2)
\end{aligned}$$

We now bound $(a), (b), (c)$ in (J.2) individually.

**Term (a).** We have,

$$\begin{aligned}
\left| -\widehat{\phi}_h(s,a)^\top (\Lambda_h^k)^{-1} \widetilde{w}_h^k \right| &\leq \left\| \widehat{\phi}_h(s,a) \right\|_{(\Lambda_h^k)^{-1}} \|\widetilde{w}_h^k\|_{(\Lambda_h^k)^{-1}} \\
&\leq \|\widetilde{w}_h^k\|_2 \left\| \widehat{\phi}_h(s,a) \right\|_{(\Lambda_h^k)^{-1}} \\
&\leq C_L H \sqrt{d} \left\| \widehat{\phi}_h(s,a) \right\|_{(\Lambda_h^k)^{-1}}. \qquad\qquad (J.3)
\end{aligned}$$

**Term (b).** Using Cauchy-Schwarz inequality and the definition of the event $\mathcal{E}$ from Lemma I.3, we have

$$\widehat{\phi}_h(s,a)^\top (\Lambda_h^k)^{-1} \bigg\{ \sum_{\tau=1}^{k-1} \widehat{\phi}_h(s_h^\tau, a_h^\tau) \Big[ \widehat{V}_{h+1}^k(s_{h+1}^\tau) - P_h^{(*,T+1)} \widehat{V}_{h+1}^k(s_{h+1}^\tau) \Big] \bigg\}$$

$$\leq \big\| \widehat{\phi}_h(s,a) \big\|_{(\Lambda_h^k)^{-1}} \bigg\| \sum_{\tau=1}^{k-1} \widehat{\phi}_h(s_h^\tau, a_h^\tau) \Big[ \widehat{V}_{h+1}^k(s_{h+1}^\tau) - P_h^{(*,T+1)} \widehat{V}_{h+1}^k(s_{h+1}^\tau) \Big] \bigg\|_{(\Lambda_h^k)^{-1}}$$

$$\lesssim dH \sqrt{\log \left( \frac{dK_{\mathrm{RFE}} H \max(\xi_{\mathrm{down}}, 1)}{\delta} \right)} \big\| \widehat{\phi}_h(s,a) \big\|_{(\Lambda_h^k)^{-1}}. \tag{J.4}$$

**Term (c).** We have

$$\widehat{\phi}_h(s,a)^\top (\Lambda_h^k)^{-1} \bigg\{ \sum_{\tau=1}^{k-1} \widehat{\phi}_h(s_h^\tau, a_h^\tau) \Big( P_h^{(*,T+1)} - \overline{P}_h \Big) \widehat{V}_{h+1}^k(s_{h+1}^\tau) \bigg\}$$

$$\overset{(i)}{\leq} \bigg| \widehat{\phi}_h(s,a)^\top (\Lambda_h^k)^{-1} \bigg\{ \sum_{\tau=1}^{k-1} \widehat{\phi}_h(s_h^\tau, a_h^\tau) \bigg\} \bigg| H \xi_{\mathrm{down}}$$

$$= \sum_{\tau=1}^{k-1} \big| \widehat{\phi}_h(s,a)^\top (\Lambda_h^k)^{-1} \widehat{\phi}_h(s_h^\tau, a_h^\tau) \big| H \xi_{\mathrm{down}}$$

$$\overset{(ii)}{\leq} \sqrt{ \left( \sum_{\tau=1}^{k-1} \big\| \widehat{\phi}_h(s,a) \big\|_{(\Lambda_h^k)^{-1}}^2 \right) \left( \sum_{\tau=1}^{k-1} \big\| \widehat{\phi}_h(s_h^\tau, a_h^\tau) \big\|_{(\Lambda_h^k)^{-1}}^2 \right)} H \xi_{\mathrm{down}}$$

$$\overset{(iii)}{\leq} H \xi_{\mathrm{down}} \sqrt{dk} \big\| \widehat{\phi}_h(s,a) \big\|_{(\Lambda_h^k)^{-1}}, \tag{J.5}$$

where $(i)$ follows from Lemma 4.3, $(ii)$ follows from Cauchy-Schwarz inequality and $(iii)$ follows from Lemma O.4.

Substituting (J.3), (J.4), (J.5), into (J.2), and denoting $\beta = C_L H \sqrt{d} + dH \sqrt{\log \left( \frac{dK_{\mathrm{RFE}} H \max(\xi_{\mathrm{down}}, 1)}{\delta} \right)} + H \xi_{\mathrm{down}} \sqrt{dk}$ we get

$$\Big| \widehat{\phi}_h(s,a)^\top \widehat{w}_h^k - \widehat{\phi}_h(s,a)^\top \widetilde{w}_h^k \Big| \lesssim \beta \big\| \widehat{\phi}_h(s,a) \big\|_{(\Lambda_h^k)^{-1}}$$

Putting everything together in (J.1), we get

$$\Big| \widehat{\phi}_h(s,a)^\top \widehat{w}_h^k - P_h^{(*,T+1)} \widehat{V}_{h+1}^k(s,a) \Big| \lesssim \beta \big\| \widehat{\phi}_h(s,a) \big\|_{(\Lambda_h^k)^{-1}} + H \xi_{\mathrm{down}},$$

which concludes the proof. $\qquad\qquad\qquad\qquad\qquad\qquad\qquad\qquad\qquad\qquad\qquad\qquad \Box$

### J.4    Proof of Lemma I.5

*Proof of Lemma I.5.* We prove the first part using backward induction on $h$. For $h = H$, we have

$$\widetilde{V}_H^*(s, r^k) = \min \bigg\{ \max_{a \in \mathcal{A}} \Big\{ r_H^k(s,a) + P_H^{(*,T+1)} \widetilde{V}_{H+1}^*(s, a, r^k) \Big\}, H \bigg\}$$

$$= \min \bigg\{ \max_{a \in \mathcal{A}} r_H^k(s,a), H \bigg\}$$

$$\leq \min \bigg\{ \max_{a \in \mathcal{A}} \Big\{ r_H^k(s,a) + \widehat{\phi}_H(s,a)^\top \widehat{w}_H^k + \beta \big\| \widehat{\phi}_H(s,a) \big\|_{(\Lambda_H^k)^{-1}} \Big\}, H \bigg\}$$

$$= \widehat{V}_H^k(s)$$

$$\leq \widehat{V}_H^k(s) + H(H - H + 1) \xi_{\mathrm{down}}.$$

Suppose for some $h + 1 \in [H]$, it holds that for all $s \in \mathcal{S}$,

$$\widetilde{V}^*_{h+1}(s, r^k) \le \widehat{V}^k_{h+1}(s) + H(H - h)\xi_{\text{down}}.$$

Then,

$$
\begin{aligned}
\widetilde{V}^*_h(s, r^k) &= \min\left\{\max_{a \in \mathcal{A}}\left\{r^k_h(s, a) + P^{(*,T+1)}_h \widetilde{V}^*_{h+1}(s, a, r^k)\right\}, H\right\} \\
&\le \max_{a \in \mathcal{A}}\left\{r^k_h(s, a) + P^{(*,T+1)}_h \widetilde{V}^*_{h+1}(s, a, r^k)\right\} \\
&\le \max_{a \in \mathcal{A}}\left\{r^k_h(s, a) + P^{(*,T+1)}_h \widehat{V}^k_{h+1}(s, a)\right\} + H(H - h)\xi_{\text{down}} \\
&\lesssim \max_{a \in \mathcal{A}}\left\{r^k_h(s, a) + \widehat{\phi}_h(s, a)^\top \widehat{w}^k_h + \beta\|\widehat{\phi}_h(s, a)\|_{(\Lambda^k_h)^{-1}}\right\} + H\xi_{\text{down}} + H(H - h)\xi_{\text{down}} \\
&= \max_{a \in \mathcal{A}}\left\{r^k_h(s, a) + \widehat{\phi}_h(s, a)^\top \widehat{w}^k_h + \beta\|\widehat{\phi}_h(s, a)\|_{(\Lambda^k_h)^{-1}}\right\} + H(H - h + 1)\xi_{\text{down}},
\end{aligned}
$$

where the last inequality follows from Lemma I.4.

Thus, we have

$$
\begin{aligned}
\widetilde{V}^*_h(s, r^k) &\lesssim \min\left\{\max_{a \in \mathcal{A}}\left\{r^k_h(s, a) + \widehat{\phi}_h(s, a)^\top \widehat{w}^k_h + \beta\|\widehat{\phi}_h(s, a)\|_{(\Lambda^k_h)^{-1}}\right\}, H\right\} + H(H - h + 1)\xi_{\text{down}} \\
&= \widehat{V}^k_h(s) + H(H - h + 1)\xi_{\text{down}},
\end{aligned}
$$

as desired. This completes the first part of the proof.

Now we prove the second part of the proof. For all $(k, h) \in [K_{\text{RFE}}] \times [H - 1]$, we denote,

$$\xi^k_h = P^{(*,T+1)}_h \widehat{V}^k_{h+1}(s^k_h, a^k_h) - \widehat{V}^k_{h+1}(s^k_{h+1}).$$

Conditioned on $\mathcal{E}$ from Lemma I.3 where $\Pr[\mathcal{E}] \ge 1 - \delta/8$,

$$
\begin{aligned}
\sum_{k=1}^{K_{\text{RFE}}} \widehat{V}^k_1(s^k_1) &\le \sum_{k=1}^{K_{\text{RFE}}}\left(r^k_1(s^k_1, a^k_1) + \widehat{\phi}_1(s^k_1, a^k_1)^\top \widehat{w}^k_1 + \beta\|\widehat{\phi}_1(s^k_1, a^k_1)\|_{(\Lambda^k_1)^{-1}}\right) \\
&= \sum_{k=1}^{K_{\text{RFE}}}\left(\widehat{\phi}_1(s^k_1, a^k_1)^\top \widehat{w}^k_1 + 2\beta\|\widehat{\phi}_1(s^k_1, a^k_1)\|_{(\Lambda^k_1)^{-1}}\right) \\
&\lesssim \sum_{k=1}^{K_{\text{RFE}}}\left(P^{(*,T+1)}_1 \widehat{V}^k_2(s^k_1, a^k_1) + 3\beta\|\widehat{\phi}_1(s^k_1, a^k_1)\|_{(\Lambda^k_1)^{-1}} + H\xi_{\text{down}}\right) \\
&= \sum_{k=1}^{K_{\text{RFE}}}\left(\xi^k_1 + \widehat{V}^k_2(s^k_2) + 3\beta\|\widehat{\phi}_1(s^k_1, a^k_1)\|_{(\Lambda^k_1)^{-1}}\right) + HK_{\text{RFE}}\xi_{\text{down}} \\
&\le \cdots \\
&\le \sum_{k=1}^{K_{\text{RFE}}}\sum_{h=1}^{H-1} \xi^k_h + \sum_{k=1}^{K_{\text{RFE}}}\sum_{h=1}^{H} 3\beta\|\widehat{\phi}_h(s^k_h, a^k_h)\|_{(\Lambda^k_h)^{-1}} + H^2 K_{\text{RFE}}\xi_{\text{down}},
\end{aligned}
$$

where the second inequality follows from Lemma I.4.

Note that for each $h \in [H - 1]$, $\{\xi^k_h\}_{k=1}^{K_{\text{RFE}}}$ is a martingale difference sequence with $|\xi^k_h| \le H$. We define the event $\mathcal{E}'$ to be the event that

$$\left|\sum_{k=1}^{K_{\text{RFE}}}\sum_{h=1}^{H-1} \xi^k_h\right| \le c'H^2\sqrt{K_{\text{RFE}}\log(K_{\text{RFE}}H/\delta)}.$$

By Azuma-Hoeffding inequality, we have $\Pr[\mathcal{E}'] \geq 1 - \delta/8$.

By Cauchy-Schwarz inequality, we have

$$\sum_{k=1}^{K_{\text{RFE}}} \sum_{h=1}^{H} \|\widehat{\phi}_h(s_h^k, a_h^k)\|_{(\Lambda_h^k)^{-1}} \leq \sqrt{K_{\text{RFE}} H \sum_{k=1}^{K_{\text{RFE}}} \sum_{h=1}^{H} \widehat{\phi}_h(s_h^k, a_h^k)^\top (\Lambda_h^k)^{-1} \widehat{\phi}_h(s_h^k, a_h^k)}.$$

Using Lemma D.2 of Jin et al. (2020b), we have

$$\sum_{k=1}^{K_{\text{RFE}}} \sum_{h=1}^{H} \widehat{\phi}_h(s_h^k, a_h^k)^\top (\Lambda_h^k)^{-1} \widehat{\phi}_h(s_h^k, a_h^k) \leq 2dH \log(K_{\text{RFE}}).$$

Conditioned on $\mathcal{E}$ and $\mathcal{E}$ where $\Pr(\mathcal{E} \cap \mathcal{E}') \geq 1 - \delta/4$, we have

$$\sum_{k=1}^{K_{\text{RFE}}} \widehat{V}_1^k(s_1^k) \leq c'H^2\sqrt{K_{\text{RFE}} \log(K_{\text{RFE}}H/\delta)} + 3\beta\sqrt{K_{\text{RFE}}H \cdot 2dH \log(K_{\text{RFE}})} + H^2 K_{\text{RFE}} \xi_{\text{down}}$$

$$\leq c\sqrt{d^3 H^4 K_{\text{RFE}} \log(dK_{\text{RFE}}H/\delta)} + H^2 K_{\text{RFE}} \xi_{\text{down}},$$

which completes the proof. $\qquad\qquad\qquad\qquad\qquad\qquad\qquad\qquad\qquad\qquad\qquad\qquad$ $\square$

## J.5 Proof of Lemma I.6

*Proof of Lemma I.6.* Denote $\Delta^k = \widetilde{V}_1^*(s_1^k, r^k) - \mathbb{E}_{s \sim \mu}[\widetilde{V}_1^*(s, r^k)]$. Note that $r^k$ depends only on the data collected during the first $k-1$ episodes. Thus, $\{\Delta^k\}_{k=1}^{K_{\text{RFE}}}$ is a martingale difference sequence. Moreover, we have $|\Delta^k| \leq H$. Using Azuma-Hoeffding inequality, with probability $1 - \delta/4$, we have

$$\left| \sum_{k=1}^{K_{\text{RFE}}} \Delta^k \right| \leq c_1 H \sqrt{K_{\text{RFE}} \log(1/\delta)},$$

where $c_1 > 0$ is an absolute constant. Therefore, we have

$$\mathbb{E}_{s \sim \mu}\left[ \sum_{k=1}^{K_{\text{RFE}}} \widetilde{V}_1^*(s, r^k) \right] \leq \sum_{k=1}^{K_{\text{RFE}}} \widetilde{V}_1^*(s_1^k, r^k) + c_1 H \sqrt{K_{\text{RFE}} \log(1/\delta)}.$$

Now, notice that for all $k \in [K_{\text{RFE}}]$, $\Lambda_h \succeq \Lambda_h^k$. Thus for all $(k, h) \in [K_{\text{RFE}}] \times [H]$, we have

$$r_h^k(\cdot, \cdot) \geq u_h(\cdot, \cdot),$$

which implies

$$V_1^*(\cdot, u_h) \leq V_1^*(\cdot, r_h^k).$$

Using Lemma I.5 and union bound, we have with probability $1 - \delta/2$,

$$\mathbb{E}_{s\sim\mu}\big[\widetilde{V}_1^*(s,u)\big] \leq \mathbb{E}_{s\sim\mu}\left[\sum_{k=1}^{K_{\text{RFE}}} \widetilde{V}_1^*(s,r^k)/K_{\text{RFE}}\right]$$

$$\leq \frac{1}{K_{\text{RFE}}}\sum_{k=1}^{K_{\text{RFE}}} \widetilde{V}_1^*(s_1^k,r^k) + c_1 H\sqrt{\frac{\log(1/\delta)}{K_{\text{RFE}}}}$$

$$\leq \frac{1}{K_{\text{RFE}}}\sum_{k=1}^{K_{\text{RFE}}} \big(\widehat{V}_1^k(s_1^k) + H^2\xi_{\text{down}}\big) + c_1 H\sqrt{\frac{\log(1/\delta)}{K_{\text{RFE}}}}$$

$$= \frac{1}{K_{\text{RFE}}}\sum_{k=1}^{K_{\text{RFE}}} \widehat{V}_1^k(s_1^k) + H^2\xi_{\text{down}} + c_1 H\sqrt{\frac{\log(1/\delta)}{K_{\text{RFE}}}}$$

$$\leq \frac{1}{K_{\text{RFE}}}\big(c\sqrt{d^3 H^4 K_{\text{RFE}}\log(dK_{\text{RFE}}H/\delta)} + H^2 K_{\text{RFE}}\xi_{\text{down}}\big) + H^2\xi_{\text{down}} + c_1 H\sqrt{\frac{\log(1/\delta)}{K_{\text{RFE}}}}$$

$$= c\sqrt{\frac{d^3 H^4 \log(dK_{\text{RFE}}H/\delta)}{K_{\text{RFE}}}} + c_1 H\sqrt{\frac{\log(1/\delta)}{K_{\text{RFE}}}} + 2H^2\xi_{\text{down}}$$

$$\leq c'\sqrt{\frac{d^3 H^4 \log(dK_{\text{RFE}}H/\delta)}{K_{\text{RFE}}}} + 2H^2\xi_{\text{down}},$$

for some absolute constant $c' > 0$. This completes the proof. $\qquad\square$

## J.6 Proof of Lemma I.7

*Proof of Lemma I.7.* Following the same argument as in the proof of Lemma I.4, for all $h \in [H]$ and $(s,a) \in \mathcal{S} \times \mathcal{A}$, with probability $1 - \delta/4$, we have

$$\left|\widehat{\phi}_h(s,a)^\top \widehat{w}_h - P_h^{(*,T+1)}\widehat{V}_{h+1}(s,a)\right| \lesssim \beta\|\widehat{\phi}_h(s,a)\|_{(\Lambda_h)^{-1}} + H\xi_{\text{down}}$$

Thus, for all $h \in [H]$ and $(s,a) \in \mathcal{S} \times \mathcal{A}$, we have

$$\widehat{Q}_h(s,a) \leq \widehat{\phi}_h(s,a)^\top \widehat{w}_h + r_h(s,a) + u_h(s,a)$$
$$\lesssim r_h(s,a) + P_h^{(*,T+1)}\widehat{V}_{h+1}(s,a) + 2\beta\|\widehat{\phi}_h(s,a)\|_{(\Lambda_h)^{-1}} + H\xi_{\text{down}}.$$

Since, $\widehat{Q}_h(s,a) \leq H$ and $u_h(\cdot,\cdot) = \min\left\{\beta\sqrt{\widehat{\phi}(\cdot,\cdot)^\top(\Lambda_h)^{-1}\widehat{\phi}(\cdot,\cdot)}, H\right\}$, we have,

$$\widehat{Q}_h(s,a) \leq r_h(s,a) + P_h^{(*,T+1)}\widehat{V}_{h+1}(s,a) + 2u_h(s,a) + H\xi_{\text{down}}.$$

This completes the first part of the proof.

Now, using induction, we prove that for all $h \in [H]$ and $(s,a) \in \mathcal{S} \times \mathcal{A}$, we have $Q_h^*(s,a,r) - H(H - h + 1)\xi_{\text{down}} \leq \widehat{Q}_h(s,a)$.

When $h = H + 1$, the claim is trivially true. Suppose for some $h \in [H]$, we have, for all $(s,a) \in \mathcal{S} \times \mathcal{A}$,

$$Q_{h+1}^*(s,a,r) - H(H - h)\xi_{\text{down}} \leq \widehat{Q}_h(s,a).$$

Note that,

$$\widehat{Q}_h(s,a) = \min\left\{\widehat{\phi}_h(s,a)^\top \widehat{w}_h + r_h(s,a) + u_h(s,a), H\right\}$$

Since $Q_h^*(s,a,r) \leq H$ and $u_h(\cdot,\cdot) = \min\left\{\beta\sqrt{\widehat{\phi}(\cdot,\cdot)^\top(\Lambda_h)^{-1}\widehat{\phi}(\cdot,\cdot)}, H\right\}$, it suffices to show that

$$Q_h^*(s, a, r) \le \widehat{\phi}_h(s, a)^\top \widehat{w}_h + r_h(s, a) + \beta \|\widehat{\phi}_h(s, a)\|_{(\Lambda_h)^{-1}} + H(H - h + 1)\xi_{\text{down}}.$$

Applying the $\max$ operator on both side of the inductive hypothesis, we get

$$V_{h+1}^*(s, r) \le \widehat{V}_{h+1} + H(H - h)\xi_{\text{down}}.$$

Now,

$$
\begin{aligned}
Q_h^*(s, a, r) &= r_h(s, a) + P_h^{(*, T+1)} V_{h+1}^*(s, a, r) \\
&\le r_h(s, a) + P_h^{(*, T+1)} \widehat{V}_{h+1}(s, a) + H(H - h)\xi_{\text{down}} \\
&\lesssim \widehat{\phi}_h(s, a)^\top \widehat{w}_h + \beta \|\widehat{\phi}_h(s, a)\|_{(\Lambda_h)^{-1}} + H\xi_{\text{down}} + H(H - h)\xi_{\text{down}} + r_h(s, a) \\
&= \widehat{\phi}_h(s, a)^\top \widehat{w}_h + r_h(s, a) + \beta \|\widehat{\phi}_h(s, a)\|_{(\Lambda_h)^{-1}} + H(H - h + 1)\xi_{\text{down}}.
\end{aligned}
$$

This completes the proof. $\qquad\square$

## K    Proof for Downstream Offline RL

First, we state the supporting lemmas that are used in the proof of Theorem 4.6. The proof of these lemmas are provided in Appendix L.

### K.1    Supporting Lemmas

The following lemma shows that the linear weight $\widehat{w}_h$ in Algorithm 4 is bounded.

**Lemma K.1** (Bounds on Weights in Algorithm 4). *For any $h \in [H]$, the weight $\widehat{w}_h$ in Algorithm 4 satisfies*

$$\|\widehat{w}_h\|_2 \le H\sqrt{dN_{\textit{off}}/\lambda_d}.$$

Next, we present our main concentration lemma for this section that upper-bounds the stochastic noise in regression.

**Lemma K.2.** *Setting* $\lambda_d = 1$, $\beta(\delta) = c_\beta(Hd\sqrt{\iota(\delta)} + H\sqrt{dN_{\textit{off}}}\xi_{down})$, *where* $\iota(\delta) = \log(HdN_{\textit{off}} \max(\xi_{down}, 1)/\delta)$, *with probability at least* $1 - \delta$, *for all* $h \in [H]$, *we have*

$$\left\| \sum_{\tau=1}^{N_{\textit{off}}} \widehat{\phi}_h(s_h^\tau, a_h^\tau) \Big[ (P_h^{(*, T+1)} \widehat{V}_{h+1})(s_h^\tau, a_h^\tau) - \widehat{V}_{h+1}(s_{h+1}^\tau) \Big] \right\|_{\Lambda_h^{-1}} \lesssim Hd\sqrt{\iota}.$$

Recall that, we define the Bellman operator $\mathbb{B}_h$ as $(\mathbb{B}_h f)(s, a) = r_h(s, a) + (P_h^{(*, T+1)} f)(s, a)$ for any $f : \mathcal{S} \times \mathcal{A} \to \mathbb{R}$. In the next lemma, we denote the Bellman estimate $(\widehat{\mathbb{B}}_h \widehat{V}_{h+1})(\cdot, \cdot) = \widehat{\phi}_h(\cdot, \cdot)^\top \widehat{w}_h$. This lemma provides an upper bound for the Bellman update error $|(\mathbb{B}_h \widehat{V}_{h+1} - \widehat{\mathbb{B}}_h \widehat{V}_{h+1})(s, a)|$ and characterizes the impact of the misspecification of the representation taken from the upstream learning.

**Lemma K.3** (Bound on Bellman update error). *Set* $\lambda_d = 1$, $\beta = c_\beta(Hd\sqrt{\iota} + H\sqrt{dN_{\textit{off}}}\xi_{down})$, *where* $\iota = \log(HdN_{\textit{off}} \max(\xi_{down}, 1)/\delta)$. *Define the following event*

$$\mathcal{E}(\delta) = \left\{ |(\mathbb{B}_h \widehat{V}_{h+1} - \widehat{\mathbb{B}}_h \widehat{V}_{h+1})(s, a)| \le \beta \|\widehat{\phi}_h(s, a)\|_{\Lambda_h^{-1}} + H\xi_{down}, \forall h \in [H] \text{ and } \forall (s, a) \in \mathcal{S} \times \mathcal{A} \right\}.$$

*Then under Assumption 4.1, we have* $\mathbb{P}(\mathcal{E}(\delta)) \ge 1 - \delta$.

**Definition K.4** (Model prediction error). For all $h \in [H]$, we define the model prediction error as,

$$l_h(s, a) = (\mathbb{B}_h \widehat{V}_{h+1})(s, a) - \widehat{Q}_h(s, a).$$

The following lemma decomposes the suboptimality gap into the summation of uncertainty metric of each step.

**Lemma K.5.** *Let $\{\widehat{\pi}\}_{h=1}^{H}$ be the output of Algorithm 4. Conditioned on the event $\mathcal{E}(\delta)$ defined in Lemma K.3, for any $h \in [H]$ and $(s,a) \in \mathcal{S} \times \mathcal{A}$, we have*

$$0 \leq l_h(s,a) \leq 2\Gamma_h(s,a),$$

*where $\Gamma_h(s,a) = \beta\big\|\widehat{\phi}_h(s,a)\big\|_{\Lambda_h^{-1}} + H\xi_{down}$. Furthermore, we have*

$$V_{P^{(*,T+1)},r}^{\pi^*}(s) - V_{P^{(*,T+1)},r}^{\widehat{\pi}}(s) \leq 2\sum_{h=1}^{H} \mathbb{E}_{\pi^*}[\Gamma_h(s_h, a_h)|s_1 = s]$$

## K.2 Proof of Theorem 4.6

We first restate Theorem 4.6.

**Theorem K.6.** *Under Assumption 4.1, setting $\lambda_d = 1$, $\beta = O(Hd\sqrt{\iota} + H\sqrt{dN_{off}}\xi_{down})$, where $\iota = \log(HdN_{off}\max(\xi_{down}, 1)/\delta)$, with probability at least $1-\delta$, the suboptimality gap of Algorithm 4 is at most*

$$V_{P^{(*,T+1)},r}^{\pi^*}(s) - V_{P^{(*,T+1)},r}^{\widehat{\pi}}(s) \leq 2H^2\xi_{down} + 2\beta\sum_{h=1}^{H} \mathbb{E}_{\pi^*}\left[\big\|\widehat{\phi}_h(s_h, a_h)\big\|_{\Lambda_h^{-1}}\big|s_1 = s\right]. \quad \text{(K.1)}$$

*Additionally if Assumption 4.5 holds, and the sample size satisfies $N_{off} \geq 40/\kappa_\rho \cdot \log(4dH/\delta)$, then with probability $1 - \delta$, we have,*

$$V_{P^{(*,T+1)},r}^{\pi^*}(s) - V_{P^{(*,T+1)},r}^{\widehat{\pi}}(s)$$

$$\leq O\left(\kappa_\rho^{-1/2}H^2d^{1/2}\xi_{down} + \kappa_\rho^{-1/2}H^2d\sqrt{\frac{\log(HdN_{off}\max(\xi_{down}, 1)/\delta)}{N_{off}}}\right). \quad \text{(K.2)}$$

*Remark* K.7. Compared to Theorem 4.4 in Jin et al. (2021), the suboptimality gap in (K.1) has an additional term $2H^2\xi_{down}$. When the linear misspecification error $\xi = \widetilde{O}(\sqrt{d/N_{off}})$ and the number of trajectories $n$ in each upstream offline task dataset satisfies $n = \widetilde{O}\big(\frac{TN_{off}}{d}\big)$, the RHS of (K.2) is dominated by $\widetilde{O}(N_{off}^{-1/2}H^2d)$ improving the suboptimality gap bound of REP-LCB (Uehara et al., 2022) by an order of $\widetilde{O}(Hd)$ under low-rank MDP with unknown representation.

*Proof of Theorem K.6.* Letting $\Gamma_h = \beta\|\widehat{\phi}_h(s,a)\|_{\Lambda_h^{-1}} + H\xi_{down}$ in Lemma K.5, with probability at least $1 - \delta$, we have

$$V_{P^{(*,T+1)},r}^{\pi^*}(s) - V_{P^{(*,T+1)},r}^{\widehat{\pi}}(s) \leq 2\sum_{h=1}^{H} \mathbb{E}_{\pi^*}[\Gamma_h(s_h, a_h)|s_1 = s]$$

$$\leq 2H^2\xi_{down} + 2\beta\sum_{h=1}^{H} \mathbb{E}_{\pi^*}\left[\big\|\widehat{\phi}_h(s_h, a_h)\big\|_{\Lambda_h^{-1}}\big|s_1 = s\right]$$

This finishes the first part of the proof. Now we will provide the suboptimality bound under the feature coverage assumption in Assumption 4.5. From Appendix B.4 of Jin et al. (2021), if $N_{off} \geq 40/\kappa_\rho \cdot \log(4dH/\delta)$, then with probability $1 - \delta/2$, for any $h \in [H]$ and $(s,a) \in \mathcal{S} \times \mathcal{A}$, we have

$$\big\|\widehat{\phi}_h(s,a)\big\|_{\Lambda_h^{-1}} \leq \sqrt{\frac{2}{\kappa_\rho}} \cdot \frac{1}{\sqrt{N_{off}}}.$$

Setting $\beta(\delta/2) = c_\beta(Hd\sqrt{\iota(\delta/2)} + H\sqrt{dN_{off}}\xi_{down})$, with probability $1 - \delta/2$, we have

$$V_{P^{(*,T+1)},r}^{\pi^*}(s) - V_{P^{(*,T+1)},r}^{\widehat{\pi}}(s) \leq 2H^2\xi_{down} + 2\beta\sum_{h=1}^{H} \mathbb{E}_{\pi^*}\left[\big\|\widehat{\phi}_h(s_h, a_h)\big\|_{\Lambda_h^{-1}}\big|s_1 = s\right].$$

Using union bound, we have the following bound with probability at least $1 - \delta$

$$V_{P^{(*,T+1)},r}^{\pi^*}(s) - V_{P^{(*,T+1)},r}^{\widehat{\pi}}(s)$$

$$\leq 2H\left(H\xi_{\text{down}} + \beta(\delta/2)\sqrt{\frac{2}{\kappa_\rho}} \cdot \frac{1}{\sqrt{N_{\text{off}}}}\right)$$

$$= O\left(\kappa_\rho^{-1/2} H^2 d^{1/2}\xi_{\text{down}} + \kappa_\rho^{-1/2} H^2 d\sqrt{\frac{\log(HdN_{\text{off}}\max(\xi_{\text{down}},1)/\delta)}{N_{\text{off}}}}\right). \qquad \text{(K.3)}$$

$\square$

## L  Proof of Supporting Lemmas in Appendix K

In this section, we provide the proofs of the lemmas that we used in the proof of Theorem 4.6.

### L.1  Proof of Lemma K.1

*Proof of Lemma K.1.* We have

$$\|\widehat{w}_h\| = \left\|\Lambda_h^{-1} \sum_{\tau=1}^{N_{\text{off}}} \widehat{\phi}_h(s_h^\tau, a_h^\tau)(r_h^\tau + \widehat{V}_{h+1}(s_{h+1}^\tau))\right\|$$

$$\leq \sqrt{\frac{N_{\text{off}}}{\lambda_d}}\left(\sum_{\tau=1}^{N_{\text{off}}} \left\|(r_h^\tau + \widehat{V}_h(s_{h+1}^\tau))\widehat{\phi}_h(s_h^\tau, a_h^\tau)\right\|_{\Lambda_h^{-1}}^2\right)^{1/2}$$

$$\leq H\sqrt{\frac{N_{\text{off}}}{\lambda_d}}\left(\sum_{\tau=1}^{N_{\text{off}}} \left\|\widehat{\phi}_h(s_h^\tau, a_h^\tau)\right\|_{\Lambda_h^{-1}}^2\right)^{1/2}$$

$$\leq H\sqrt{\frac{dN_{\text{off}}}{\lambda_d}},$$

where the first inequality follows from Lemma O.5 and the fact that the largest eigenvalue of $\Lambda_h^{-1}$ is at most $1/\lambda_d$, second inequality follows from the fact that $r_h^\tau \in [0, 1]$ and $|\widehat{V}_{h+1}(s)| \leq H - 1$ for all $s \in \mathcal{S}$ and the last inequality follows from Lemma O.4. $\square$

### L.2  Proof of Lemma K.2

*Proof of Lemma K.2.* The value function $\widehat{V}_{h+1}$ has the parametric form of

$$V(\cdot) = \min\left\{\max_{a \in \mathcal{A}} w^\top \phi(\cdot, a) - \beta\sqrt{\phi(\cdot, a)^\top \Lambda^{-1} \phi(\cdot, a)}, H - h + 1\right\},$$

where $w \in \mathbb{R}^d$ and positive definite matrix $\Lambda$ is such that its minimum eigenvalue satisfies $\lambda_{\min}(\Lambda) \geq \lambda_d$. From Lemma K.1, we have $\|\widehat{w}_h\| \leq H\sqrt{dN_{\text{off}}/\lambda_d}$. Thus, applying Lemma O.4 and Lemma O.9, we have, for any fixed $\varepsilon > 0$ and for all $h \in [H]$, with probability at least $1 - \delta$,

$$\left\|\sum_{\tau=1}^{N_{\text{off}}} \widehat{\phi}_h(s_h^\tau, a_h^\tau)\left[(P_h^{(*,T+1)}\widehat{V}_{h+1})(s_h^\tau, a_h^\tau) - \widehat{V}_{h+1}(s_{h+1}^\tau)\right]\right\|_{\Lambda_h^{-1}}^2$$

$$\leq 4H^2\left[\frac{d}{2}\log\left(\frac{N_{\text{off}} + \lambda_d}{\lambda_d}\right) + \log\frac{\mathcal{N}_\varepsilon}{\delta}\right] + \frac{8N_{\text{off}}^2 \varepsilon^2}{\lambda_d}$$

$$\leq 4H^2\left[\frac{d}{2}\log\left(\frac{N_{\text{off}} + \lambda_d}{\lambda_d}\right) + d\log\left(1 + \frac{4H\sqrt{dN_{\text{off}}}}{\varepsilon\sqrt{\lambda_d}}\right) + d^2\log\left(1 + \frac{8\sqrt{d}\beta^2}{\lambda_d\varepsilon^2}\right) + \log\frac{1}{\delta}\right] + \frac{8N_{\text{off}}^2\varepsilon^2}{\lambda_d}$$

$$\leq 4H^2\left[\frac{d}{2}\log\left(\frac{N_{\text{off}} + \lambda_d}{\lambda_d}\right) + d\log\left(1 + \frac{4H\sqrt{dN_{\text{off}}}}{\varepsilon\sqrt{\lambda_d}}\right) + d^2\log\left(1 + \frac{8\sqrt{d}\beta^2}{\lambda_d\varepsilon^2}\right) + \log\frac{1}{\delta}\right] + \frac{8N_{\text{off}}^2\varepsilon^2}{\lambda_d}.$$

$$\text{(L.1)}$$

Setting $\varepsilon = dH/N_{\text{off}}$, $\lambda_d = 1$, $\beta = c_\beta(Hd\sqrt{\iota} + H\sqrt{dN_{\text{off}}}\xi_{\text{down}})$, where $\iota = \log(HdN_{\text{off}}\max(\xi_{\text{down}},1)/\delta)$, we can further upper bound (L.1) by

$$4H^2\left[\frac{d}{2}\log(1+N_{\text{off}}) + d\log(1+4d^{-1/2}N_{\text{off}}^{3/2}) + d^2\log(1+8d^{-3/2}H^{-2}N_{\text{off}}^2\beta^2) + \log(1/\delta)\right] + 8d^2H^2$$

$$\lesssim H^2d\log N_{\text{off}} + H^2d\log(d^{-1/2}N_{\text{off}}^{3/2}) + H^2\log(1/\delta) + H^2d^2\log(Hd^{1/2}\iota N_{\text{off}}^3\xi_{\text{down}}) + d^2H^2$$

$$\lesssim H^2d^2\iota$$

Therefore, we have

$$\left\|\sum_{\tau=1}^{N_{\text{off}}}\widehat{\phi}_h(s_h^\tau, a_h^\tau)\Big[(P_h^{(*,T+1)}\widehat{V}_{h+1})(s_h^\tau, a_h^\tau) - \widehat{V}_{h+1}(s_{h+1}^\tau)\Big]\right\|_{\Lambda_h^{-1}} \lesssim Hd\sqrt{\iota}.$$

$\square$

## L.3 Proof of Lemma K.3

*Proof of Lemma K.3.* For $h \in [H]$, we define $\widehat{w}_h^* = \int_{s'}\widehat{\mu}^*(s')\widehat{V}_{h+1}(s')ds'$. Then we have

$$\left|(\mathbb{B}_h\widehat{V}_{h+1} - \widehat{B}_h\widehat{V}_{h+1})(s,a)\right|$$

$$= \left|(\mathbb{B}_h\widehat{V}_{h+1} - \overline{\mathbb{B}}_h\widehat{V}_{h+1} + \overline{\mathbb{B}}_h\widehat{V}_{h+1} - \widehat{B}_h\widehat{V}_{h+1})(s,a)\right|$$

$$\leq \left|(\mathbb{B}_h\widehat{V}_{h+1} - \overline{\mathbb{B}}_h\widehat{V}_{h+1})(s,a)\right| + \left|(\overline{\mathbb{B}}_h\widehat{V}_{h+1} - \widehat{B}_h\widehat{V}_{h+1})(s,a)\right|$$

$$= \left|(P_h^{(*,T+1)}\widehat{V}_{h+1} - \overline{P}_h\widehat{V}_{h+1})(s,a)\right| + \left|(\overline{P}_h\widehat{V}_{h+1})(s,a) - \widehat{\phi}_h(s,a)^\top\widehat{w}_h\right|$$

$$\overset{(i)}{\leq} H\left\|P_h^{(*,T+1)}(\cdot|s,a) - \langle\widehat{\phi}_h(s,a),\widehat{\mu}_h^*(\cdot)\rangle\right\|_{\text{TV}} + \left|\int_{s'}\langle\widehat{\phi}_h(s,a),\widehat{\mu}_h^*(s')\rangle\widehat{V}_{h+1}(s')ds' - \widehat{\phi}_h(s,a)^\top\widehat{w}_h\right|$$

$$\overset{(ii)}{\leq} H\xi_{\text{down}} + \left|\widehat{\phi}_h(s,a)^\top(\widehat{w}_h^* - \widehat{w})\right|, \tag{L.2}$$

where $(i)$ follows from the fact that $|\widehat{V}_{h+1}(s)| \leq H$ for all $s \in \mathcal{S}$ and $(ii)$ follows from Lemma 4.3.

We now decompose the second term in (L.2). Recall that $\Lambda_h = \sum_{\tau=1}^{N_{\text{off}}}\widehat{\phi}_h(s_h^\tau, a_h^\tau)\widehat{\phi}_h(s_h^\tau, a_h^\tau)^\top + \lambda_d I_d$ and $\widehat{w}_h = \Lambda_h^{-1}\sum_{\tau=1}^{N_{\text{off}}}\widehat{\phi}_h(s_h^\tau, a_h^\tau)\widehat{V}_{h+1}(s_{h+1}^\tau)$. Then, we have

$$\widehat{\phi}_h(s,a)^\top(\widehat{w}_h^* - \widehat{w})$$

$$= \widehat{\phi}_h(s,a)^\top\Lambda_h^{-1}\left\{\left(\sum_{\tau=1}^{N_{\text{off}}}\widehat{\phi}_h(s_h^\tau, a_h^\tau)\widehat{\phi}_h(s_h^\tau, a_h^\tau)^\top + \lambda_d I_d\right)\widehat{w}_h^* - \left(\sum_{\tau=1}^{N_{\text{off}}}\widehat{\phi}_h(s_h^\tau, a_h^\tau)\widehat{V}_{h+1}(s_{h+1}^\tau)\right)\right\}$$

$$= \underbrace{\lambda_d\widehat{\phi}_h(s,a)^\top\Lambda_h^{-1}\widehat{w}_h^*}_{(a)} + \underbrace{\widehat{\phi}_h(s,a)^\top\Lambda_h^{-1}\left\{\sum_{\tau=1}^{N_{\text{off}}}\widehat{\phi}_h(s_h^\tau, a_h^\tau)\Big[(P_h^{(*,T+1)}\widehat{V}_{h+1})(s_h^\tau, a_h^\tau) - \widehat{V}_{h+1}(s_{h+1}^\tau)\Big]\right\}}_{(b)}$$

$$+ \underbrace{\widehat{\phi}_h(s,a)^\top\Lambda_h^{-1}\left\{\sum_{\tau=1}^{N_{\text{off}}}\widehat{\phi}_h(s_h^\tau, a_h^\tau)\Big[(\overline{P}_h\widehat{V}_{h+1} - P_h^{(*,T+1)}\widehat{V}_{h+1})(s_h^\tau, a_h^\tau)\Big]\right\}}_{(c)} \tag{L.3}$$

We now provide an upper bound for each of the terms in (L.3).

**Term (a).** We have

$$\lambda_d\widehat{\phi}_h(s,a)^\top\Lambda_h^{-1}\widehat{w}_h^* \overset{(i)}{\leq} \lambda_d\left\|\widehat{\phi}_h(s,a)\right\|_{\Lambda_h^{-1}}\|\widehat{w}_h^*\|_{\Lambda_h^{-1}}$$

$$\overset{(ii)}{\leq} \sqrt{\lambda_d}\left\|\widehat{\phi}_h(s,a)\right\|_{\Lambda_h^{-1}}\|\widehat{w}_h^*\|_2$$

$$\overset{(iii)}{\leq} \sqrt{\lambda_d Hd}\left\|\widehat{\phi}_h(s,a)\right\|_{\Lambda_h^{-1}}, \tag{L.4}$$

where $(i)$ follows from Cauchy-Schwarz inequality, $(ii)$ follows from the fact that the largest eigenvalue of $\Lambda_h^{-1}$ is at most $1/\lambda_d$ and $(iii)$ follows from Assumption 3.1 and $|\widehat{V}_{h+1}(s)| \leq H$ for all $s \in \mathcal{S}$.

**Term (b).** We have

$$\widehat{\phi}_h(s,a)^\top \Lambda_h^{-1} \left\{ \sum_{\tau=1}^{N_{\text{off}}} \widehat{\phi}_h(s_h^\tau, a_h^\tau) \left[ (P_h^{(*,T+1)} \widehat{V}_{h+1})(s_h^\tau, a_h^\tau) - \widehat{V}_{h+1}(s_{h+1}^\tau) \right] \right\}$$

$$\leq \left\| \widehat{\phi}_h(s,a) \right\|_{\Lambda_h^{-1}} \left\| \sum_{\tau=1}^{N_{\text{off}}} \widehat{\phi}_h(s_h^\tau, a_h^\tau) \left[ (P_h^{(*,T+1)} \widehat{V}_{h+1})(s_h^\tau, a_h^\tau) - \widehat{V}_{h+1}(s_{h+1}^\tau) \right] \right\|_{\Lambda_h^{-1}}$$

$$\lesssim Hd\sqrt{\iota} \left\| \widehat{\phi}_h(s,a) \right\|_{\Lambda_h^{-1}} \tag{L.5}$$

where the first inequality comes from Cauchy Schwarz inequality and the last inequality follows from Lemma K.2.

**Term (c).** We have

$$\widehat{\phi}_h(s,a)^\top \Lambda_h^{-1} \left\{ \sum_{\tau=1}^{N_{\text{off}}} \widehat{\phi}_h(s_h^\tau, a_h^\tau) \left[ (\overline{P}_h \widehat{V}_{h+1} - P_h^{(*,T+1)} \widehat{V}_{h+1})(s_h^\tau, a_h^\tau) \right] \right\}$$

$$\leq \left| \widehat{\phi}_h(s,a)^\top \Lambda_h^{-1} \left( \sum_{\tau=1}^{N_{\text{off}}} \widehat{\phi}_h(s_h^\tau, a_h^\tau) \right) \right| \cdot H\xi_{\text{down}}$$

$$= \left| \sum_{\tau=1}^{N_{\text{off}}} \widehat{\phi}_h(s,a)^\top \Lambda_h^{-1} \widehat{\phi}_h(s_h^\tau, a_h^\tau) \right| \cdot H\xi_{\text{down}}$$

$$\leq \sqrt{\left( \sum_{\tau=1}^{N_{\text{off}}} \left\| \widehat{\phi}_h(s,a) \right\|_{\Lambda_h^{-1}}^2 \right) \left( \sum_{\tau=1}^{N_{\text{off}}} \left\| \widehat{\phi}_h(s_h^\tau, a_h^\tau) \right\|_{\Lambda_h^{-1}}^2 \right)} \cdot H\xi_{\text{down}}$$

$$\leq H\xi_{\text{down}} \sqrt{dN_{\text{off}}} \left\| \widehat{\phi}_h(s,a) \right\|_{\Lambda_h^{-1}}, \tag{L.6}$$

where the first inequality follows from $|\widehat{V}_{h+1}(s)| \leq H$ for all $s \in \mathcal{S}$ and from Lemma 4.3, the second inequality follows from Cauchy-Schwarz inequality and the last inequality follows from Lemma O.4.

Setting $\lambda_d = 1$, $\beta = c_\beta(Hd\sqrt{\iota} + H\sqrt{dN_{\text{off}}}\xi_{\text{down}})$, where $\iota = \log(HdN_{\text{off}} \max(\xi_{\text{down}}, 1)/\delta)$ and combining (L.2) to (L.6), we get for any $h \in [H]$ and for any $(s,a) \in \mathcal{S} \times \mathcal{A}$, with probability at least $1 - \delta$,

$$\left| (\mathbb{B}_h \widehat{V}_{h+1} - \widehat{\mathbb{B}}_h \widehat{V}_{h+1})(s,a) \right| \leq \beta \left\| \widehat{\phi}_h(s,a) \right\|_{\Lambda_h^{-1}} + H\xi_{\text{down}}.$$

This completes the proof. $\qquad \square$

## L.4 Proof of Lemma K.5

*Proof of Lemma K.5.* Recall that

$$\widehat{Q}_h(\cdot, \cdot) = \min\{\widehat{\phi}_h(\cdot, \cdot)^\top \widehat{w}_h - \Gamma_h(\cdot, \cdot), H - h + 1\}^+$$

If $\widehat{\phi}_h(s,a)^\top \widehat{w}_h - \Gamma_h(s,a) \leq 0$, $\widehat{Q}_h(s,a) = 0$. Then, we have $l_h(s,a) = (\mathbb{B}_h \widehat{V}_{h+1})(s,a) - \widehat{Q}_h(s,a) = (\mathbb{B}_h \widehat{V}_{h+1})(s,a) > 0$.

If $\widehat{\phi}_h(s,a)^\top \widehat{w}_h - \Gamma_h(s,a) > 0$, we have

$$l_h(s,a) = (\mathbb{B}_h \widehat{V}_{h+1})(s,a) - \widehat{Q}_h(s,a)$$

$$\geq (\mathbb{B}_h \widehat{V}_{h+1})(s,a) - \widehat{\phi}_h(s,a)^\top \widehat{w}_h + \Gamma_h(s,a)$$

$$= (\mathbb{B}_h \widehat{V}_{h+1})(s,a) - (\widehat{\mathbb{B}}_h \widehat{V}_{h+1})(s,a) + \Gamma_h(s,a)$$

$$\geq 0,$$

where the last inequality follows from conditioning on the event $\mathcal{E}(\delta)$. Thus, we have $l_h(s,a) \geq 0$ for any $h \in [H]$ and $(s,a) \in \mathcal{S} \times \mathcal{A}$.

We now show that $l_h(s,a) \leq 2\Gamma_h(s,a)$. Observe that

$$
\begin{aligned}
\widehat{\phi}_h(s,a)^\top \widehat{w}_h - \Gamma_h(s,a) &= (\widehat{\mathbb{B}}_h \widehat{V}_{h+1})(s,a) - \Gamma_h(s,a) \\
&\leq (\mathbb{B}_h \widehat{V}_{h+1})(s,a) \\
&\leq H - h + 1,
\end{aligned}
$$

where the first inequality follows from the conditioning on the event $\mathcal{E}(\delta)$ and the last inequality follows from the fact that $r_h \in [0,1]$ and $\widehat{V}_{h+1} \in [0, H-h]$.

Now,

$$
\begin{aligned}
\widehat{Q}_h(s,a) &= \min\{\widehat{\phi}_h(s,a)^\top \widehat{w}_h - \Gamma_h(s,a), H - h + 1\}^+ \\
&= \max\{\widehat{\phi}_h(s,a)^\top \widehat{w}_h - \Gamma_h(s,a), 0\} \\
&\geq \widehat{\phi}_h(s,a)^\top \widehat{w}_h - \Gamma_h(s,a) \\
&= (\widehat{\mathbb{B}}_h \widehat{V}_{h+1})(s,a) - \Gamma_h(s,a).
\end{aligned}
$$

Now, from the definition of $l_h$, we have

$$
\begin{aligned}
l_h(s,a) &= (\mathbb{B}_h \widehat{V}_{h+1})(s,a) - \widehat{Q}_h(s,a) \\
&\leq (\mathbb{B}_h \widehat{V}_{h+1})(s,a) - (\widehat{\mathbb{B}}_h \widehat{V}_{h+1})(s,a) + \Gamma_h(s,a) \\
&\leq 2\Gamma_h(s,a),
\end{aligned}
$$

where the last inequality follows from the conditioning on the event $\mathcal{E}(\delta)$.

Finally, we obtain

$$
\begin{aligned}
&V^{\pi^*}_{P^{(*,T+1)},r}(s) - V^{\widehat{\pi}}_{P^{(*,T+1)},r}(s) \\
&\leq -\sum_{h=1}^{H} \mathbb{E}_{\widehat{\pi}}[l_h(s_h,a_h)|s_1 = s] + \sum_{h=1}^{H} \mathbb{E}_{\pi^*}[l_h(s_h,a_h)|s_1 = s] \\
&\leq 2\sum_{h=1}^{H} \mathbb{E}_{\pi^*}[\Gamma_h(s_h,a_h)|s_1 = s]
\end{aligned}
$$

where the first inequality follows from Lemma O.3 and definition of $\widehat{\pi}$, and the last inequality follows because with probability at least $1 - \delta$, we have $0 \leq l_h(s,a) \leq 2\Gamma_h(s,a)$ for any $h \in [H]$ and $(s,a) \in \mathcal{S} \times \mathcal{A}$. $\qquad\square$

## M   Proof for Downstream Online RL

In this section for notational simplicity we denote $V^\pi_{h,P^{(*,T+1)},r^{T+1}}(s)$ and $Q^\pi_{h,P^{(*,T+1)},r^{T+1}}(s,a)$ by $V^\pi_h(s)$ and $Q^\pi_h(s,a)$ respectively.

First, we state the supporting lemmas that are used in the proof of Theorem 4.7. The proof of these lemmas are provided in Appendix N.

### M.1   Supporting Lemmas

The following concentration lemma for online RL upper-bounds the stochastic noise in regression. The proof is omitted since it is quite similar to the one of Lemma K.2.

**Lemma M.1.** *Setting* $\lambda_d = 1$, $\beta_n = c_\beta(Hd\sqrt{\iota_n(\delta)} + H\sqrt{dn}\xi_{down} + C_L\sqrt{Hd})$, *where* $\iota_n = \log(Hdn\max(\xi_{down},1)/\delta)$, *with probability at least* $1 - \delta/2$, *for all* $h \in [H]$ *and any* $n \in [N]$, *we have*

$$
\left\| \sum_{\tau=1}^{n-1} \widehat{\phi}_h(s_h^\tau, a_h^\tau) \left[ (P_h^{(*,T+1)} \widehat{V}_{h+1}^n)(s_h^\tau, a_h^\tau) - \widehat{V}_{h+1}^n(s_{h+1}^\tau) \right] \right\|_{\Lambda_h^{-1}} \lesssim Hd\sqrt{\iota_n}.
$$

The next lemma recursively bounds the difference between the value function maintained in Algorithm 5 (with-out bonus) and the true value function of any policy $\pi$. We provide a bound for this difference using their expected difference at next step, plus an error term. This error term is upper-bounded by the bonus term with high probability.

**Lemma M.2.** *There exists an absolute constant $c_\beta$ such that for $\beta_n = c_\beta(Hd\sqrt{\iota_n(\delta)}+H\sqrt{dn}\xi_{down}+C_L\sqrt{Hd})$, where $\iota_n = \log(Hdn\max(\xi_{down},1)/\delta)$, and for any policy $\pi$, with probability at least $1-\delta/2$, for any $(s,a) \in \mathcal{S} \times \mathcal{A}$, $n \in [N_{on}]$ and $h \in [H]$, we have*

$$\widehat{\phi}_h(s,a)^\top \widehat{w}_h^n - Q_h^\pi(s,a) = P_h^{(*,T+1)}(\widehat{V}_{h+1}^n - V_{h+1}^\pi)(s,a) + \Delta_h^n(s,a),$$

*for some $\Delta_h^n(s,a)$ that satisfies $\|\Delta_h^n(s,a)\| \leq \beta_n\|\widehat{\phi}_h(s,a)\|_{(\Lambda_h^n)^{-1}} + 2H\xi_{down}$.*

We now prove optimism of the estimated value function in the following lemma.

**Lemma M.3** (Optimism of value function). *With probability at least $1-\delta/2$, for any $(s,a) \in \mathcal{S} \times \mathcal{A}$, $h \in [H]$ and $n \in [N_{on}]$, we have*

$$\widehat{Q}_h^n(s,a) \geq Q_h^*(s,a) - 2H(H-h+1)\xi_{down}.$$

Lemma M.2 also easily transforms a recursive formula for the value function difference $\delta_h^n = \widehat{V}_h^n(s_h^n) - V_h^{\pi^n}(s_h^n)$. The following lemma will be useful in proving the regret bound for the online downstream task.

**Lemma M.4** (Recursive formula). *Let $\delta_h^n = \widehat{V}_h^n(s_h^n) - V_h^{\pi^n}(s_h^n)$ and $\xi_{h+1}^n = \mathbb{E}[\delta_{h+1}^n|s_h^n, a_h^n] - \delta_{h+1}^n$. Then for any $(n,h) \in [N_{on}, H]$ with probability at least $1-\delta/2$, we have*

$$\delta_h^n \leq \delta_{h+1}^n + \xi_{h+1}^n + \beta_n\|\widehat{\phi}_h(s_h^n, a_h^n)\|_{(\Lambda_h^n)^{-1}} + 2H\xi_{down}.$$

## M.2 Proof of Theorem 4.7

We are now ready to prove the main theorem in this section. We first restate Theorem 4.7.

**Theorem M.5.** *Let $\widetilde{\pi}$ be the uniform mixture of $\pi^1, \ldots, \pi^{N_{on}}$ in Algorithm 5. Under Assumption 4.1, setting $\lambda_d = 1$, $\beta_n = O(Hd\sqrt{\iota_n(\delta)} + H\sqrt{dn}\xi_{down} + C_L\sqrt{Hd})$, where $\iota_n = \log(Hdn\max(\xi_{down},1)/\delta)$, with probability $1-\delta$, the suboptimality gap of Algorithm 5 satisfies*

$$V_{P^{(*,T+1)},r}^* - V_{P^{(*,T+1)},r}^{\widetilde{\pi}} \leq \widetilde{O}(H^2d\xi_{down} + H^2d^{3/2}N_{on}^{-1/2}). \tag{M.1}$$

*Remark* M.6. As we use estimated representation $\{\widehat{\phi}_h\}_{h=1}^H$ from the upstream tasks, we get an extra term $H^2d\xi_{down}$ in the suboptimality gap above. When the linear misspecification error $\xi = \widetilde{O}(\sqrt{d/N_{off}})$ and the of trajectories $n$ in each upstream offline task dataset satisfies $n = \widetilde{O}\left(\frac{TN_{on}}{d}\right)$, the RHS of (M.1) is dominated by $\widetilde{O}(H^2d^{3/2}N_{on}^{-1/2})$ improving the suboptimality gap bound of REP-UCB (Uehara et al., 2022) [3] by an order of $\widetilde{O}(H^{3/2}K\sqrt{d})$ under low-rank MDP with unknown representation which attests to the benefit of upstream representation learning.

*Proof of Theorem M.5.* We first bound the cumulative regret by

$$\sum_{n=1}^{N_{on}} \left(V_{P^{(*,T+1)},r}^* - V_{P^{(*,T+1)},r}^{\pi^n}\right)$$

$$\overset{(i)}{\leq} \sum_{n=1}^{N_{on}} \left(V_1^n - V_{P^{(*,T+1)},r}^{\pi^n}\right) + 2H^2N_{on}\xi_{down}$$

$$\overset{(ii)}{\leq} \sum_{n=1}^{N_{on}}\sum_{h=1}^{H} \left[\xi_h^n + \beta_n\|\widehat{\phi}_h(s_h^n, a_h^n)\|_{(\Lambda_h^n)^{-1}} + 2H\xi_{down}\right] + 2H^2N_{on}\xi_{down}$$

$$\leq \underbrace{\sum_{n=1}^{N_{on}}\sum_{h=1}^{H} \xi_h^n}_{(a)} + \underbrace{\sum_{n=1}^{N_{on}}\sum_{h=1}^{H} \beta_n\|\widehat{\phi}_h(s_h^n, a_h^n)\|_{(\Lambda_h^n)^{-1}}}_{(b)} + 4H^2N_{on}\xi_{down} \tag{M.2}$$

---

[3]We convert the $1/(1-\gamma)$ horizon dependence in REP-UCB Uehara et al. (2022) to H. We further rescale their suboptimality gap by a factor of $H^2$ as we do not assume the sum of rewards to be within $[0,1]$.

where $(i)$ follows from Lemma M.3 and $(ii)$ follows from Lemma M.4.

Note that in term $(a)$, $\{\xi_h^n\}_{n=1,h=1}^{N_{\mathrm{on}},H}$ is a martingale difference sequence with $|\xi_h^n| \leq 2$. By Azuma-Hoeffding inequality, we have, with probability at least $1 - \delta/4$,

$$\left| \sum_{n=1}^{N_{\mathrm{on}}} \sum_{h=1}^{H} \xi_h^n \right| \leq \sqrt{8 N_{\mathrm{on}} H \log(8/\delta)}. \tag{M.3}$$

For term $(b)$, we have

$$\sum_{n=1}^{N_{\mathrm{on}}} \sum_{h=1}^{H} \beta_n \left\| \widehat{\phi}_h(s_h^n, a_h^n) \right\|_{(\Lambda_h^n)^{-1}}$$

$$\overset{(i)}{\leq} \sum_{h=1}^{H} \sqrt{\sum_{n=1}^{N_{\mathrm{on}}} \beta_n^2} \sqrt{\sum_{n=1}^{N_{\mathrm{on}}} \left\| \widehat{\phi}_h(s_h^n, a_h^n) \right\|_{(\Lambda_h^n)^{-1}}^2}$$

$$\overset{(ii)}{\lesssim} \sum_{h=1}^{H} \sqrt{2 c_\beta^2 (H^2 d^2 \iota_n N_{\mathrm{on}} + H^2 N_{\mathrm{on}}^2 d \xi_{\mathrm{down}}^2)} \sqrt{2d \log(1 + N_{\mathrm{on}}/(d\lambda))}$$

$$\leq H \sqrt{2 c_\beta^2 (H^2 d^2 \iota_n N_{\mathrm{on}} + H^2 N_{\mathrm{on}}^2 d \xi_{\mathrm{down}}^2)} \sqrt{4d \log N_{\mathrm{on}}}$$

$$\overset{(iii)}{\leq} 2\sqrt{2} c_\beta \left( H^2 \sqrt{d^3 \iota_n N_{\mathrm{on}} \log N_{\mathrm{on}}} + H^2 d N_{\mathrm{on}} \xi_{\mathrm{down}} \sqrt{\log N_{\mathrm{on}}} \right), \tag{M.4}$$

where $(i)$ follows from Cauchy-Schwarz inequality, $(ii)$ follows from Lemma O.6, and $(iii)$ follows from the inequality that $\sqrt{x+y} \leq \sqrt{x} + \sqrt{y}$ for all $x, y \geq 0$.

Combining (M.2), (M.3) and (M.4) we get

$$\sum_{n=1}^{N_{\mathrm{on}}} \left( V_{P^{(*,T+1)},r}^* - V_{P^{(*,T+1)},r}^{\pi^n} \right) \lesssim \widetilde{O}(H^2 d N_{\mathrm{on}} \xi_{\mathrm{down}} + H^2 \sqrt{d^3 N_{\mathrm{on}}}).$$

Dividing both sides by $N_{\mathrm{on}}$, we get

$$V_{P^{(*,T+1)},r}^* - V_{P^{(*,T+1)},r}^{\widetilde{\pi}} \leq \widetilde{O}(H^2 d \xi_{\mathrm{down}} + H^2 d^{3/2} N_{\mathrm{on}}^{-1/2}).$$

This completes the proof. $\qquad\qquad\square$

# N  Proof of Supporting Lemmas in Appendix M

In this section, we provide the proofs of the lemmas that we used in the proof of Theorem 4.7.

## N.1  Proof of Lemma M.2

*Proof of Lemma M.2.* For any policy $\pi$, we define $w_h^\pi = \int V_{h+1}^\pi(s') \widehat{\mu}^*(s') ds'$. Note that $\widehat{\phi}_h(s, a)^\top w_h^\pi = \overline{P}_h V_{h+1}^\pi(s, a)$ and by Lemma 4.3, we have $\|w_h^\pi\| \leq C_L \sqrt{d}$.

Now, we derive the following

$$\widehat{\phi}_h(s, a)^\top \widehat{w}_h^n - Q_h^\pi(s, a)$$
$$= \widehat{\phi}_h(s, a)^\top \widehat{w}_h^n - \widehat{\phi}_h(s, a)^\top w_h^\pi + \widehat{\phi}_h(s, a)^\top w_h^\pi - Q_h^\pi(s, a)$$
$$\leq \left( \widehat{\phi}_h(s, a)^\top \widehat{w}_h^n - \widehat{\phi}_h(s, a)^\top w_h^\pi \right) + \left| \widehat{\phi}_h(s, a)^\top w_h^\pi - Q_h^\pi(s, a) \right| \tag{N.1}$$

Using Lemma 4.3, we bound the second term as

$$\left| Q_h^\pi(s, a) - \widehat{\phi}_h(s, a)^\top w_h^\pi \right| = \left| P_h^{(*,T+1)} V_{h+1}^\pi(s, a) - \overline{P}_h V_{h+1}^\pi(s, a) \right|$$
$$\leq H \xi_{\mathrm{down}}, \tag{N.2}$$

where we used the observation $|V_{h+1}^\pi| \le H$.

Now, the first term in (N.1) can be bounded by

$$\widehat{\phi}_h(s,a)^\top \widehat{w}_h^n - \widehat{\phi}_h(s,a)^\top w_h^\pi$$

$$= \widehat{\phi}_h(s,a)^\top (\Lambda_h^n)^{-1} \sum_{\tau=1}^{n-1} \widehat{\phi}_h(s_h^\tau, a_h^\tau) \widehat{V}_{h+1}^n(s_{h+1}^\tau) - \widehat{\phi}_h(s,a)^\top w_h^\pi$$

$$= \widehat{\phi}_h(s,a)^\top (\Lambda_h^n)^{-1} \left\{ \sum_{\tau=1}^{n-1} \widehat{\phi}_h(s_h^\tau, a_h^\tau) \widehat{V}_{h+1}^n(s_{h+1}^\tau) - \lambda_d w_h^\pi - \sum_{\tau=1}^{n-1} \widehat{\phi}_h(s_h^\tau, a_h^\tau) \overline{P}_h V_{h+1}^\pi \right\}$$

$$= \underbrace{-\lambda_d \widehat{\phi}_h(s,a)^\top (\Lambda_h^n)^{-1} w_h^\pi}_{(a)} + \underbrace{\widehat{\phi}_h(s,a)^\top (\Lambda_h^n)^{-1} \left\{ \sum_{\tau=1}^{n-1} \widehat{\phi}_h(s_h^\tau, a_h^\tau) \left[ \widehat{V}_{h+1}^n(s_{h+1}^\tau) - P_h^{(*,T+1)} \widehat{V}_{h+1}^n(s_{h+1}^\tau) \right] \right\}}_{(b)}$$

$$+ \underbrace{\widehat{\phi}_h(s,a)^\top (\Lambda_h^n)^{-1} \left\{ \sum_{\tau=1}^{n-1} \widehat{\phi}_h(s_h^\tau, a_h^\tau) \overline{P}_h (\widehat{V}_{h+1}^n - V_{h+1}^\pi)(s_h^\tau, a_h^\tau) \right\}}_{(c)}$$

$$+ \underbrace{\widehat{\phi}_h(s,a)^\top (\Lambda_h^n)^{-1} \left\{ \sum_{\tau=1}^{n-1} \widehat{\phi}_h(s_h^\tau, a_h^\tau) \left( P_h^{(*,T+1)} - \overline{P}_h \right) \widehat{V}_{h+1}^n(s_{h+1}^\tau) \right\}}_{(d)} \tag{N.3}$$

We now bound $(a), (b), (c), (d)$ in (N.3) individually.

**Term (a).** We have,

$$\left| -\lambda_d \widehat{\phi}_h(s,a)^\top (\Lambda_h^n)^{-1} w_h^\pi \right| \le \left\| \widehat{\phi}_h(s,a) \right\|_{(\Lambda_h^n)^{-1}} \left\| \lambda_d w_h^\pi \right\|_{(\Lambda_h^n)^{-1}}$$

$$\le \sqrt{\lambda_d} \| w_h^\pi \|_2 \left\| \widehat{\phi}_h(s,a) \right\|_{(\Lambda_h^n)^{-1}}$$

$$\le C_L \sqrt{\lambda_d H d} \left\| \widehat{\phi}_h(s,a) \right\|_{(\Lambda_h^n)^{-1}}, \tag{N.4}$$

where the first inequality follows from Cauchy-Schwarz inequality, the second inequality follows from the fact that the largest eigenvalue of $(\Lambda_h^n)^{-1}$ is at most $1/\lambda_d$ and the last inequality follows from Assumption 3.1 and $|V_{h+1}^\pi(s)| \le H$ for all $s \in \mathcal{S}$.

**Term (b).** Using Cauchy-Schwarz inequality and Lemma M.1, we have

$$\widehat{\phi}_h(s,a)^\top (\Lambda_h^n)^{-1} \left\{ \sum_{\tau=1}^{n-1} \widehat{\phi}_h(s_h^\tau, a_h^\tau) \left[ \widehat{V}_{h+1}^n(s_{h+1}^\tau) - P_h^{(*,T+1)} \widehat{V}_{h+1}^n(s_{h+1}^\tau) \right] \right\}$$

$$\le \left\| \widehat{\phi}_h(s,a) \right\|_{(\Lambda_h^n)^{-1}} \left\| \sum_{\tau=1}^{n-1} \widehat{\phi}_h(s_h^\tau, a_h^\tau) \left[ \widehat{V}_{h+1}^n(s_{h+1}^\tau) - P_h^{(*,T+1)} \widehat{V}_{h+1}^n(s_{h+1}^\tau) \right] \right\|_{(\Lambda_h^n)^{-1}}$$

$$\lesssim H d \sqrt{\iota_n} \left\| \widehat{\phi}_h(s,a) \right\|_{(\Lambda_h^n)^{-1}}. \tag{N.5}$$

**Term (c).** We have

$$\widehat{\phi}_h(s,a)^\top (\Lambda_h^n)^{-1}\bigg\{ \sum_{\tau=1}^{n-1} \widehat{\phi}_h(s_h^\tau, a_h^\tau)\overline{P}_h(\widehat{V}_{h+1}^n - V_{h+1}^\pi)(s_h^\tau, a_h^\tau)\bigg\}$$

$$\leq \bigg|\widehat{\phi}_h(s,a)^\top (\Lambda_h^n)^{-1}\bigg\{ \sum_{\tau=1}^{n-1} \widehat{\phi}_h(s_h^\tau, a_h^\tau)\widehat{\phi}_h(s_h^\tau, a_h^\tau)^\top \int (\widehat{V}_{h+1}^n - V_{h+1}^\pi)(s')\widehat{\mu}_h^*(s')ds'\bigg\}\bigg|$$

$$= \bigg|\widehat{\phi}_h(s,a)^\top (\Lambda_h^n)^{-1}(\Lambda_h^n - \lambda_d I) \int (\widehat{V}_{h+1}^n - V_{h+1}^\pi)(s')\widehat{\mu}_h^*(s')ds'\bigg|$$

$$\leq \bigg|\widehat{\phi}_h(s,a)^\top \int (\widehat{V}_{h+1}^n - V_{h+1}^\pi)(s')\widehat{\mu}_h^*(s')ds'\bigg| + \bigg|\lambda_d\widehat{\phi}_h(s,a)^\top (\Lambda_h^n)^{-1} \int (\widehat{V}_{h+1}^n - V_{h+1}^\pi)(s')\widehat{\mu}_h^*(s')ds'\bigg|$$

$$\overset{(i)}{\leq} \big|\overline{P}_h(\widehat{V}_{h+1}^n - V_{h+1}^\pi)(s,a)\big| + C_L\sqrt{\lambda_d H d}\big\|\widehat{\phi}_h(s,a)\big\|_{(\Lambda_h^n)^{-1}}$$

$$\overset{(ii)}{\leq} P_h^{(*,T+1)}(\widehat{V}_{h+1}^n - V_{h+1}^\pi)(s,a) + H\xi_{\text{down}} + C_L\sqrt{\lambda_d H d}\big\|\widehat{\phi}_h(s,a)\big\|_{(\Lambda_h^n)^{-1}}, \tag{N.6}$$

where $(i)$ follows from similar steps as in (N.4) and $(ii)$ follows from Lemma 4.3.

**Term (d).** We have

$$\widehat{\phi}_h(s,a)^\top (\Lambda_h^n)^{-1}\bigg\{ \sum_{\tau=1}^{n-1} \widehat{\phi}_h(s_h^\tau, a_h^\tau)\Big(P_h^{(*,T+1)} - \overline{P}_h\Big)\widehat{V}_{h+1}^n(s_{h+1}^\tau)\bigg\}$$

$$\overset{(i)}{\leq} \bigg|\widehat{\phi}_h(s,a)^\top (\Lambda_h^n)^{-1}\bigg\{ \sum_{\tau=1}^{n-1} \widehat{\phi}_h(s_h^\tau, a_h^\tau)\bigg\}\bigg| \cdot H\xi_{\text{down}}$$

$$= \sum_{\tau=1}^{n-1} \big|\widehat{\phi}_h(s,a)^\top (\Lambda_h^n)^{-1}\widehat{\phi}_h(s_h^\tau, a_h^\tau)\big| \cdot H\xi_{\text{down}}$$

$$\overset{(ii)}{\leq} \sqrt{\bigg(\sum_{\tau=1}^{n-1} \big\|\widehat{\phi}_h(s,a)\big\|_{(\Lambda_h^n)^{-1}}^2\bigg)\bigg(\sum_{\tau=1}^{n-1} \big\|\widehat{\phi}_h(s_h^\tau, a_h^\tau)\big\|_{(\Lambda_h^n)^{-1}}^2\bigg)} \cdot H\xi_{\text{down}}$$

$$\overset{(iii)}{\leq} H\xi_{\text{down}}\sqrt{dn}\big\|\widehat{\phi}_h(s,a)\big\|_{(\Lambda_h^n)^{-1}}, \tag{N.7}$$

where $(i)$ follows from Lemma 4.3 and $|\widehat{V}_{h+1}^n(s)| \leq H$ for all $s \in \mathcal{S}$, $(ii)$ follows from Cauchy-Schwarz inequality and $(iii)$ follows from Lemma O.4.

Substituting (N.4), (N.5), (N.6), (N.7) into (N.3), and setting $\lambda_d = 1$ we get

$$\widehat{\phi}_h(s,a)^\top \widehat{w}_h^n - \widehat{\phi}_h(s,a)^\top w_h^\pi$$

$$\lesssim c_\beta\big(Hd\sqrt{\iota_n(\delta)} + H\sqrt{dn}\xi_{\text{down}} + C_L\sqrt{Hd}\big)\big\|\widehat{\phi}_h(s,a)\big\|_{(\Lambda_h^n)^{-1}}$$

$$\quad + P_h^{(*,T+1)}(\widehat{V}_{h+1}^n - V_{h+1}^\pi)(s,a) + H\xi_{\text{down}}$$

$$= \beta_n\big\|\widehat{\phi}_h(s,a)\big\|_{(\Lambda_h^n)^{-1}} + P_h^{(*,T+1)}(\widehat{V}_{h+1}^n - V_{h+1}^\pi)(s,a) + H\xi_{\text{down}}. \tag{N.8}$$

Combining (N.8), (N.2) in (N.1) completes the proof. □

## N.2 Proof of Lemma M.3

*Proof of Lemma M.3.* We use backward induction for our proof. First, we prove the base case, at the last step $H$. By Lemma M.2, we have

$$\big|\widehat{\phi}_H(s,a)\widehat{w}_H^n - Q_H^\pi(s,a)\big| = \big|P_H^{(*,T+1)}(\widehat{V}_{H+1}^n - V_{H+1}^\pi)(s,a) + \Delta_H^n(s,a)\big|$$

$$\leq \beta_n\big\|\widehat{\phi}_H(s,a)\big\|_{(\Lambda_H^n)^{-1}} + 2H\xi_{\text{down}}.$$

Thus, we have

$$\widehat{Q}_H^n(s,a) = \min\left\{\widehat{\phi}_H(s,a)^\top \widehat{w}_H^n + \beta_n \big\|\widehat{\phi}_H(s,a)\big\|_{(\Lambda_H^n)^{-1}}, H - h + 1\right\}$$
$$\geq Q_H^*(s,a) - 2H\xi_{\mathrm{down}}.$$

Now, suppose the statement holds true at step $h+1$ and consider the step $h$. Using Lemma M.2, we have

$$\widehat{\phi}_h(s,a)^\top \widehat{w}_h^n - Q_h^*(s,a) = \Delta_h^n(s,a) + P_h^{(*,T+1)}(\widehat{V}_{h+1}^n - V_{h+1}^*)(s,a)$$
$$\geq -\beta_n\big\|\widehat{\phi}_h(s,a)\big\|_{(\Lambda_h^n)^{-1}} - 2H\xi_{\mathrm{down}} - 2H(H-h)\xi_{\mathrm{down}}$$
$$= -\beta_n\big\|\widehat{\phi}_h(s,a)\big\|_{(\Lambda_h^n)^{-1}} - 2H(H-h+1)\xi_{\mathrm{down}}.$$

Therefore,

$$\widehat{Q}_h^n(s,a) = \min\{\widehat{\phi}_h(s,a)^\top \widehat{w}_h^n + \beta_n\|\widehat{\phi}(s,a)\|_{(\Lambda_h^n)^{-1}}, H-h+1\}^+$$
$$\geq Q_h^*(s,a) - 2H(H-h+1)\xi_{\mathrm{down}},$$

which completes the proof. $\qquad\square$

### N.3   Proof of Lemma M.4

*Proof of Lemma M.4.* By Lemma M.2, for any $(s,a) \in \mathcal{S} \times \mathcal{A}$, $h \in [H]$ and $n \in [N_{\mathrm{on}}]$, with probability at least $1 - \delta/2$, we have

$$\widehat{Q}_h^n(s,a) - Q_h^{\pi^n}(s,a) \leq \Delta_h^n(s,a) + P_h^{(*,T+1)}(\widehat{V}_{h+1}^n - V_{h+1}^{\pi^n})(s,a)$$
$$\leq \beta_n\big\|\widehat{\phi}_h(s,a)\big\|_{(\Lambda_h^n)^{-1}} + 2H\xi_{\mathrm{down}} + P_h^{(*,T+1)}(\widehat{V}_{h+1}^n - V_{h+1}^{\pi^n})(s,a).$$

And finally, by definition of $\pi^n$ in Algorithm 5, we have $\pi^n(s_h^n) = a_h^n = \mathrm{argmax}_{a \in \mathcal{A}}\widehat{Q}_h^n(s_h,a)$. This implies $\widehat{Q}_h^n(s_h^n,a_h^n) - Q_h^{\pi^n}(s_h^n,a_h^n) = \widehat{V}_h^n(s_h^n) - V_h^{\pi^n}(s_h^n) = \delta_h^n$. Thus, we have

$$\delta_h^n \leq \delta_{h+1}^n + \xi_{h+1}^n + \beta_n\big\|\widehat{\phi}_h(s,a)\big\|_{(\Lambda_h^n)^{-1}} + 2H\xi_{\mathrm{down}}.$$

$$\square$$

## O   Auxiliary Lemmas

### O.1   Miscellaneous Lemmas

The following lemma measures the difference between two value functions under two MDPs and reward functions. Here, we use a shorthand notation $P_h V_{h+1}(s_h,a_h) = \mathbb{E}_{s \sim P_h(\cdot|s_h,a_h)}[V(s)]$.

**Lemma O.1** (Simulation lemma (Dann et al., 2017))**.** *Consider two MDPs with transition kernels $P_1$ and $P_2$, and reward function $r_1$ and $r_2$ respectively. Given a policy $\pi$, we have,*

$$V_{h,P_1,r_1}^\pi(s_h) - V_{h,P_2,r_2}^\pi(s_h) = \sum_{h'=h}^{H} \mathbb{E}_{\substack{s_{h'} \sim (P_2,\pi) \\ a_{h'} \sim \pi}} \big[r_1(s_{h'},a_{h'}) - r_2(s_{h'},a_{h'})$$
$$+ (P_{1,h'} - P_{2,h'})V_{h'+1,P_1,r_1}^\pi(s_{h'},a_{h'}) \,|\, s_h\big]$$
$$= \sum_{h'=h}^{H} \mathbb{E}_{\substack{s_{h'} \sim (P_1,\pi) \\ a_{h'} \sim \pi}} \big[r_1(s_{h'},a_{h'}) - r_2(s_{h'},a_{h'})$$
$$+ (P_{1,h'} - P_{2,h'})V_{h'+1,P_2,r_2}^\pi(s_{h'},a_{h'}) \,|\, s_h\big]$$

We use the following lemma to deal with distribution shift in offline RL setting.

**Lemma O.2** (Distribution shift lemma ([Chang et al., 2021](#))). *Consider two distributions $\rho_1 \in \Delta(\mathcal{S} \times \mathcal{A})$ and $\rho_2 \in \Delta(\mathcal{S} \times \mathcal{A})$, and a feature mapping $\phi : \mathcal{S} \times \mathcal{A} \to \mathbb{R}^d$. Denote $C := \sup_{x \in \mathbb{R}^d} \frac{x^\top \mathbb{E}_{s,a \sim \rho_1} \phi(s,a) \phi(s,a)^\top x}{x^\top \mathbb{E}_{s,a \sim \rho_2} \phi(s,a) \phi(s,a)^\top x}$. Then for any positive definite matrix $\Lambda$, we have*

$$\mathbb{E}_{s,a \sim \rho_1} \phi(s,a)^\top \Lambda \phi(s,a) \leq C \mathbb{E}_{s,a \sim \rho_2} \phi(s,a)^\top \Lambda \phi(s,a).$$

We use the following lemma to bound the suboptimality in downstream offline RL.

**Lemma O.3** (Decomposition of Suboptimality, Lemma 3.1 in ([Jin et al., 2021](#))). *Let $\widehat{\pi} = \{\widehat{\pi}_h\}_{h=1}^H$ be the policy such that $\widehat{V}_h(s) = \langle \widehat{Q}_h(s,\cdot), \widehat{\pi}_h(\cdot|s) \rangle_{\mathcal{A}}$ and for each step $h \in [H]$, define the model evaluation error as $l_h(s,a) = (\mathbb{B}_h \widehat{V}_{h+1})(s,a) - \widehat{Q}_h(s,a)$. For any $\widehat{\pi}$ and $s \in \mathcal{S}$, we have*

$$V_1^{\pi^*}(s) - V_1^{\widehat{\pi}}(s) = -\sum_{h=1}^H \mathbb{E}_{\widehat{\pi}}[l_h(s_h, a_h)|s_1 = s] + \sum_{h=1}^H \mathbb{E}_{\pi^*}[l_h(s_h, a_h)|s_1 = s]$$

$$+ \sum_{h=1}^H \mathbb{E}_{\pi^*}[\langle \widehat{Q}_h(s_h, \cdot), \pi_h^*(\cdot|s_h) - \widehat{\pi}_h(\cdot|s_h) \rangle_{\mathcal{A}}|s_1 = s].$$

## O.2 Inequalities for summations

**Lemma O.4** (Lemma D.1 in [Jin et al. (2020b)](#)). *Let $\Lambda_h = \lambda I + \sum_{i=1}^t \phi_i \phi_i^\top$, where $\phi_i \in \mathbb{R}^d$ and $\lambda > 0$. Then it holds that*

$$\sum_{i=1}^t \phi_i^\top (\Lambda_h)^{-1} \phi_i \leq d.$$

**Lemma O.5** (Lemma D.5 in ([Ishfaq et al., 2021](#))). *Let $A \in \mathbb{R}^{d \times d}$ be a positive definite matrix where its largest eigenvalue $\lambda_{\max}(A) \leq \lambda$. Let $x_1, \ldots, x_k$ be $k$ vectors in $\mathbb{R}^d$. Then it holds that*

$$\left\| A \sum_{i=1}^k x_i \right\| \leq \sqrt{\lambda k} \left( \sum_{i=1}^k \|x_i\|_A^2 \right)^{1/2}.$$

**Lemma O.6** (Lemma G.2 in ([Agarwal et al., 2020b](#))). *Consider a sequence of semidefinite matrices $X_1, \ldots, X_N \in \mathbb{R}^{d \times d}$ with $\mathrm{Tr}(X_n) \leq 1$ for all $n \in [N]$. Define $M_0 = \lambda I$ and $M_n = M_{n-1} + X_n$. Then*

$$\sum_{n=1}^N \mathrm{Tr}(X_n M_{n-1}^{-1}) \leq 2 \log \frac{\det(M_N)}{\det(M_0)} \leq 2d \log(1 + N/(\lambda d)).$$

## O.3 Covering numbers and self-normalized processes

**Lemma O.7** (Lemma D.4 in [Jin et al. (2020b)](#)). *Let $\{s_i\}_{i=1}^\infty$ be a stochastic process on state space $\mathcal{S}$ with corresponding filtration $\{\mathcal{F}_i\}_{i=1}^\infty$. Let $\{\phi_i\}_{i=1}^\infty$ be an $\mathbb{R}^d$-valued stochastic process where $\phi_i \in \mathcal{F}_{i-1}$, and $\|\phi_i\| \leq 1$. Let $\Lambda_k = \lambda I + \sum_{i=1}^k \phi_i \phi_i^\top$. Then for any $\delta > 0$, with probability at least $1 - \delta$, for all $k \geq 0$, and any $V \in \mathcal{V}$ with $\sup_{s \in \mathcal{S}} |V(s)| \leq H$, we have*

$$\left\| \sum_{i=1}^k \phi_i \{ V(s_i) - \mathbb{E}[V(s_i) \,|\, \mathcal{F}_{i-1}] \} \right\|_{\Lambda_k^{-1}}^2 \leq 4H^2 \left[ \frac{d}{2} \log\left( \frac{k+\lambda}{\lambda} \right) + \log \frac{\mathcal{N}_\varepsilon}{\delta} \right] + \frac{8k^2 \epsilon^2}{\lambda},$$

*where $\mathcal{N}_\varepsilon$ is the $\varepsilon$-covering number of $\mathcal{V}$ with respect to the distance $\mathrm{dist}(V, V') = \sup_{s \in \mathcal{S}} |V(s) - V'(s)|$.*

**Lemma O.8** (Covering number of Euclidean ball, [Vershynin (2018)](#)). *For any $\varepsilon > 0$, the $\varepsilon$-covering number, $\mathcal{N}_\varepsilon$, of the Euclidean ball of radius $B > 0$ in $\mathbb{R}^d$ satisfies*

$$\mathcal{N}_\varepsilon \leq \left( 1 + \frac{2B}{\varepsilon} \right)^d \leq \left( \frac{3B}{\varepsilon} \right)^d.$$

**Lemma O.9** ($\varepsilon$-covering number (Jin et al., 2020b)). *Let $\mathcal{V}$ denote a class of functions mapping from $\mathcal{S}$ to $\mathbb{R}$ with the following parametric form*

$$V(\cdot) = \min\left\{ \max_{a \in \mathcal{A}} w^\top \phi(\cdot, a) + \beta\sqrt{\phi(\cdot, a)^\top \Lambda^{-1} \phi(\cdot, a)}, H \right\},$$

*where the parameters $(w, \beta, \Lambda)$ satisfy $\|w\| \le L$, $\beta \in [0, B]$ and the minimum eigenvalue satisfies $\lambda_{\min}(\Lambda) \ge \lambda$. Assume $\|\phi(s, a)\| \le 1$ for all $(s, a)$ pairs, and let $\mathcal{N}_\varepsilon$ be the $\varepsilon$-covering number of $\mathcal{V}$ with respect to the distance $dist(V, V') = \sup_s |V(s) - V'(s)|$. Then,*

$$\log \mathcal{N}_\varepsilon \le d \log(1 + 4L/\varepsilon) + d^2 \log(1 + 8d^{1/2} B^2 / (\lambda \varepsilon^2)).$$

