# OpenReview forum: "Offline Multitask Representation Learning for Reinforcement Learning"
_NeurIPS.cc/2024/Conference — NeurIPS 2024 poster_

### Official Review · Reviewer_oHwh · 2024-07-02

**Soundness:** 3
**Presentation:** 2
**Contribution:** 2
**Rating:** 5
**Confidence:** 2

**Summary:**

This paper investigates how representation learning in offline multitask low-rank RL can improve sample complexity when using the learned representations in downstream reward-free  RL, offline and online RL settings.
The paper assumes that the new task shares the same representation as the upstream tasks making up the offline data set.
The paper provides theoretical results but does not provide any empirical evaluation.
Rather intuitively, their result shows that learning a representation from the offline multitask data set can improve sample complexity in new downstream tasks when compared to learning the new downstream task from scratch.

**Strengths:**

Representation learning and large-scale offline data sets have shown much promise for learning policies.
As such, investigating the theory of learning representations from multi-task offline data seems an important research direction.
I found the related work section very beneficial for pulling different pieces of the paper together. This really helped me understand things.

**Weaknesses:**

I should start off by saying that I have little experience with the theoretical aspects of representation learning for RL and there were many aspects of this paper I had to take as a given without fully understanding.
I have put a confidence of 2 to reflect his.

Whilst this paper addresses an important and interesting problem, I did find it hard to follow.
This is likely due to the fact I do not research theoretical aspects of RL.
Nevertheless, I do think the paper should be digestible to a wider audience than it currently is.
In particular, the authors could do a better job of communicating intuition for the different equations that are introduced.
I will now detail some suggestions with the aim of helping the authors communicate their work more clearly.

At the end of Section 3, it was not clear to me what the main result is. I would advise the authors to summarise the main result at the end of the section.
This would really help make this paper an easier read.
Similarly in Section 4.
What is the main takeaway from this section?

Equation 3.4 is introduced rather abruptly. That is, I did not understand its purpose until later in the paper when the authors stated that they wanted to improve sample complexity by improving exploration.
I would advise the authors to state this reasoning when they introduce Equation 3.4.

I am confused by the notation for $h \in [H]$, $t \in [T]$ and $i \in [N]$. Aren't these discrete numbers? If so, the authors should use curly braces instead, e.g. $h \in \\{0,\ldots,H\\}$


## Minor comments and typos
- Line 61 - $H$ and $d$ aren't defined yet so this doesn't make much sense. I found the contributions had way too many technical details which does not make any sense to the reader given what they've read so far. Consider shortening these and removing unnecessary technical details.
- Equation 2.1 - What is $\mathbb{P}$? This is not defined until way later in the paper.
- Line 42 - "We develop new" should be "We develop a new"
- Line 69 - MDPs are discrete time by definition so it is confusing to state you consider discrete-time MDPs. It almost implied MDPs are not usually discrete time.
- Sections shouldn't lead straight into sub-sections, e.g. Section 2 to 2.1 and Section 4 to 4.1
    - You should add a few sentences explaining what you are introducing in this section.
- Line 137 - "was" should be "were"
- Line 152 - "we construct lower confidence" should be "we construct a lower confidence"
- Line 159 - "can" should be "has"
- Line 177 - "compared to concentrability coefficient" should be "compared to the concentrability coefficient"
- Line 215 - "sequel" is the wrong word here. It sounds like you mean the next paper...
- Line 233 - "task" should be "tasks"
- Line 233 - "assume" should be "assumes"
- Line 233 - "for reward-free" should be "for the reward-free"
- Table 1 - It is not clear from the caption that this is for the reward-free setting. This should be made clear in the caption.
- Line 299 - "such" should be "this"
- Line 343 - "study the multitask RL" should be "study multitask RL"
- Line 347 - "setting" should be "settings"

**Questions:**

- Why is $\phi$ shared and $\mu$ is not? This should be made clear in the text.
- What is $\lambda$ in Equation 3.3?
- Why are equations not referenced as Equation X but instead as (X)? You could be referring to an equation/section etc...

**Limitations:**

I do not think the limitations are adequately discussed.
The proofs utilise a lot of assumptions and the paper would benefit from a clear paragraph demonstrating the limitations associated with these assumptions. What are the practical implications of these assumptions?

---

> ### Author Rebuttal · Authors · 2024-08-07
>
> We thank the reviewer for their valuable time and effort in providing detailed feedback on our work. We hope our response will fully address all of the reviewer's points.
>
> Thank you so much for your suggestion on how to improve the readability for a broader audience. We will definitely add more intuition for each of the different equations that are presented in the main paper.
>
> ### Main take-away for Section 3
> The main take-away for the main result (Theorem 3.3) in Section 3 is highlighted in L187-193. Namely, in Theorem 3.3, Equation (3.6) provides the statistical scaling rate for the average accuracy of the estimated transition kernels of the $T$ MDPs. Equation (3.7) implies that  if the optimal policy $\pi_t^*$ is covered by the offline data for all $t \in$ {1, 2, ..., T} as characterized by multitask relative condition number $C^*$, then the output policy of MORL is able to compete against it on average as well.
>
> ### Main take-away for Section 4
> On high-level the main take-away for section 4 is that representation learning in upstream multitask offline dataset can help in improving the sample complexity, statistical rate of sub-optimality gap and regret bound in reward-free, offline and online RL respectively.
>
> The main take-away for Theorem 4.4 is provided in Remark 4.5. Due to space constraint in the main paper, the main take-away for Theorem 4.7 is provided in Remark K.7 (L1042-L1046) in Appendix K. Similarly, the main take-away for Theorem 4.8 is provided in Remark M.6 (L1140-L1145) in Appendix M.
>
> ### Other comments
> Thanks for suggesting to put the purpose of introducing Equation 3.4 earlier. We will do it in our next version. Essentially, Equation 3.4 is used for inducing pessimism in the learned policy in Algorithm 1. The notation $[N]$ indicates the set {1, 2, ..., N}. Even though it’s a standard notation, we will clarify it in our next version. Thanks for the other minor comments and pointing out some typos. We will correct and update them.
>
> ### Questions
> 1. By assuming $\mu$ is not shared across tasks, we ensure that the transition kernel of each task is different from one another. However, for theoretical analysis it is imperative to make an assumption that will connect all the MDP tasks in some manner. This is why we assume that the feature $\phi$ is shared across tasks.
>
> 2. Here $\lambda$ is a regularization factor in Equation 3.3.
>
> 3. Thanks for your suggestion regarding equations as Equation X instead of as (X). We will update this in our next version as per your suggestion.
>
> ### Limitations
>
> We have already discussed the assumptions made at length in L230-L241. We will add further discussion on how these assumptions can be limiting.
>
> We hope we have addressed all of your questions/concerns. If you have any further questions, we would be more than happy to answer them and if you don’t, would you kindly consider increasing your score?

---

> > ### Comment · Reviewer_oHwh · 2024-08-09
> >
> > Thank you for answering my questions. After reading the rebuttal, the other reviews, and the other rebuttal comments, I have decided to maintain my score of 5. There seems to be broad agreement amongst reviewers that this paper lacks intuition and is hard to follow. I suggest the authors take on board the reviews and rework their paper. With that said, I do think this paper has potential and I am happy for it to be published if all other reviewers believe it should. However, in its current form, I do not feel like I can give a score higher than 5.
> >
> > Thank you for explaining where the limitations are discussed. As both Reviewer **Qe4K** and myself overlooked these limitations, I suggest you help the reader find them, for example, name the paragraph with \paragraph{Limitations}. I'd also note that it's pretty common practice to discuss the limitations of your work in the discussion or conclusion with a clearly labelled paragraph.

---

> > > ### Author Response · Authors · 2024-08-11
> > >
> > > We thank the reviewer for their suggestions on how to make the discussed limitation more easily findable. We will incorporate these suggestions in our next version.
> > >
> > > Once again thank you so much for your time in reviewing the paper and providing meaningful suggestions on how to improve the paper.

---

### Official Review · Reviewer_fKn9 · 2024-07-05

**Soundness:** 3
**Presentation:** 3
**Contribution:** 3
**Rating:** 6
**Confidence:** 3

**Summary:**

This paper introduces the Multitask Offline Representation Learning (MORL) algorithm, which aims to enhance sample efficiency in offline multitask reinforcement learning (RL). By learning a shared representation from pre-collected datasets of different tasks, modeled by low-rank Markov Decision Processes (MDPs), the authors demonstrate the theoretical benefits of this method for various downstream RL tasks, including reward-free, offline, and online scenarios. The paper provides extensive theoretical analysis and introduces specific data-coverage assumptions, presenting the first theoretical results showing the advantages of representation learning in offline multitask RL.

**Strengths:**

1. The paper addresses the novel problem of offline multitask representation learning in RL with a new algorithm (MORL) that incorporates innovative techniques such as joint model learning via MLE and penalty functions for pessimism. The approach is well-grounded in theory and fills a relevant gap in the literature.
2. The submission is technically sound with rigorous theoretical analysis supporting the proposed algorithm. Detailed proofs of the main theoretical results and thorough explanations of the algorithm's components are provided.
3. The paper is clearly written and well-organized, with a logical progression from problem introduction to algorithm presentation and theoretical results. The definitions and assumptions are clearly stated, and the proofs are detailed and comprehensible.
4. The results are important as they provide the first theoretical demonstration of the benefits of multitask representation learning in offline RL. The approach has potential applications in domains where collecting data online is infeasible, making it highly relevant for real-world problems.

**Weaknesses:**

1. The empirical validation through experiments is limited, lacking comparisons with other state-of-the-art algorithms to support the theoretical claims.
2. The theoretical proofs need additional explanations or visual aids, with a high-level overview before detailed proofs to improve accessibility for a broader audience.

**Questions:**

1. How does the performance of MORL scale with the number of tasks and the size of the state and action spaces?
2. Are there any practical considerations or limitations in applying MORL to real-world datasets that were not covered in the paper?

**Limitations:**

The authors have adequately addressed the limitations of their work, particularly in terms of the assumptions required for the theoretical analysis. However, more discussion on the potential practical limitations and how they can be mitigated would be beneficial.

---

> ### Author Rebuttal · Authors · 2024-08-07
>
> We thank the reviewer for their valuable time and effort in providing detailed feedback on our work. We hope our response will fully address all of the reviewer's points.
>
> *  As discussed in Additional Related Work in Appendix A under **Offline Data Sharing in RL**, there have been numerous empirical works that studied the benefit of using offline datasets from multiple tasks to accelerate downstream learning. The aim of this paper is to provide a framework for investigating this approach from a theoretical lens. While empirical validation is limited in this paper, we hope that the theoretical analysis presented in this work will provide insights for algorithmic design principles that can be used to design empirically better performing algorithms compared to current state-of-the-art algorithms.
>
> *  Thanks a lot for your suggestion regarding improving the accessibility of the proofs for a broader audience. We will add a proof roadmap section along with a visual proof roadmap in our next version. Again, thanks a lot for the great suggestion!
>
> ### Questions
>
> 1.  As can be observed from Equation 3.7, the sub-optimality gap (in average sense) of MORL does not depend on the size of the state and action space. Instead it depends on $d$ - the dimension of the feature $\phi$. Moreover, the average accuracy of the estimated transition kernels of the $T$ MDPs scales at the rate $\tilde{O}(\sqrt{\frac{\log |\Phi|}{nT} + \frac{\log|\Psi|}{n}})$.
>
> 2. In our setting we assumed that the different tasks share the feature representation $\phi$ under low-rank MDP structure. However, in most real-world datasets this might be difficult to satisfy exactly. It would be an interesting future work to study how we can extend this work to the case where the features are not exactly shared but instead shared with some perturbed approximation.
>
> Finally, we are glad that the reviewer felt that we have adequately addressed the limitations of the work in terms of the assumptions used for theoretical analysis. As per the suggestion of the reviewer, we will add more discussion on potential practical limitations of MORL along with how to possibly mitigate them in our next version.
>
> We hope we have addressed all of your questions/concerns. If you have any further questions, we would be more than happy to answer them and if you don’t, would you kindly consider increasing your score?

---

> > ### Comment · Reviewer_fKn9 · 2024-08-11
> > **Official Comment by Reviewer fKn9**
> >
> > Thank you for your detailed rebuttal and for addressing the points I raised in my review. I appreciate your clarification on the scalability of the MORL algorithm, particularly that the sub-optimality gap depends on the feature dimension rather than the state and action space sizes. I have also reviewed the other feedback and acknowledge the broader concerns about the paper's clarity. Your plan to include a proof roadmap and visual aids in the revised version is a positive step that will improve readability. I believe your responses have effectively addressed my concerns, and while I will maintain my current score, I look forward to the improvements in the final version.

---

> > > ### Author Response · Authors · 2024-08-12
> > > **Reply to reviewer fKn9**
> > >
> > > Dear reviewer fKn9,
> > >
> > > Thank you for spending time checking our rebuttal. We promise to incorporate the suggestions you made ( a proof roadmap and visual aids) in the final version, and we authors are still available here to answer your questions until tomorrow in case you have any final questions.
> > >
> > > Best, Authors

---

### Official Review · Reviewer_Z4xa · 2024-07-08

**Soundness:** 3
**Presentation:** 3
**Contribution:** 3
**Rating:** 7
**Confidence:** 1

**Summary:**

This paper provides a theoretical analysis of representation learning in Multi-Task Offline RL.
Specifically they consider a setting in which the transition kernels have a low-rank decomposition $P(s'|s,a)=\langle \phi(s'),\mu(s,a)\rangle$, such that $\mu(s,a)$ depends on the task but $\phi(s')$ is shared by all tasks.
The authors propose an algorithm for this setting that first learns a transition model and then does planning on the model with a low-confidence-penalized reward.
They provide a suboptimality bound, based on bounding the TV distance between estimated and true transition kernels.
The authors also investigate the application of this acquired representation in down-stream online, offline and reward-free tasks.

**Strengths:**

* The paper is very well written, assumptions that are made are explained and well justified.
 * The exploration of down-stream applications of Multi-Task Offline RL is of large practical interest.

**Weaknesses:**

**I don't work in theory, please see the below more as minor comments.**

 * [89] analyzes a very similar setting but is only mentioned in the appendix as concurrent work. Strictly speaking, [89] was uploaded ot arxiv three months before the deadline, so it is not considered concurrent as per NeurIPS guidelines. Either way, it should probably be discussed more prominently and not just in the Appendix.

 * It would have been nice to specify which improvements in the bound are results of which decisions. For example, a conservative reward $r-b$ in planning vs the normal reward $r$, or how exactly using the low-rank assumption rather than a linearity assumption affects the analysis.
Similarly it would be nice to provide an interpretation for the different terms in (3.7), but I'm also not sure if that is really possible to do in a meaningful way.

 * The paper treats a setting with non-stationary reward and transition functions. It's is not discussed why the paper treats this setting and what effect it has on the analysis.

* The approach name and acronym MORL (Multitask Offline Representation Learning) is a somewhat confusing choice, as MORL is frequently used to denote Multi-Objective RL.

* The notation is somewhat confusing at times with $t$ representing tasks, $h$ representing timesteps in the MDP, $\tau$ indexing transitions in the downstream offline dataset and $i$ indexing samples in the upstream dataset. The usage of non-stationary policies, rewards,  and transition functions also does not help to reduce subscripts.

* The checklist instruction part was meant to be removed before submission, i.e. the instructions in L1253,1254

**Questions:**

* Why was a nonstationary setting analyzed, and how did this affect the analysis? Is it possible to achieve a better regret by considering a stationary reward and transition function?

**Limitations:**

Assumptions are discussed sufficiently, so I have no concerns about additional limitations.

---

> ### Author Rebuttal · Authors · 2024-08-07
>
> We thank the reviewer for their valuable time and effort in providing detailed feedback on our work. We hope our response will fully address all of the reviewer's points.
>
> * Thanks for your suggestion regarding [89]. A preprint version of our results appeared publicly around at the same time of the preprint of [89]. We will provide a more detailed discussion of it and move the discussion from the Appendix to the main paper.
>
> *  A conservative reward $r-b$ allows us to show near-pessimism in the average sense as depicted in Lemma 3.5. This is possible through a concentration argument for the penalty term $\hat{b}_h^{(t)}$ as defined in Equation (3.4). Under linearity assumption as in linear MDP, the representation $\phi^*$ is assumed to be known a priori. This allows one to use linear regression analysis to derive point-wise model uncertainty quantification and that can serve as a conservative penalty. However, this is not the case under low-rank assumption. It would be interesting to see how we can interpret each term in Equation 3.7. But as the reviewer mentioned, it is unclear how to do it in a meaningful way. Our motivation for representing Equation 3.7 in this way was to provide a fine-grained view for each term.
>
> *  We considered non-stationary reward and transition functions in our work as it is a standard practice in literature that assumes episodic MDP with finite horizon that have either low-rank MDP or linear MDP structure. Without non-stationarity, the dependency on $H$ would change in the resulting bounds.  Indeed, with stationary reward and transition function one can achieve a better regret bound in terms of $H$ dependency.
>
> *  Thanks for pointing out potential confusion with the acronym MORL. We will try to come up with an alternative acronym in the next version.
>
> *  We totally understand that it can be challenging to follow the notations sometimes as there are so many moving parts involved that add up in the form of subscript and superscript. We tried our best to keep things concise and clear as much as possible. We would be happy to incorporate any suggestion that the reviewer might have to make the notations easier to follow.
>
>
> We hope we have addressed all of your questions/concerns. If you have any further questions, we would be more than happy to answer them and if you don’t, would you kindly consider increasing your score?

---

> > ### Comment · Reviewer_Z4xa · 2024-08-08
> >
> > I would like to thank the authors for their additional explanation, especially of the assumptions made in the problem setting and their consequences.
> >
> > I will raise my score from 6 to 7, as this paper seems useful and novel to me.
> > However, I am keeping my confidence at 1, as I am unfortunately not very familiar with this area.

---

> > > ### Author Response · Authors · 2024-08-11
> > > **Thank you!**
> > >
> > > We thank the reviewer for their positive review and increasing the score.

---

### Official Review · Reviewer_Qe4K · 2024-07-20

**Soundness:** 3
**Presentation:** 3
**Contribution:** 2
**Rating:** 5
**Confidence:** 2

**Summary:**

This paper proposes a representation learning algorithm for offline multitask reinforcement learning. The proposed algorithm, MORL, is designed for offline multitask RL in low-rank MDPs. The learned representation is examined in downstream RL in reward-free, offline, and online scenarios.

**Strengths:**

1. Detailed theoretical analysis of the proposed algorithm.
1. Better sample complexity than existing algorithms.

**Weaknesses:**

I do not conduct research in theoretical offline RL, so I cannot provide an accurate evaluation for this paper. I will try to provide some comments based on my understanding.
1. This paper lacks a comparison with other multitask RL algorithms. Table 1 only compares it with single-task methods. Section 4.3 does not provide any comparison with other algorithms.
1. The contributions to downstream online and offline RL are not clear.

Minor issues:
1. Equation (3.2) should be $P(s'|s,a)$... where $s'$ is missing.

**Questions:**

1. In line 295 in Section 4.3, why do you assume that reward function $r^{T+1}$ is unknown?
1. Could you provide any experimental results to support the claims in the paper?

**Limitations:**

The paper lacks a limitations section.

---

> ### Author Rebuttal · Authors · 2024-08-07
>
> We thank the reviewer for their valuable time and effort in providing detailed feedback on our work. We hope our response will fully address all of the reviewer's points.
>
> ### Points raised as weaknesses
>
> 1. To our knowledge, there is only one other concurrent offline multitask RL theory work [89]. We provided detailed comparison with that work in Appendix A: Additional Related Work. However, as described in our discussion the theoretical results of [89] are not directly comparable to our theoretical results. Table 1 only compares with single-task methods as there is no existing work that studies reward-free RL in multi-task settings. The goal of Table 1 is to show that performing offline multitask representation learning in the upstream task, can help improve sample complexity of downstream reward-free tasks compared to single task counterparts. For offline and online downstream tasks we provided comparison with other algorithms in Remark K.7 (Appendix K.2) and Remark M.6 (Appendix M.2). We provided them in the appendix due to space constraints in the main paper.
>
> 2. As we mentioned in the Introduction section where we highlighted our contributions, for the downstream part, our core contribution is in the reward-free setting. We provided results for offline and online downstream settings as complementary results.
>
> ### Questions
> 1. In Section 4.3, for downstream offline and online RL tasks, we assume the reward function $r^{T+1}$ to be unknown. We made this assumption to make our result more general. Note that, knowing the reward function $r^{T+1}$ would have made the downstream task easier to solve.
>
> 2. As discussed in Additional Related Work in Appendix A under **Offline Data Sharing in RL**, there have been numerous empirical works that studied the benefit of using offline datasets from multiple tasks to accelerate downstream learning. The aim of this paper is to provide a framework for investigating this approach from a theoretical lens. While empirical validation is limited in this paper, we hope that the theoretical analysis presented in this work will provide insights for algorithmic design principles that can be used to design empirically better performing algorithms compared to current state-of-the-art algorithms.
>
>
> ### Limitations
>
> As all other reviewers have acknowledged, we have addressed limitations of our works, especially through a thorough discussion of the assumptions made for the theoretical analysis. Moreover, as per the suggestion of the Reviewer fKn9, we will add more discussion on potential practical limitations of MORL along with how to possibly mitigate them in our next version.
>
>
> We hope we have addressed all of your questions/concerns. If you have any further questions, we would be more than happy to answer them and if you don’t, would you kindly consider increasing your score?

---

> > ### Comment · Reviewer_Qe4K · 2024-08-12
> >
> > Thanks for the authors' response. My concerns are addressed in the rebuttal. I have raised my score to 5 accordingly.

---

> > > ### Author Response · Authors · 2024-08-12
> > > **Reply to reviewer Qe4K**
> > >
> > > Dear reviewer Qe4K,
> > >
> > > Thank you for checking our rebuttal. We authors are still available here to answer your questions until tomorrow in case you have any last-minute questions.
> > >
> > > Best, Authors

---

### Decision · Program_Chairs · 2024-09-25

**Decision:**

Accept (poster)

**Comment:**

Although there has been much empirical work in representation learning for offline multitask RL, there has been relatively little theoretical work. This paper helps close this gap by developing a novel theoretical framework to address this problem, specifically focusing on low-rank MDPs. The paper overall seems technically solid, makes good contributions to the field, and is likely to have a solid impact. Its framework is well developed and supported by extensive proofs. As pointed out in the reviews, there are two shortcomings of this paper that would improve it — one semi-major and one minor: 1) the semi-major issue is that the paper lacks empirical validation of the theoretical ideas. Adding this would help connect the work to the well-established and complementary empirical literature on the same topic, and help establish some practical implications of the developed theory. 2) As a minor point, the paper would benefit from more intuitive explanations and illustrations to help broaden its appeal beyond the pure theory audience.